



# Technical note: A framework for causal inference applied to solar radiation and temperature effects on dissolved gaseous mercury

Hans-Martin Heyn[1] and Michelle Nerentorp Mastromonaco[2]

[1]Computer Science and Engineering, University of Gothenburg and Chalmers, 40530 Göteborg, Sweden
[2]IVL Swedish Environmental Research Institute, 40014 Göteborg, Sweden

**Correspondence:** Hans-Martin Heyn (hans-martin.heyn@gu.se)

**Abstract.**

Environmental science usually requires researchers to rely on observational data alone. However, researchers want to identify causal relationships and not only correlations between pollutant behaviour and other environmental factors such as weather. Previously it has been shown that solar radiation associates with the volatilisation and evasion of the hazardous pollutant mercury from sea surfaces into the atmosphere. Statistical and machine learning methods can help find and quantify such associations. However, association does not imply causation, and inferring causal relationships from observational data alone remains a significant challenge. Here, we aim to create an 'easy-to-follow' framework, to be used by environmental researchers, for using prior scientific knowledge encoded as graphical causal models to enable causal inference and to estimate effect sizes of different related factors using collected field data. We demonstrate the framework through a case study estimating the effect sizes of solar radiation and sea surface temperature on dissolved gaseous mercury (DGM) in seawater measured at the west coast of Sweden. Our causal analysis reveals that 32% of the total effect of solar radiation on DGM is mediated indirectly via changes in sea surface temperature. Wind and instrumentation acted as confounders, biasing effect estimates by 4.5%. Results from the case study show that our proposed framework allows for a rigorous design, validation, and reporting of causal inference in environmental science. It shows potential in modelling causes of pollutant dynamics and quantifying the effect of regulating policies such as the Minamata Convention For Mercury.

## 1 Introduction

Environmental, and particularly atmospheric monitoring and research, rely on direct measurements and modelling to understand the processes involved in the formation, transport, evasion, and deposition of different pollutants. The purpose of environmental monitoring is often defined and driven by national and international legislations and directives, defined within the European Union (Council of the EU, 2004), or through international conventions such as the United Nations Framework Convention on Climate Change (United Nations, 1992) and the Convention on Long-range Transboundary Air Pollution CLRTAP (United Nations Economic Commission for Europe, 2024; United Nations, 1979). Depending on legislative and pollutant, the requirements for monitoring could include either modelling, estimations, or direct and continuous measurements with yearly reporting to co-operative programs under CLRTAP, e.g., the European Monitoring and Evaluation Programme (EMEP,





2023). For mercury, directive 2004/107/EG describes the need of ensuring collecting publicly available information about concentrations in air and deposition in every Member State of the European Union (European Parliament and Council, 2004). The directive further demands that indicative measurements of mercury shall be performed at background sites at a spatial resolution of 100 000 km$^2$. Within the United Nations, a global convention, aimed to protect human health and the environment from the exposure of mercury, was agreed on in January 2013 and was named the Minamata Convention on Mercury (United Nations Environment Programme, 2013). The convention entered into force in August 2017 and is today ratified by more than 150 parties all over the world. At present, an Effectiveness Evaluation Group has been selected to assess the effectiveness of the convention by studying trends and changes in mercury concentrations in different media (United Nations Environment Programme, 2024).

This leads to another aspect of environmental monitoring: the use of computer models to analyse and interpret complex environmental data to make predictions about unseen or future data points, to understand patterns and trends, and to evaluate the effectiveness of current legislatives. Learning about the dependencies between variables from observational data allows us to build predictors that provide necessary estimates of previously unseen data. These predictors are statistical learning machines, which can take the form of simple regression models or even highly complex and opaque ML models such as convolutional neural networks (CNNs). However, environmental researchers are not only interested in building opaque, black-box prediction machines that magically predict future data points from observational data. Instead, they are particularly interested in understanding cause-effect relationships to suggest interventions that reduce pollutants in the environment. Causal knowledge, or in other words the analysis of cause-effect relationships, is one of the 'fundamental goals of science' (Vowels et al., 2022; Rose and van der Laan, 2011).

This paper reports the results of a case study on extracting causal knowledge about the contribution of different environmental processes to the formation of dissolved gaseous mercury (DGM) in seawater and subsequent mercury evasion to air from observational data. Although measurements of gaseous mercury (Hg) in water is not yet a requirement within EU directives, the results from novel continuous measurements of Hg in surface waters are used as an example. This paper proposes a framework for obtaining and reporting causal knowledge from environmental observational data. Through a case study, this paper explains how to build statistical models that not only predict future values of an outcome variable but also allow the inference of causal relationships between predictor variables and the outcome using observational data.

## 1.1 Environmental monitoring of mercury emissions

Mercury is considered by the World Health Organization to be one of the top ten chemicals or substances to be of major concern to public health (Cohen et al., 2005). This volatile and toxic element is found naturally in various environmental compartments and originates from both natural and anthropogenic sources, such as artisanal small scale gold mining, burning of fossil fuels and various industrial activities. As a gas, mercury is stable and has a long residence time in air, resulting in a global spread via the atmosphere to remote, pristine, and vulnerable environments such as the polar regions. Mercury is deposited from air by dry and wet deposition, often via oxidation from Hg$^0$ to Hg$^{II}$. Oxidised mercury in seawater, for example, transforms more easily into methylmercury which can bioaccumulate in aquatic food chains. However, it can also



reduce back to its elemental form and evade back into the atmosphere, where it is capable of fast long-range transport. Mercury

evasion from sea surfaces accounts for almost 50% of the annual contributions to the atmospheric mercury load. This is because much of the oceans' surfaces are supersaturated with elemental mercury compared to the atmosphere, resulting in net water-to-air evaporation (AMAP, 2021). Understanding the drivers behind formation of DGM and subsequent flux is key to understand the bioavailability of methylmercury in seawater and supporting global models with information about spatial and temporal variability (Soerensen et al., 2013). Mercury flux models for seawater suggest that the flux, and thus the fluctuation

of DGM concentration in surface waters, is mostly influenced by environmental factors such as wind speed, temperature, photochemistry and microbial activity (Soerensen et al., 2013; Johnson, 2010; Kuss et al., 2009). What controls the formation of DGM is also debated, and it is discussed that it is formed by demethylation processes of methyl- and dimethylmercury in the subsurface ocean (Munson et al., 2018). Demethylation could be either abiotic or biotic, with an abiotic process being photo-demethylation controlled by solar radiation (AMAP, 2021). The connection between the formation of DGM and solar

radiation has previously been studied in various environmental compartments such as the sea, lakes, soils, and salt marshes (Xie et al., 2019; Sizmur et al., 2017; Dill et al., 2006; Gårdfeldt et al., 2001; Amyot et al., 1997). In several studies, the relationship between DGM and solar radiation was quantified by determining correlation coefficients of 0.66 (Cane Creek Lake, USA; Dill et al. (2006)), 0.99 (river near Knobesholm, Sweden; Gårdfeldt et al. (2001)), and 0.39 (coastal Minamata Bay, Japan; Marumoto and Imai (2015)), suggesting that solar radiation can be an important, though site-dependent, predictor of DGM.

Other studies report similar relationships between mercury evasion and solar radiation with correlation coefficients of 0.7, (Adriatic Sea; Floreani et al. (2019)) and 0.5-0.9 (Wuijang River, China; Fu et al. (2013)). However, there is probably more to the story of how the concentration of DGM is influenced by external factors such as solar radiation. Sizmur et al. (2017) hypothesised that the formation of DGM probably is affected by a combination of solar radiation and increased temperature. Zhang et al. (2006) performed a Pearson analysis of DGM and various factors measured at Lake St. Pierre (Canada), including

air and water temperature, wind speed and solar radiation. The analysis showed significant correlations between DGM and all of these factors. Other aspects, such as water depth, have also been shown to have an effect on how strong the influence of solar radiation is for the formation of DGM in surface seawater, since the correlation mainly has been observed at the coast (Nerentorp Mastromonaco et al., 2017; Fantozzi et al., 2013; Ferrara et al., 2003; Lanzillotta et al., 2002; Andersson et al., 2007). The reason has been suggested to be due to greater vertical and turbulence mixing (Nerentorp Mastromonaco et al.,

2017; Lanzillotta et al., 2002), lower friction velocities and surface roughness  (Fantozzi et al., 2013), and the presence of dissolved organic carbon (DOC) and suspended particles  (Ferrara et al., 2003; Amyot et al., 1997).

In summary, we see that merely calculating and discussing correlations will not capture the underlying causal mechanisms by which different environmental forcings influence DGM. What is needed instead is an approach that can separate correlation from causation while also accounting for cause–effect relationships among the forcings themselves, such as solar radiation

influencing temperature.



## 1.2 Outline and purpose of the paper

The intention of this paper is to provide a discussion of the role of graphical causal models in environmental research and to present suggestions on how effect sizes from observational data in environmental science can be systematically obtained and reported.

In Section 2, the paper describes the used case study of continuous measurements of DGM in seawater, carried out on the west coast of Sweden in 2020. Section 3 introduces a framework for causal inference using observational data. Section 4 then describes how the proposed framework for causal inference is applied to the case data for inferring the effect sizes of different forcing, such as solar radiation, sea surface temperature, wind speed, and speed of the instrument feeding water pump on measured Hg concentrations $C_{MW}$. Section 5 presents the results from the case study and in Section 6 these results and the

application of the framework for causal inference using observational environmental data are discussed, leading to concluding remarks and suggestions for further research presented in Section 7.

## 2 Description of case study: Continuous measurements of Hg concentration in seawater

The measurement campaign was conducted from 2019-12-05 to 2020-10-08 at the Kristineberg Marine Research Station, located on the Swedish West-coast (58.25013°N, 11.44485°E). Kristineberg is located at the entry of the Gullmarsfjord in

the Skagerrak Sea which is classified as a natural reserve. With its shallow waters it serves as an important reproduction site for shellfish. The data for this study were collected during the period 2024-04-01 to 2024-04-25, which is an interesting time period for our case study due to the good mixture between dark and sunlit hours in Scandinavia at this time of the year.

### 2.1 Measurements of Hg concentration in surface water

The experimental set-up for measuring DGM in surface seawater is presented in Figure 1. The measurements were conducted

in shallow waters (<10 m depth) at the harbour of Kristineberg. At the sampling site, a commercial 12V bilge pump[1] with the capacity to pump 32 l/min was installed on a chain, attached to a buoy and an anchor, at 1 m below the buoy to keep the same water depth to the surface, disregarding tide or waves, see Figure 1 a). A cage of fine mesh net was installed around the pump to reduce pump clogging problems due to algal growth (not shown in figure). The seawater was pumped from the sampling site to the measurement site through a rubber tubing, lying on the bottom of the sea. The distance between the sampling site and the

measurement site was less than 10 m. At the measurement site (Figure 1 b)), a weather protected box shielded all equipment for measuring Hg and other parameters. A temperature controlled radiator kept the equipment at a constant temperature. The flow of incoming seawater from the measuring site was measured and controlled using a water meter and a valve. The regulated water flow in Figure 1 is denoted $r_W$. A constant water flow was hard to obtain due to the pump experienced problems with growth of algae and clogging, despite installing a protective net. During April 2020, $r_w$ varied between 0 and 40 l/min, with

an average flow rate of 2.8 l/min.

---

[1]Art.no. 25-9741, Biltema





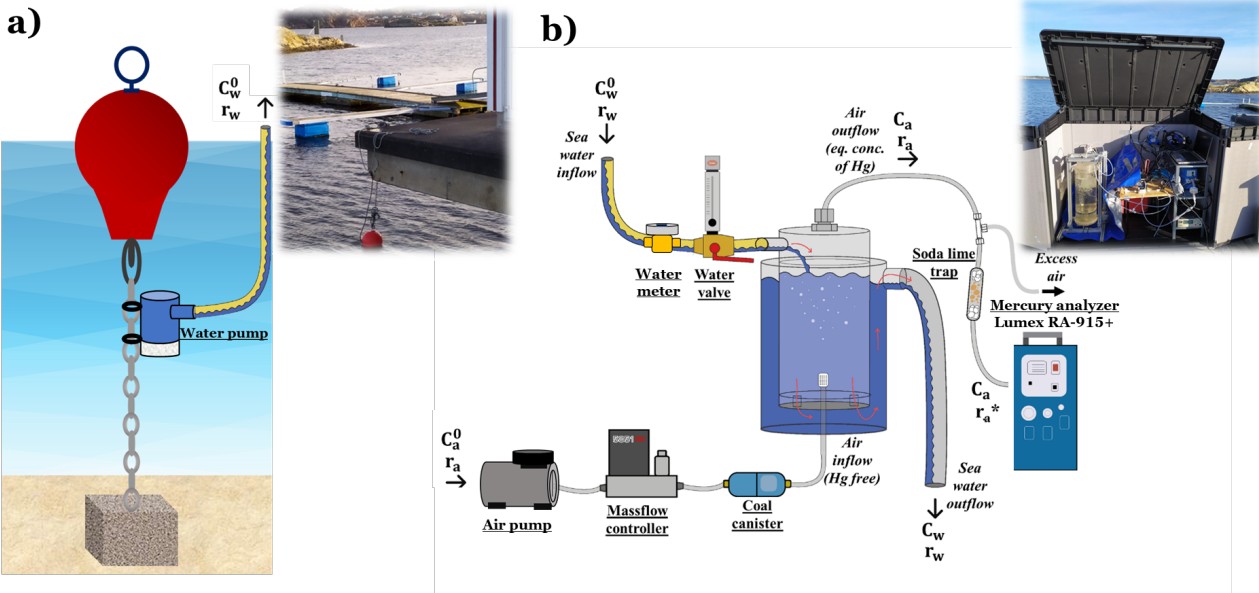

**Figure 1.** Experimental set-up for measuring gaseous mercury in surface water: a) sampling site consisting of a red buoy which holds up a chain that is attached to the bottom with an anchor. The water pump is fixated on the chain about 1 m below the buoy; b) measurement site containing the continuous equilibrium system for extracting gaseous mercury from seawater and the Lumex RA-914+ mercury analyser

An automated continuous equilibrium system was used to continuously extract gaseous mercury from incoming seawater. The system, developed by Andersson et al. (2008b) and further used in e.g., Nerentorp Mastromonaco et al. (2017) and in Osterwalder et al. (2020), consist of an inner cylinder in which incoming seawater enters continuously from the top. A purging system, consisting of a glass frit, was installed at the bottom of the inner cylinder. The air used for purging was pumped using an air pump. A massflow controller regulated the air flow ($r_a$) to a fixated 1.5 l/min. Ambient air normally contains small amount of Hg ($c_{a0}$ in Figure 1b), but in our set-up, a coal canister was used to remove Hg from the purging air.

After purging through the water in the inner cylinder, the air outflow, now containing the equilibrium concentration of extracted gaseous mercury ($C_{MW}$), was first led through a soda lime trap and a polytetrafluoroethylene (PTFE, 2 $\mu$m pore size, 47 mm diameter) filter to prevent particles and moisture from entering the analyser. The purged sea water flowed out via the bottom of the inner cylinder, moving upwards inside the outer cylinder and was then discharged back to the sea via a rubber tubing. The purpose of the backflow system using an outer cylinder is to serve as isolation to keep the temperature in the inner cylinder as stable as possible during purging. All small tubing in the system was made of FEP (fluorinated ethylene propylene).

Finally, the equilibrium concentration of extracted gaseous mercury ($C_{MW}$) in the air outflow was analysed every minute using a Lumex RA-914+ atomic absorbance spectrometer. Calibration of the instrument was performed automatically every three hours by adjusting the baseline using mercury free air. The detection limit of the instrument has previously been determined to be 0.5 ng/m3, having a precision of $\pm5\%$ (Nerentorp Mastromonaco et al., 2017). The blank of the automated continuous





equilibrium system was determined by turning off the water pump, letting water within the system be depleted of gaseous Hg. The detection limit of the system was determined to 0.32 pg/L, calculated as three times the standard deviation of the blanks.

$C_{MW}$ measured with the analyser can be used to calculate the concentration of dissolved gaseous mercury (DGM) in incoming seawater. If $C_{a0}$ is removed from Hg, the equation to calculate DGM can be simplified to: (Andersson et al., 2008b):

$$\text{DGM} = C_{MW}\left(\frac{1}{H'} + \frac{r_A}{r_w}\right), \tag{1}$$

where $C_{MW}$ is the measured Hg concentration in the air outflow from the purging system, $H'$ is the dimensionless Henry's law constant that describes the partitioning of mercury between the gaseous and aqueous phase. The variables $r_A$ and $r_w$ denote the flow rates of purging air and seawater, respectively. Henry's law constant for mercury in seawater has previously been determined by Andersson et al. (2008a) to be temperature dependent and is calculated as

$$H' = e^{(-2404.3/T(K))+6.29}. \tag{2}$$

## 2.2 Measurements of solar radiation and surface water temperature

Solar radiation, denoted as *Sol*, was measured using a pyranometer sensor from Renke, model RS-TBQ measuring both direct and diffuse solar irradiance with a measurement range of $0 - 2000$ W/m$^2$, a resolution of 1 W/m$^2$ and an accuracy of $\pm3\%$. The sensor was installed on the roof of the measurement box, having no shadowing. The output of the sensor was read by an Arduino embedded computer which transformed the output voltage into irradiance in W/m$^2$ and sent the result to a central computer for logging alongside the other measurements. The temperature of the incoming seawater was measured at the inlet to the automated continuous equilibrium system using a DS18B20 temperature probe connected and processed by the same Arduino embedded computer.

Other weather data, such as wind speed, were downloaded from a weather station situated in the Gullmarsfjord close to the research station. These measurements are run by the University of Gothenburg and available online (of Gothenburg, 2024).

All variables have been resampled to 5 minutes intervals using a moving average filter. This has been done for two reasons: firstly, we did not suspect that the dynamics of the observed effects are faster than 10 minutes, which is twice the new sampling rate and therefore fulfils the Nyquist–Shannon sampling theorem. Secondly, we reduced the amount of datapoints which allowed for faster processing in the modelling steps.

## 3 A framework for causal inference using observational data in environmental research

The workflow to estimate effect sizes in this study is presented as a framework for researchers conducting causal inference from observational data. It builds on existing causal inference workflows proposed in behavioural science (Deffner et al., 2022) and software engineering (Furia et al., 2024). However, unlike these domains, environmental research relies to a much



higher degree on data from intervention-free observational studies, motivating a tailored framework for causal inference using observational environmental data and prior scientific knowledge from researchers:

---

**A framework for causal inference using observational data in environmental research**

1. Formulate clear research questions.
2. Encode prior scientific knowledge and assumptions as a set of different plausible causal models in the form of directed acyclic graphs (DAGs).
3. Derive independence criteria from the causal models.
4. Generate simulated data based on causal models and identified independence criteria.
5. Build statistical or machine-learning (ML) models for each alternative causal model.
6. Verify the statistical or ML models on the simulated data and show that the models can estimate given parameters and identify independence relations in simulated data.
7. Run the models on observational data.
8. Check independence criteria and parameter estimates.
9. Validate the plausibility, workability, and adequacy of the statistical / ML models.
10. Report and interpret the results.

---

The proposed framework consists of ten steps which can be divided into three groups: steps for incorporating and testing prior knowledge (steps 1, 2, 3, and 8), steps for collecting evidence of plausibility, workability, and adequacy (steps 4, 6, and 9), and steps for generating answers to research questions (steps 5, 7, and 10). The steps are interlinked, which suggests an order in which we propose to execute them as shown in Figure 2.

In steps 1-3, research questions are formulated, and prior scientific knowledge and assumptions are encoded in a set of possible graphical causal models. A causal model describes "relevant features of the world and how they interact with each other" (Pearl et al., 2016). Which features are relevant depends on the research questions and scope of the investigation. A causal model indicates how a change in one variable changes the value of another variable, regardless of the functional form of the relationship. Causal models include two types of variables. The first type, exogenous variables, are not explained further within the model. For example, solar radiation $Sol$ affects ocean surface temperature $T_S$ ($Sol \rightarrow T_S$). Here, we treat $Sol$ as an exogenous variable without modelling upstream mechanisms such as the sun's fusion reactions. The second type, endogenous variables, by contrast, are always descendants of at least one exogenous variable. Directed Acyclic Graphs (DAGs) provide an accessible visualisation of causal models with nodes representing the variables of a system of interest and directed edges representing the direction of cause-and-effect. The causal model of the earlier example $Sol \rightarrow T_S$ is in the form of a DAG, with the arrow $\rightarrow$ indicating that solar radiation is a cause of changes in surface temperature, and not the other way around. The graphical representation of causal models through DAGs is qualitative, i.e., it provides information about the direction of cause-and-effects between variables, but it does not provide information about the strength or functional properties of the causal relationships. Pearce and Lawlor (2016) provide an overview of properties of DAGs representing causal models:





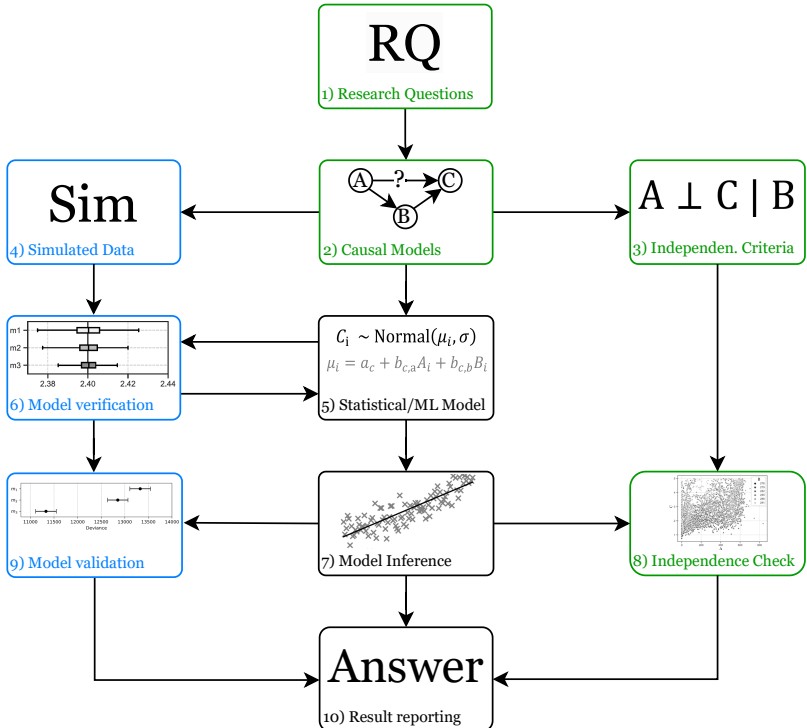

**Figure 2.** A workflow for causal inference of statistical models using observational data and prior knowledge about plausible cause-effect relationships.

– DAGs are directed, i.e., the arrows can only have a single head pointing from the cause towards the effect.

– DAGs are acyclic, meaning that there must not be a series of arrows (i.e., a path) resulting in a cyclic path. Causal models are acyclic because a variable at a given point in time cannot be the cause of itself.

– An arrow indicates that (we assume) a causal relationship exists between two variables, where the first variable has an effect on the second variable.

– The absence of an arrow indicates that the values of two variables are not causally related, i.e., a change in the value of one variable has no effect on the value of the second variable.

– The outcome variable is typically an endogenous variable which is the target of an investigation and effect of interest which value is influenced by parent variables of the causal system.

– The exposure variables represent causes of interest. They are usually nodes with outgoing edges. Exposure variables are exogenous variables in a causal model if they have only outgoing edges. Otherwise, they are endogenous variables which value are being influenced by other variables, such as for example sensor noise effects. Exogenous variables are always root nodes in DAGs, i.e., they do not have incoming edges.





It is necessary to define the direction of cause-and-effect a-priori, because statistical models cannot distinguish between cause and effect as they only identify association but not causation. If the direction of cause and effect is not known, or if the existence of a causal relationship is uncertain a-priori, several alternative causal models can be proposed. Based on the proposed causal models, independence criteria are derived using mathematical methods such as d-separation (Pearl et al., 2016). Tools exist, such as DAGitty (Textor et al., 2016) that automatically derive these independence criteria from graphical causal models.

In Step 4, simulated data are generated for each of the proposed causal models. These simulations emulate the suggested causal relationships to create artificial outcome variable data. Then, in Step 5, the statistical models or ML models are defined. The prior knowledge of causal relationships can be used to define the statistical models accordingly, which may result in several different possible statistical models representing different prior assumptions. Some statistical models, such as generalised linear models (GLMs) for Bayesian data analysis (BDA), allow the explicit inclusion of additional a-priori knowledge for example through the selection of predictor variables or the choice of prior distributions. In Step 6, the correctness of the models is verified on simulated data. Each model is verified against each of the simulated data sets, where each data set represents one of the possible causal models. It is tested whether the models can correctly represent prior knowledge (e.g., through prior predictive plots) and whether the models can infer the effect sizes that were set to generate the simulated outcome data. This not only tests whether the models work as intended, but also how the different models behave under different assumed causal relationships.

The models are run on the observed data in Step 7. Then, in Step 8, the inferred parameters are used, to check for statistical independence between variables. This can be accompanied by manual independence checks such as scatter plots. The results of the independence check are compared with the previously established independence criteria to provide evidence for a final candidate model. Step 9 then aims to collect evidence for the plausibility, workability, and adequacy of the models. Plausibility can be established by checking that the defining parameters and settings of the models, such as the choice of a-priori distributions, follow the prior knowledge of the researchers. Workability can be assessed by checking that the models can infer results without encountering computational problems, such as numerical difficulties or divergences. Adequacy of the models can be established by inspecting posterior predictive plots and checking that the models can infer the outcome variable reasonably well. The different proposed models can be compared in terms of predictive capability (e.g., using divergence measures). However, the model with the highest predictive power is not necessarily the model that allows for the extraction of correct causal knowledge from the estimated parameters which is further discussed as supplementary material in Appendix D.

In Step 10, a causal knowledge is extracted from the statistical models and possible answers to the research questions are given. In addition to the outcome of the model inference itself, the answer should also report on the validation results of the models, the outcome of the independence checks, and the included prior assumptions and knowledge encoded in the causal models. In the analysis and discussion, we have assumed that the regression coefficients can be interpreted as effect sizes. This assumption requires the use of linear models[2] and it requires that the underlying causal model is correct. The latter is the reason why it is important that prior knowledge and assumptions in the form of a causal model are communicated as part of the research answer. Without knowing the underlying assumed causal model, it is impossible to judge whether the regression

---

[2]see Pearl et al. (2016) for a detailed explanation of why regression coefficients can be treated as effect sizes in linear models




coefficients can be correctly interpreted as effect sizes. It is a combination of all the elements in this framework that allow for the extraction of causal knowledge from the inference results: a) prior knowledge and assumptions, b) simulation, verification, and validation outcomes, and c) inference results from the models.

## 4 Applying the framework for causal inference using observational data in environmental research to inferring the effect of solar radiation on $C_{MW}$

In this section we explore the use of the proposed framework for causal inference using observational data in environmental research described in Section 3 by following workflow outlined in Figure 2 using observational data from the case study described in Section 2.

### 4.1 Step 1: Formulate clear research questions.

The observed association between two variables may be the result of several competing causal relationships between other
observed and unobserved variables. Researchers may then be interested in identifying the direct effect and the total effect between an observed variable and an observed outcome. The difference is that the direct effect quantifies only the part of the association that relates to a direct influence of the variable of interest on the outcome. The total effect, however, also includes associations that are due to indirect effects which, for example, are mediated by other variables.

---

**Definition of direct and indirect effect**

Figure 3 illustrates the difference between the direct effect and the indirect effect. Variable $A$ has a *direct effect* on variable $B$:

$$\text{Direct Effect}_{A \to B} = b_{b,a}. \tag{3}$$

But, there is also a second causal path from $A$ to $B$, mediated by a third variable $C$. This leads to an additional *indirect effect* of $A$ on $B$, mediated by $C$:

$$\text{Indirect Effect}_{A \to B} = b_{c,a} \cdot b_{b,c}. \tag{4}$$

The *total effect* of $A$ on $B$ is the sum of *direct* and *indirect effect*:

$$\text{Total Effect}_{A \to B} = \text{Direct Effect}_{A \to B} + \text{Indirect Effect}_{A \to B}. \tag{5}$$

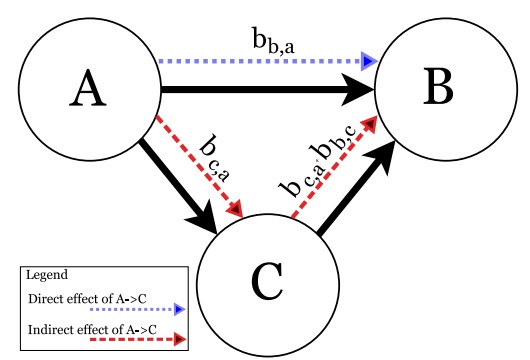

**Figure 3.** Example of direct and indirect effect.

---

250 Causal models allow us to distinguish between a direct effect, which includes the part of the total effect of a forcing that acts immediately on an outcome, and the indirect effect, which accounts for the share of the effect size that is mediated through another factor.





---

**Definition of the effect size**

The effect size is a quantitative measure of the strength of the relation between two variables and indicates the magnitude of change on one variable given a change in another variable. For example, an effect size of 0 means there is zero change in variable B when variable A is changing, indicating no association between these two. An effect size of 1, on the other hand, means that a one-unit increase in variable A results in an equivalent one-unit increase in variable B. If the two variables are standardised, which is sometimes referred to as z-scores, an effect size of one implies that a one standard deviation increase in variable A leads to a one standard deviation increase in variable B. Effect sizes of standardised variables, therefore, provide a means to compare the relative influence of different variables on one another.

---

Earlier studies investigating the correlations between DGM concentration, solar radiation and temperature have experienced temporal limitations. This study provides data from long-term measurements of gaseous Hg concentration, solar radiation and surface seawater temperature, with the aim to investigate the hypothesis that there exist correlations between these factors. By using causal modelling, this study aims to quantify the direct and indirect effect sizes of solar radiation on Hg concentration in seawater.

---

**Research Question 1**

What is the direct and indirect effect of solar radiation on measured Hg concentration in seawater and subsequently calculated DGM?

---

The presence of additional factors can influence one or more predictor variables and the outcome at the same time can lead to confounding and bias in the estimated effect size between a predictor variable and the outcome. In our observational study, we considered wind, measured as wind speed $W$, and the speed of the water pump, measured as $r_W$, as potential confounding factors. We expected wind to affect $C_{MW}$ by diffusing mercury and to affect $T_S$ through heat transfer and evaporation of water. Furthermore, we know that a fluctuating pump speed can influence the measured Hg concentration $C_{MW}$. We also assume that the pump speed can be influenced by solar radiation and temperature (e.g., through the growth of algae that clog the pump inlet) and by wind which causes waves that can affect the inlet of the pump. The identification of these confounding factors led to the development of an additional model that was also tested in this study. This was done to identify how important they are to our measured $C_{MW}$ and how they would affect the desired study of association between *Sol* and $C_{MW}$. Ultimately, this study aims to use causal and statistical modelling to infer the effect sizes of different forcings, such as solar radiation, sea surface temperature, wind speed, and water flow on $C_{MW}$, and subsequently calculated DGM.

---

**Research Question 2**

What are the effect sizes of additional confounding factors affecting the measured Hg concentration in seawater and subsequently calculated DGM?

---




## 4.2 Step 2: Encode prior scientific knowledge and assumptions as a set of different plausible causal models in the form of DAGs.

In this study, we intended to identify the effect of solar radiation *Sol* on the measured Hg concentration $C_{MW}$. Figure 4 provides

an overview of three plausible models of the cause-effect relationships between the variables in this observational study, given the assumptions that *Sol* always affects $T_S$.

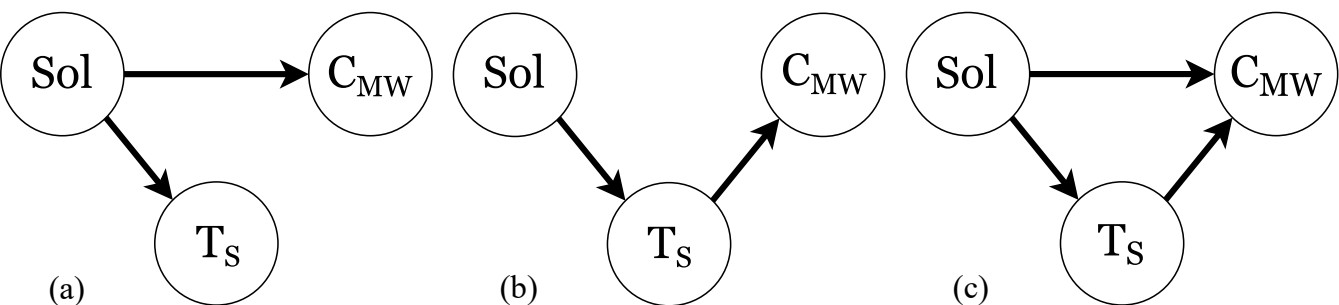

**Figure 4.** Directed Acyclic Graphs (DAGs), which encode assumptions about causal relationships between variables in the dataset.*Sol*: solar radiation; $T_s$: surface temperature of water; $C_{MW}$: Hg concentration. (a) Model $m_1$ representing an effect of *Sol* on $C_{MW}$ and on $T_S$. (b) Model $m_2$ representing an effect of *Sol* on $C_{MW}$ mediated by $T_S$. (c) Model $m_3$ representing an effect of *Sol* on $T_s$ and $C_{MW}$ and an effect of *Sol* on $T_S$.

The identification of the confounding factors wind speed $W$ and water pump speed $r_W$ resulted in the development of a fourth model, $m_4$, presented in Figure 5. We first had to investigate which of the "base" models $m_1$ - $m_3$ is most plausible given the data. Then, we included the external factors in $m_4$ and estimated if the influence of the external factors is so strong

that they could compromise our choice of "base" model[3]

## 4.3 Step 3: Derive independence criteria from the causal models.

The DAGs shown in Figures 4 provide information about plausible cause-and-effect relations and expected associations between the variables of interest. We expect an association between two variables if there is a cause-effect relationship between them[4]. However, association alone does not provide information about the direction of cause and effect. If we find an asso-

ciation between solar radiation (*Sol*) and surface temperature ($T_S$) in the data, we assume from experience that it is the sun radiation that causes the change in surface temperature. On the other hand, the absence of an arrow between two variables suggests that there should be no association in the data. In the DAG depicted in Figure 4 (a), the solar radiation acts as common

---

[3]As an example: Based on the data, we assume there is a causal relationship between *Sol* and $C_{MW}$. Including the external factor $r_W$ (water pump speed) means that some of the effect between *Sol* and $C_{MW}$ can be explained by the influence of $r_W$. We can then estimate how strong this would have to be in order to "take over" the entire cause-effect relationship between *Sol* and $C_{MW}$ and thereby invalidate our previous assumption. If that estimate of the influence of $r_w$ however is unplausibly high, we can assume that *Sol* indeed has a direct effect on $C_{MW}$.

[4]In extraordinary rare circumstances, two causes can perfectly "cancel each other out" which can masks the association between the causes and the common effect (Pearl et al., 2016)





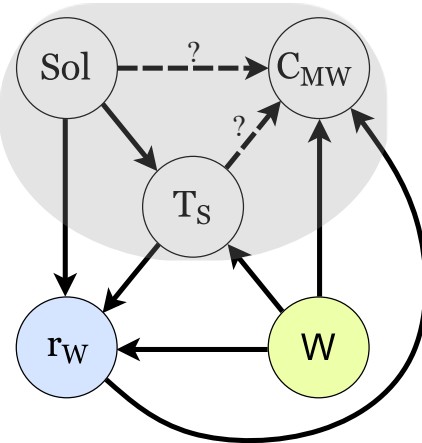

**Figure 5.** Model $m_4$ extending models $m_1 - m_3$ with confounding external factors $W$ representing wind and $r_W$ representing water pump speed.

cause on the measured Hg concentration ($C_{MW}$) and the water surface temperature. Because of *Sol* acting as common cause, we will see a spurious correlation between $C_{MW}$ and $T_S$ in the data. This is called confounding and, in the DAG of Figure 4

(a), *Sol* is therefore a confounding variable. Association can flow "anti-causally", i.e., against the direction of cause-effect, from $T_S$ to *Sol* and from there in the direction of cause to $C_{MW}$. Pearl et Mackenzie describe this as "association that flows in any direction along the edges" (Pearl and Mackenzie, 2018). However, the flow of association between $C_{MW}$ and $T_S$ can be blocked by conditioning on *Sol*. Then, given that we condition the data on *Sol*, $T_S$ and $C_{MW}$ become independent.

However, many researchers encounter the problem that a number of different causal models are possible given the correla-
295 tions seen in the data. In our case, assuming that there is correlation between *Sol*, $T_S$, and $C_{MW}$, any of the cause models shown in Figure 4 is plausible. There are two approaches that combined can help us identify the correct causal model: prior scientific knowledge and testing independence criteria. In fact, the causal models already include prior scientific knowledge by assuming the *directions of cause and effect* because there is no point in adding other compatible models in which the direction of cause and effect is obviously wrong, for example by assuming a change in $T_S$ could causes a change in *Sol*. Each DAG in Figure 4
suggests a set of conditional independence criteria for the research problem that can be used to validate or invalidate the assumptions about cause-effect relationships. Table 1 provides the implied independence criteria for the DAGs of Figure 4. For

**Table 1.** Implied statistical independence criteria based on assumed causal relationships in Figure 4.

| Model | Implied independence criteria |
|-------|-------------------------------|
| $m_1$ | $T_S \perp C_{MW} \mid Sol$ |
| $m_2$ | $Sol \perp C_{MW} \mid T_S$ |
| $m_3$ | No implied statistical independence. |





our study, we used two approaches to test independence, namely visual inspection of scatterplots and regression analysis. We expected the coefficients of the independent variables to be close to zero if the independence assumption holds. Non-parametric measures of association, such as the Spearman's rank correlation coefficient are not well suited to testing conditional independence between three continuous variables because these measures typically assess association, not independence, between only two continuous variables.

### 4.4 Step 4: Generate simulated data based on causal models and identified independence criteria.

Simulated data were generated for each of the proposed causal models. For brevity, the details and results of the simulation are presented as supplementary material in Appendix B.

### 4.5 Step 5: Build statistical or machine-learning (ML) models for each alternative causal model.

**Statistical modelling**

The aim of the statistical modelling for this research was to explore the causal relationships between the variables in the dataset. Initially, we built three Bayesian generalised linear regression models (GLMs) based on the causal models of Figures 4. GLMs are a flexible family of statistical models widely used in empirical science (Furia et al., 2019; Gelman et al., 2020). They extend linear regression methods by introducing link functions and data type specific probability distributions which provides more flexible and accurate modelling of relations between variables in the dataset. We applied Bayesian data analysis (BDA), in combination with linear regression models, as a simple form of iterative ML. It is iterative in the sense that availability of new data in BDA allows for an update, not a complete recalculation, of the inference results. We followed a mathematical description of the statistical model which is standard in statistical journals and outlined in three steps in Chapter 4.2 of McElreath (2020):

- Data are represented as variables. Unobservable characteristics like averages, rates, etc. are defined as parameters. Each variable is either defined through other variables or probability distributions. Initial probability distributions assigned to the parameters are called priors.

- The combination of variables, parameters, and their probability distributions form a joint generative model which can be used to analyse relations between variables, estimate parameters, infer unobserved variables, and predict future observations.

- All predictor variables are standardised to avoid numerical problems due to large values and to allow for negative values (Marquardt, 1980). Standardisation means that all values are centred on zero and that the data have a standard deviation of one. This approach is also known as z-score normalisation.

After we identified model $m_3$ as most appropriate (see Section 5.2), we extended it to model $m4$, shown in Figure 5, by introducing the external confounding elements wind $W$ and pump speed $r_W$.

The mathematical definition of the models are listed in Table 2.



**Table 2.** Model specifications.

| Model | Specification | |
|---|---|---|
| $m_1$: | $C_{MW_i} \sim \text{Normal}(\mu_i, \sigma)$ $\mu_i = a_c + b_{c,s} \cdot Sol_i$ $a_c \sim \text{Normal}(0, 1)$ $b_{c,s} \sim \text{Normal}(0.5, 0.5)$ $\sigma \sim \text{Exponential}(1)$ | (6) |
| $m_2$: | $C_{MW_i} \sim \text{Normal}(\mu_i, \sigma)$ $T_{S_i} \sim \text{Normal}(\nu_i, \tau)$ $\mu_i = a_c + b_{c,t} \cdot T_{S_i}$ $\nu_i = a_t + b_{t,s} \cdot Sol_i$ $a_c \sim \text{Normal}(0, 1)$ $a_t \sim \text{Normal}(0.5, 1)$ $b_{c,t},\, b_{t,s} \sim \text{Normal}(0.5, 0.5)$ $\sigma, \tau \sim \text{Exponential}(1)$ | (7) |
| $m_3$: | $C_{MW_i} \sim \text{Normal}(\mu_i, \sigma)$ $T_{S_i} \sim \text{Normal}(\nu_i, \tau)$ $\mu_i = a_c + b_{c,s} \cdot Sol_i + b_{c,t} \cdot T_{S_i}$ $\nu_i = a_t + b_{t,s} \cdot Sol_i$ $a_c \sim \text{Normal}(0, 1)$ $a_t \sim \text{Normal}(0.5, 1)$ $b_{c,s},\, b_{c,t},\, b_{t,s} \sim \text{Normal}(0.5, 0.5)$ $\sigma, \tau \sim \text{Exponential}(1)$ | (8) |
| $m_4$: | $C_{MW_i} \sim \text{Normal}(\mu_i, \sigma)$ $T_{S_i} \sim \text{Normal}(\nu_i, \tau)$ $r_{W_i} \sim \text{Normal}(\xi_i, \chi)^{\ddagger}$ $\mu_i = a_c + b_{c,s} \cdot Sol_i + b_{c,t} \cdot T_{S_i} + b_{c,w}^{\dagger} \cdot W_i + b_{c,r}^{\ddagger} \cdot r_W$ $\nu_i = a_t + b_{t,s} \cdot Sol_i + b_{t,w}^{\dagger} \cdot W_i$ $\xi_i = a_r + b_{r,s}^{\ddagger} \cdot Sol_i + b_{r,t}^{\ddagger} \cdot T_{S_i} + b_{r,w}^{\dagger\ddagger} \cdot W_i$ $a_c, a_r \sim \text{Normal}(0, 1)$ $a_t \sim \text{Normal}(0.5, 1)$ $b_{c,s},\, b_{c,t},\, b_{c,w}^{\dagger},\, b_{c,r}^{\ddagger},\, b_{t,s},\, b_{t,w}^{\dagger},\, b_{r,s}^{\ddagger},\, b_{r,t}^{\ddagger},\, b_{r,w}^{\ddagger} \sim \text{Normal}(0.5, 0.5)$ $\sigma, \tau, \chi \sim \text{Exponential}(1)$ | (9) |

$^{\dagger}$Terms that relate to the wind speed $W$. $^{\ddagger}$Terms that were relate to the speed-pump $r_w$.

The four resulting statistical models for predicting $C_{MW}$ differ due to different assumptions about cause-effect relationships as shown in Figures 4 and 5. However, the models share a common structure: they contain definitions of the *likelihood* that define probability distributions of the outcome (in our case the measured $C_{MW}$) given a set of parameters that define this probability distribution, for example through the mean $\mu$ and standard deviation $\sigma$ for Normal distributions. Models $m_2$ and

$m_3$ contain two likelihoods because in both models *Sol* affects the surface temperature $T_s$ which then affects the measured





$C_{MW}$. Therefore, we needed to define two linked regression models, one for $T_s$ with *Sol* as the predictor variable, and one for $C_{MW}$ with $T_s$ as the predictor variable for model $m_2$, and both *Sol* and $T_s$ as the predictor variables for model $m_3$. The parameters of the likelihood are declared in the form of either a deterministic function assignment (denoted through $=$) or stochastic relations (denoted through $\sim$), i.e., a mapping of a variable or parameter onto a distribution. When a parameter (e.g.,

$b_{(c,t)}$) is mapped onto a distribution (e.g., the Normal distribution), this distribution is then referred to as a *prior distribution*.

**Variables**

There are four main variables in models: $C_{MW_i}$ represents the measured Hg concentration, associated via Equation (1) to DGM. It is chosen as outcome variable for all models. The index $i$ denotes an individual data frame in the total dataset. $T_{S_i}$ represents the surface temperature of the water and is a predictor variable in models $m_2$ - $m_3$. $Sol_i$ is another predictor variable

in all models and represents solar radiation. In models $m_2$ and $m_3$, $Sol_i$ is also a predictor for $T_{S_i}$. The additional model $m_4$ includes the confounding variables $W$ and $r_W$. The biasing influence of confounding variables on our effect size estimates can be compensated for by including them in the prediction model for $C_{MW_i}$ and $T_{S_i}$. In this way, we "block" any anti-causal flow of association between $T_S$ and $C_{MW}$ via $W$ and $r_W$. The model $m_4$ including the external influences also includes parameters describing the effect size of these external influencing factors $W$ and $r_w$: $b_{c,w}$ and $b_{c,r}$ represents the effect size between $C_{MW}$

and $W$, respectively $r_W$, $b_{t,w}$ and $b_{c,t}$ between $T_S$ and $W$, respectively $r_W$, $b_{r,s}$ between *Sol* and $r_W$, and $b_{r,w}$ between $W$ and $r_W$. The parameter $\xi$ represents the estimated standard variation for the $r_W$ data.

**Likelihood**

The role of the likelihood is to link the observed data to the distributions of the inferred model parameters (Furia et al., 2019). The likelihood provides a distribution function of how well the model explains the observed data given an inferred set of

distributions for the model parameters. We assumed that the observed data can be represented through a Normal distribution because the central limit theorem states that the distribution of the sum of a large number of independent processes approaches a Normal distribution and environmental phenomena typically involve the aggregation of a large number of underlying processes. Furthermore, we had no reason to assume another distribution for the outcome $C_{MW}$. As part of the GLM, the likelihood distribution is further specified through a linear model, i.e., the mean (e.g., $\mu$ is constructed through a linear combination of

other parameters (e.g., $a_c + b_{c,s}$) and predictor variables (e.g., $Sol_i$). This link function between the likelihood and the parameter distribution of the model can take any functional relation based on the prior knowledge of the researcher. We decided to use a linear model because we do not have any prior knowledge that would justify the use of any other functional form, and it represents the most conventional functional relation between an outcome and its predictor variables. Bayes' rule is then used to iteratively calculate a posterior distribution for the parameters of the model given the prior distributions, the observed data,

and considering the likelihood, i.e., how well the currently assumed parameters explain the data:

$$posterior \propto prior \cdot likelihood \tag{10}$$





**Priors**

In general, a *prior* tells researchers what is known about a parameter before they see any observed data. A prior can, but do not have to, incorporate prior knowledge about the probability distribution of the parameter. If researchers choose to incorporate prior knowledge, the prior is referred to as informative prior. A prior can be informative in the sense that researchers assume a certain probability distribution family (e.g., Normal distribution) which they expect the parameter to follow and in the sense of the characteristics (e.g., mean and standard deviation) of that distribution. In Bayesian statistics, it is also possible to use uninformative priors which do not convey any a-prior assumptions about the parameters. It is important to note that priors (informative or not) are continuously updated with the available observed data. With each iteration in BDA, the posterior will be used as new prior for the next iteration. That means, that the more data is available, the less influence prior beliefs have because with each iterative update, the prior distribution will be more influenced by the data distribution and thus will approach the true parameter distribution. That is why it is possible to use completely uninformative priors, e.g., a uniform distribution representing a flat line. The use of informative prior however allows for faster statistical inference with less available data given that the informative prior is closer to the true distribution then an uninformative prior. Furthermore, an informative prior can act as form of regularisation when data is noisy or contains many outliers that could cause instabilities in the statistical model. For the models of this study, we used weakly informative priors for all parameters in the sense that we assumed exponential distributions for the parameters related to the variances because these must always be positive. For all other parameters we use Normal distributions as priors for two reasons: Firstly, to ensure that the outcome variable $C_{MW}$ is normally distributed, it makes mathematically sense to also assume that, in a linear combination, all elements of that combination are normally distributed. Secondly, the Normal distribution can represent a wide range of shapes from perfectly symmetric to slightly skewed which makes it a suitable choice if no other strong information is available about the shape of the prior distribution. The prior predictive plots simulation results for our models $m_1$ to $m_4$ are presented as supplementary material in Appendix C.

### 4.6 Step 6: Verify the models on the simulated data.

In this step we show that the models can estimate the parameters set for the simulation and identify independence relations in simulated data. Each model was verified on the simulated data sets created in Step 4. It was tested whether the effect sizes match the values set when generating the simulated outcome data. The results of the parameter estimates for all models under simulated data are given in Appendix B.

### 4.7 Step 7: Run the models on observational data.

Inferring data from the models means predicting the distributions of the parameters as well as predicting the posterior distributions of the outcome (i.e., $C_{MW}$ in all models) given the observed data for $T_S$, *Sol*, $C_{MW}$, and in case of model $m_4$ the external influences $W$ and $r_W$. We applied a stochastic process known as Markov Chain Monte Carlo (MCMC) to find the posterior distribution. MCMC is considered a standard approach for Bayesian Data Analysis and an introduction to the technique is available in Brooks et al. (2011). In brief, MCMC works by running a Markov chain that samples data from posterior





distribution. In a Markov chain, the next state only depends on the current state and not on any other states before that. Different techniques, such as Hamiltonian Monte Carlo (HMC), can be applied to find the next state in the Markov chain. In HMC, the next state is not determined purely at random, but instead through the gradient of the posterior distribution which allows an adaptive step size for each new state. To find the posterior distributions for the parameter and outcome using MCMC with HMC, we used the R-packages rethinking (Richard McElreath, 2023) and Rstan (Stan Development Team, 2024).

### 4.8 Step 8: Checking of independence criteria and parameter estimates.

Initially, scatter plots were created to check for statistical independence between the variables. These are presented and discussed as supplementary material in the Appendix E. Furthermore, the parameter estimates can be used to identify statistical independence between variables, which is discussed as part of the results in Section 5.2. The results of the independence check were compared with the established independence criteria to provide evidence for a final candidate model.

### 4.9 Step 9: Validating the plausibility, workability, and adequacy of the models.

Furia et al. (2022) recommend validating three characteristics of statistical models for BDA: The models must be *plausible*, *workable*, and *adequate*.

**Plausibility**

*Plausibility* can be achieved by checking that the model is "consistent with (expert) knowledge about the data domain" (Furia et al., 2022). The key aspect here is that the models should include reasonable priors that they are consistent with expert knowledge and that are neither too permissive nor too constraining, especially in the case of limited available data (McElreath, 2020). The plausibility of the priors can be checked using prior predictive simulations which we provide as supplementary material in Appendix C.

**Workability**

Second, the models must be *workable*, i.e., it must be computationally possible to fit the models to the provided data without, for example, encountering numerical difficulties. Numerical difficulties can arise from multicollinearity (Furia et al., 2022): if two variables in the data are very strongly correlated, the model cannot determine the ratio of contribution to the outcome between the two variables and we cannot determine the effect size for each of the variables. To check the workability of a model, the sampling process can be independently repeated a few times and the similarity between each sampling process can be estimated through the ratio of within-to-between chain variance $\hat{R}$. An $\hat{R}$-value close to 1 indicates a stationary posterior distribution which is desirable. Another metric that provides an indication of the workability of the model is the effective sample size. It indicates the size of samples that are not autocorrelated, i.e., it measures how much information is lost due to information redundancy between samples. A recommendation is that the effective sample size should be at least 10% for each estimated parameter (Furia et al., 2022). Other indicators of numerical problems are divergent transition warnings that





statistical toolboxes such as rethinking or the underlying Stan library may raise when evaluating the posterior distributions of
the models. As part of the model validation for the proposed models we provide the $\hat{R}$-values and effective sample sizes for
each model together with the detailed inference results in Appendix F. We also checked for warnings of divergent transitions
while training the models on the data.

**Adequacy**

A final aspect of model validation is that the models must be *adequate* for the problem under investigation. An adequate model
can generate data that are similar to the observational data (Furia et al., 2022). Adequacy can be tested by comparing the
*posterior predictive plots* with the empirical data. A posterior predictive plot visualises the posterior predictive distribution
which is obtained by evaluating the posterior distributions of the likelihood parameters, in our case $\mu$ and $\sigma$, and then sampling
from the resulting probability distribution. We provide posterior predictive plots for each model as part of the model evalu-
ation in Figure 11. Statistical models can also be compared to each other using information criteria. The *Watanabe-Akaike*
*Information Criterion* (*WAIC*) is an example of an information criterion commonly used for relative comparison of statistical
models (Watanabe and Opper, 2010). It approximates the relative out-of-sample Kullback-Leibler (KL) divergence, i.e., it tells
us something about the relative "statistical distance" of a model from the data. Because information criteria rely in some way
on the KL divergence, which is not a metric but a relative measure, *WAIC*, and most other information criteria, are also relative
measures of the adequacy of the models. They do not provide an absolute metric that can be used to determine the absolute
adequacy of an individual model. Instead, information criteria such as *WAIC* can be used to compare models against each other.
We provide *WAIC* scores for all models as part of the model evaluation in Section 5.3 and in Figure 12.

### 4.10  Step 10: Report and interpret the results.

The results, discussions and answers to research questions are presented in Sections 5 and 6.

## 5  Results

### 5.1  Measured data

The data used for this study were collected between April 1st 2020 (13:20) and April 25th 2020 (02:29). Figure 6 and Table 3
summarise and present the observed data. The data have a coverage rate of 92% for all measured parameters with 6149 valid
data points. The month of April was chosen due to the good data coverage, which makes it already visually possible to see
some co-variations between the variables in the data.

The measured $C_{MW}$ in Figure 6 (c) show clear diurnal patterns with peaking concentrations during daytime and lower
concentrations during night-time. This coincides with the measured solar radiation shown in Figure 6 (a). Solar radiation
shows clear diurnal patterns with higher values during the day and lower values during the night, as to be expected. It is even
possible to see in the patterns of the data that the April 11th was a cloudy day, as less radiation was measured at noon. This



**Table 3.** Summary statistics of the measured parameters $C_{MW}$, solar radiation (Sol), surface water temperature ($T_S$), windspeed ($W$), and speed of the water pump ($r_W$) during April 2020.

|  | $DGM$ [pg/L] | $C_{MW}$ [pg/L] | $Sol$ [W/m$^2$] | $T_S$ [K (°C)] | $W$ [m/s] | $r_W$ [l/min] |
|---|---|---|---|---|---|---|
| Average | 14 | 2.4 | 155 | 282 (8.1) | 6.9 | 2.8 |
| Max | 28 | 5.0 | 836 | 293 (20) | 25 | 14 |
| Min | 5 | 0.7 | 0.0 | 276 (2.5) | 0 | 0 |
| Standard deviation | 4 | 0.9 | 206 | 2.8 | 4.4 | 1.7 |
| Number of data points | 5282 | 6149 | 6149 | 6149 | 6149 | 6149 |

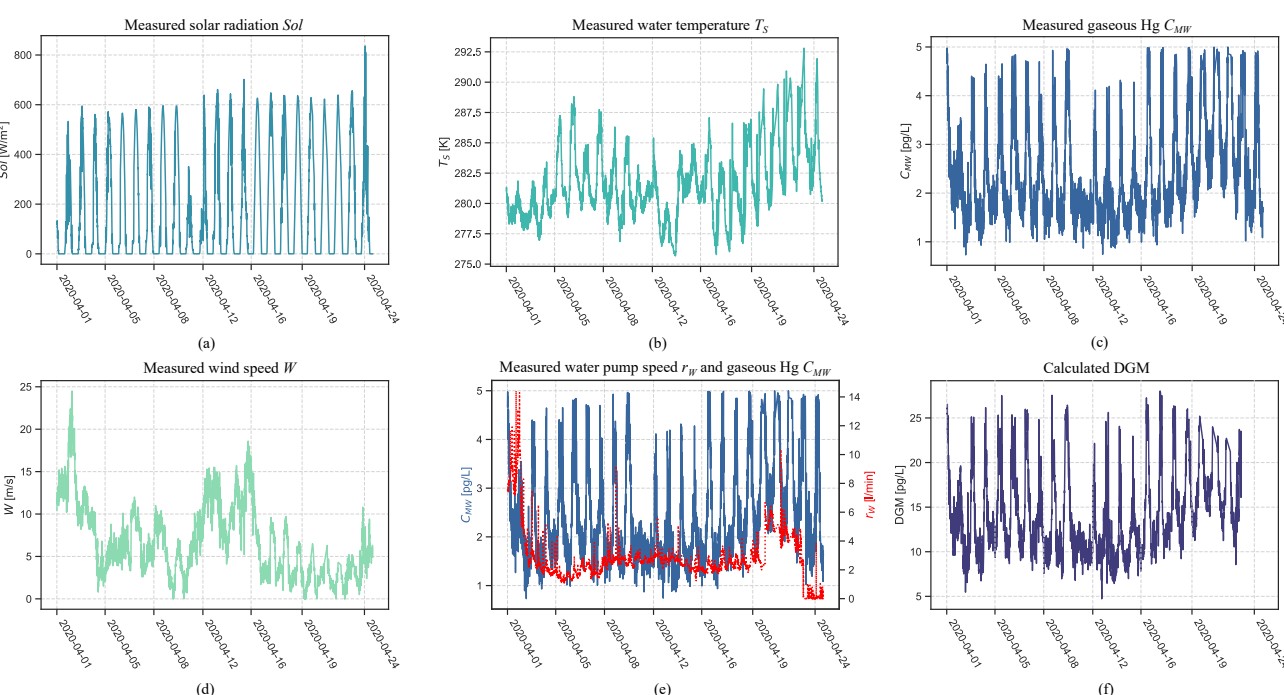

**Figure 6.** Observational data collected in April 2020 used in the analysis.

explains why the peak in measured $C_{MW}$ was lower at this time. The measured surface water temperature, plotted in Figure 6
(b), also shows variations with higher temperatures during the day and lower temperatures during the night, suggesting an association between solar radiation and surface water temperature. In Figure 6 (d) we also plotted $C_{MW}$ to show how the distinct covariation of the pump speed $r_W$ and the measured Hg concentration $C_{MW}$.





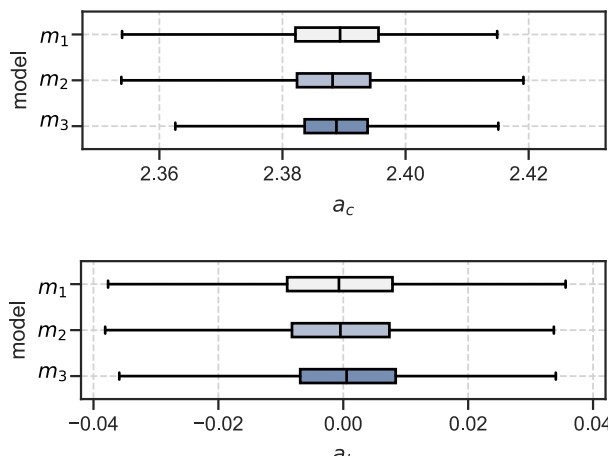

**Figure 7.** Coefficient estimates for $a_c$, the constant term for $C_M W$, and $a_T$, the constant term for $T_S$.

## 5.2 Parameter estimates and causal inference

After fitting the models to the observed data, we obtained posterior distributions for each parameter of the models. Figure 7

shows the estimates for the constant parameters $a_C$ and $a_T$ of the models listed in Table 2. The first parameter $a_C$ is the constant offset in the $Sol \rightarrow C_{MW} \leftarrow T_S$ part of the models. It is approximately equal to the mean of the $C_{MW}$ values in data which is 2.4 pg/L. The estimate for $a_T$ is close to zero for all models because we have standardised the predictor parameters $Sol$ and $T_S$. Figure 8 provides the estimates for the coefficients of effect sizes $b_{t,s}$, $b_{c,s}$, and $b_{c,t}$. These estimates are important to answer what the direct and indirect effect of solar radiation on Hg concentration are (RQ1).

None of the effect size estimates are zero, which means that we cannot assume any statistical independence between the variables. We confirmed these results using manual examination of scatter plots, which is outlined as supplementary material in Appendix E. Hereafter, we propose **model $m_3$ is the most plausible causal model for the variables of interest**.

**Detailed argumentation on why model $m_3$ is most plausible**

The parameter $b_{c,s}$ is the effect size between the standardised predictor variable $Sol$ and $C_M W$. It gives us an estimate of the

strength of the association, and hence the assumed causal relationship, between solar radiation and measured Hg $C_{MW}$. The estimated mean value for $b_{c,s}$ is 0.547 for model $m_1$ and 0.360 for model $m_3$. Model $m_2$ has no estimate for $b_{c,s}$ because it assumed (incorrectly) a-priori that there is no causal relationship between $Sol$ and $C_{MW}$. Because we used standardised predictor variables, these effect size estimates imply that one standard deviation of solar radiation is associated with a change in $C_{MW}$ of $0.547\frac{pg}{L}$, and $0.360\frac{pg}{L}$ respectively for model $m_3$. Dividing the estimated effect size $b_{c,s}$ by the original standard

deviation of the data for Sol gives the de-standardised effect size estimate $b_{c,s}^o$:

$$b_{c,s}^o = \frac{b_{c,s}}{\sigma_{Sol}}. \tag{11}$$





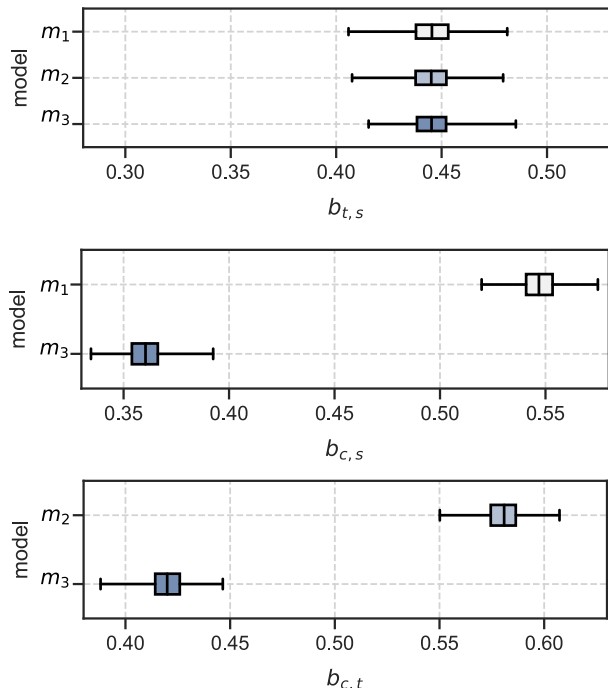

**Figure 8.** Coefficient estimates for the effect size of solar radiation on temperature ($b_{t,s}$), the effect size of solar radiation on Hg concentration ($b_{c,s}$), and the effect size of sea surface temperature on Hg concentration ($b_{c,t}$).

Using the standard deviation of Sol given in Table 3, model $m_1$ estimates that an increase of solar radiation by $1 \frac{W}{m^2}$ is associated with an average increase in $C_{MW}$ of $\frac{0.547}{206.29} \cdot \frac{pg}{L} (\frac{W}{m^2})^{-1} = 0.00265 \frac{pg}{L} (\frac{W}{m^2})^{-1}$. Model $m_3$ estimates an increase in $C_{MW}$ of $\frac{0.360}{206.29} \cdot \frac{pg}{L} (\frac{W}{m^2})^{-1} = 0.00175 \frac{pg}{L} (\frac{W}{m^2})^{-1}$, which is 66% of the estimate from $m_1$. Assuming that there is a causal relationship between solar radiation and $C_{MW}$, which model gives the correct estimate of the strength of the causal relationship between Sol and $C_{MW}$?

**Total effect and direct effect**

In fact, both estimates of the strength of the causal relationship are correct. However, model $m_1$ provides with $b_{c,s}$ the total effect. Model $m_3$ provides instead the direct effect of *Sol* on $C_{MW}$. If there is a causal relationship between *Sol* on $C_{MW}$, there is also a "flow" of association between these variables. This "flow" of association can take different "routes" between the variables. In our case, solar radiation *Sol* has a direct effect on the concentrations of Hg ($C_{MW}$), which means that we can measure an association between these two variables. Its effect size is estimated in our models as $b_{c,s}$:

$$\text{Direct Effect}_{Sol \rightarrow C_{MW}} = b_{c,s}. \tag{12}$$

However, *Sol* also affects the sea surface temperature $T_S$, which in turn affects $C_{MW}$. Besides a direct effect, *Sol* also has an indirect effect on $C_{MW}$, moderated by $T_S$. The indirect effect of *Sol* on $C_{MW}$ can be calculated by multiplying the effect sizes



for the paths $Sol \rightarrow T_S$ and $T_S \rightarrow C_{MW}$:

$$\text{Indirect Effect}_{Sol \rightarrow C_{MW}} = b_{t,s} \cdot b_{c,t}. \tag{13}$$

In summary, a part of the association between $Sol$ and $C_{MW}$ "flows" via $T_S$. The total effect of $Sol$ on $C_{MW}$ is the sum of direct effect and indirect effect.

$$\text{Total Effect}_{Sol \rightarrow C_{MW}} = b_{c,s} + b_{t,s} \cdot b_{c,t}. \tag{14}$$

For model $m_3$, the direct effect equals, on average, to $b_{c,s} = 0.360$ and the indirect effect equals, on average, to $b_{t,s} \cdot b_{c,t} = 0.445 \cdot 0.420 = 0.187$, as illustrated in Figure 9 (b). Using Equation (14), the total effect of $Sol$ on $C_{MW}$ is then $0.360 + 0.187 = 0.547$, which is exactly the estimate for $b_{c,s}$ that model $m_1$ provided. By removing the path between $T_S$ and $C_{MW}$, we do not allow a "flow" of association between $T_S$ and $C_{MW}$ in model $m_1$, as illustrated in Figure 9 (a). The association related

to the indirect effect of $Sol$ on $C_{MW}$ that is mediated by $T_S$, therefore cannot flow through $T_S$ and is instead "rerouted" onto the direct path between $Sol$ and $C_{MW}$ (red, wide dotted line in Figure 9 (a)). Consequently, the effect size $b_{c,s}$, estimated by model $m_1$, is the sum of the direct and indirect effects, thus in fact the total effect of $Sol$ on $C_{MW}$.

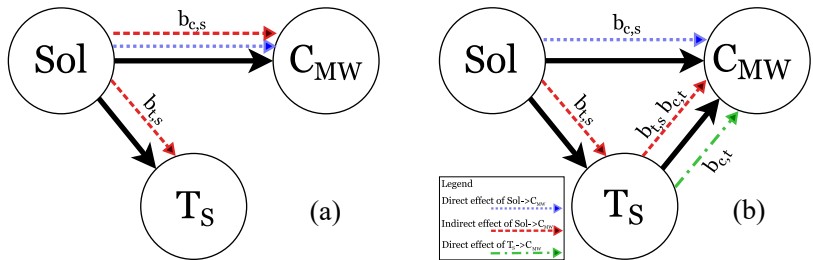

**Figure 9.** DAGs with illustration of flow of association. Red, wide dotted line: association of solar radiation $Sol$ on Hg concentration $C_{MW}$, mediated by sea surface temperature $T_S$. Blue, dense dotted line: association of $Sol$ on $C_{MW}$, not influenced by $Sol$. Green, mixed dotted line: association of $T_S$ on $C_{MW}$. (a) Model $m_1$ where the path $T_S \rightarrow C_{MW}$ is not allowed and the indirect effect of $Sol$ on $C_{MW}$ is therefore "rerouted" on top of the direct effect of $Sol$ on $C_{MW}$. (b) Model $m_3$ where the path $T_S \rightarrow C_{MW}$ is allowed and the indirect effect of $Sol$ on $C_{MW}$ therefore can flow via $T_S$.

A similar line of argumentation can be provided for the difference of the estimate of $b_{c,t}$ between model $m_2$ and model $m_3$. Because the path $Sol \rightarrow C_{MW}$ is not allowed in model $m_2$, the association related to the direct effect of $Sol$ on $C_{MW}$ is

"rerouted" on the path $Sol \rightarrow T_S \rightarrow C_{MW}$. This results in an overestimate of the direct effect $b_{c,t}$ of $T_S$ on $C_{MW}$. Due to the misspecification of models $m_1$ and $m_2$, these two models result in a higher standard deviation $\sigma$ for the $C_{MW}$ data as shown in Figure 10 because the models cannot correctly map all association between $Sol$, $T_S$, and $C_{MW}$. However, the standard deviation $\tau$ for the relation $Sol \rightarrow T_S$ is identical for all models, because the path between $Sol$ and $T_S$ exists in all models. Table 4 summarises the estimated effect sizes for model $m_3$.





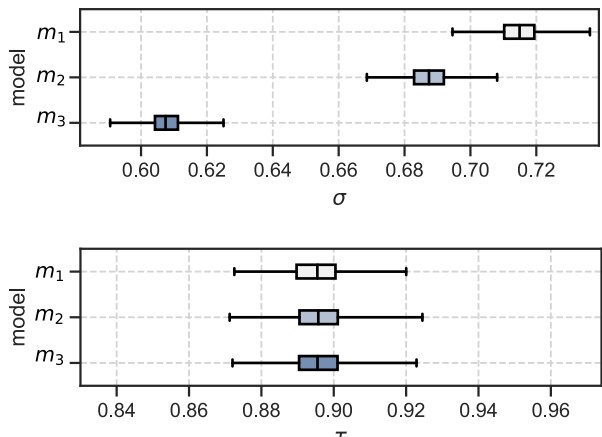

**Figure 10.** Estimates of standard deviation $\sigma$ for $C_{MW}$ and standard deviation $\tau$ for $T_S$.

### Adding estimates for the effect of external influencing factors in model $m_4$

After having identified the most plausible "core" model, we extended it by adding estimates of external influencing factors. Wind, denoted as $W$, is an *environmental factor* that could affect the measurements of $C_{MW}$. The speed of the water pump, $r_W$, however, is a *instrument-intrinsic* factor that could have effect on the measurements for $C_{MW}$. In line with other predictor variables, we chose identical priors for $W$ and $r_W$ based on a prior predictive check. The results of the estimated effected sizes on $C_{MW}$ of the updated model $m_4$ are presented in Table 4, while all parameter estimates are listed in Appendix F. The average effect size of wind $W$ on surface temperature $T_S$, denoted as $b_{t,w}$ is $-0.24$. This agrees with our expectations that an increase in wind speed results in a lowering of the sea surface temperature. The estimated average effect size of $W$ on Hg concentration $C_{MW}$, denoted as $b_{c,w}$ is $-0.13$. The estimated average effect size of wind is smaller compared to[5] the effect sizes of surface temperature $b_{c,t} = 0.43$ and solar radiation $b_{c,s} = 0.38$. As we expected after inspecting Figure 6 (e), the water pump speed $r_W$ has a distinct effect on the measured Hg concentration $C_{MW}$, with an estimated average effect size of $b_{c,r} = 0.27$. An interesting observation can be made when estimating the effect sizes of the other environmental variables, *Sol*, $T_S$, and $W$, on the water pump speed. The effect sizes of solar radiation and sea surface temperature on the water pump speed are small with average effect sizes of $b_{r,s} - 0.04$ and $b_{r,t} - 0.09$ respectively. But the influence of wind speed on the water pump speed is strong with an average effect size of $b_{r,w} = 0.27$. All inferred effect sizes are listed in Table F1 in Appendix F. A possible explanation for the strong relationship between wind speed and water pump speed is that due to high winds, sea waves could have cleared the inlet of the pump from algae, resulting in a higher flow speed. In summary, the inclusion of the confounding external factors wind $W$ and water pump speed $r_W$ lead to a noticeable increase in the estimated effect size of solar radiation *Sol* on measured Hg concentration $C_{MW}$ by 6.3% compared to the estimate of model $m_3$.

---

[5]The effect sizes of the different predictors are comparable because all predictor variables were standardised.




**Table 4.** Estimates of the direct, indirect, and total effects based on observed data in April 2020 without (model $m_3$) and with (model $m_4$) recognition of wind and pump speed as confounding external influences. Bold values are mean values; standard deviations are in parentheses; 90% confidence intervals are depicted in square brackets.

| Effect on measured gaseous Hg $C_{MW}$ | Parameters | Standardised value (std. dev.) [90% confidence interval] | | De-standardised value (std. dev.) [90% confidence interval] | | Unit for de-standardised values | Change due to external influences |
|---|---|---|---|---|---|---|---|
| | | Model $m_3$ | Model $m_4$ | Model $m_3$ | Model $m_4$ | | |
| Direct effect of solar radiation *Sol* | $b_{c,s}$ | **0.360** (0.009) [0.345, 0.374] | **0.383** (0.008) [0.371, 0.395] | **1.75·10$^{-3}$** (4.36·10$^{-5}$) [1.67, 1.81]·10$^{-3}$ | **1.86·10$^{-3}$** (3.67·10$^{-5}$) [1.80, 1.92]·10$^{-3}$ | $\frac{\text{pg}}{\text{L}} \cdot \left(\frac{\text{W}}{\text{m}^2}\right)^{-1}$ | +6.3% |
| Indirect effect of *Sol* mediated by sea surface temperature $T_S$ | $b_{t,s}b_{c,t}+$ $b_{r,s}b_{c,r}$ | **0.189** (0.006) [0.177, 0.197] | **0.186** (0.007) [0.175, 0.198] | **0.92·10$^{-3}$** (2.91·10$^{-5}$) [0.86, 0.96]·10$^{-3}$ | **0.90·10$^{-3}$** (3.66·10$^{-5}$) [0.84, 0.96]·10$^{-3}$ | $\frac{\text{pg}}{\text{L}} \cdot \left(\frac{\text{W}}{\text{m}^2}\right)^{-1}$ | -1.8% |
| Total effect of *Sol* | $b_{c,s}+$ $b_{t,s}b_{c,t}+$ $b_{r,s}b_{c,r}$ | **0.549** (0.011) [0.530, 0.564] | **0.572** (0.011) [0.553, 0.590] | **2.65·10$^{-3}$** (5.33·10$^{-5}$) [2.57, 2.73]·10$^{-3}$ | **2.77·10$^{-3}$** (5.44·10$^{-5}$) [2.68, 2.86]·10$^{-3}$ | $\frac{\text{pg}}{\text{L}} \cdot \left(\frac{\text{W}}{\text{m}^2}\right)^{-1}$ | +4.5% |
| Direct effect of $T_S$ | $b_{c,t}$ | **0.420** (0.009) [0.405, 0.435] | **0.429** (0.008) [0.417, 0.441] | **1.53·10$^{-1}$** (3.28·10$^{-3}$) [1.48, 1.58]·10$^{-1}$ | **1.56·10$^{-1}$** (2.91·10$^{-3}$) [1.51, 1.61]·10$^{-1}$ | $\frac{\text{pg}}{\text{L}} \cdot \text{K}^{-1}$ | +1.96% |
| Direct effect of wind speed $W$ | $b_{c,w}$ | – | **-0.125** (0.007) [-0.137, -0.114] | – | **-2.87·10$^{-2}$** (1.61·10$^{-3}$) [-3.14, -2.60]·10$^{-2}$ | $\frac{\text{pg}}{\text{L}} \cdot \left(\frac{\text{m}}{\text{s}}\right)^{-1}$ | – |
| Direct effect of pump speed $r_W$ | $b_{c,r}$ | – | **0.265** (0.007) [0.253, 0.277] | – | **2.21·10$^{-3}$** (6.67·10$^{-5}$) [2.11, 2.31]·10$^{-3}$ | $\frac{\text{pg}}{\text{L}} \cdot \left(\frac{\text{L}}{\text{min}}\right)^{-1}$ | – |

## 5.3 Results from model validation on observed data

We have already established that model $m_3$ and its extension with external influences to model $m_4$ are most *plausible*. Furthermore, all models are *workable* meaning that it is computationally possible to fit the models to the data provided. Evidence for this claim is that the $R^2$-values for all parameters in all models are close to 1 indicating stable posterior distributions as listed in Table F1 in Appendix. Further evidence of the workability of the models are the effective sample sizes $n_{\text{eff}}$ which indicate that the samples contain sufficient information to infer the model parameters. The overall sample size was $n = 6149$. The smallest

effective sample size occurred for parameter $b_{r,t}$ of model $m_4$ with $n_{\text{eff}} = 811$ which is more than the recommended ratio of at least 10% of the total sample size. We did not receive any numerical warnings during the training of the models. Finally, we also checked that the models are *adequate* by comparing the posterior predictive plots with the empirical data as shown in Figure 11 a) – d). The closer the points in these plots are to the black diagonal line, the better a model can predict the outcome variable. It is obvious that model $m_3$ and $m_4$ seem most adequate to predict $C_{MW}$. This is further confirmed by plotting the

posterior distribution densities, as shown in Figure 11 e). The posterior density of $C_{MW}$ in this plot is closest to the observed data density for model $m_4$. We have also calculated the $R^2$-values, shown in Figure 11 f), using the mean of the posterior predictions for $C_{MW}$ for all models. Model $m_3$ has an $R^2$-value of 0.544 and model $m_4$ has the highest $R^2$-value of all models




with a value of 0.626, which indicates a moderate to good fit to the data. It is important to remember that model $m_3$ only accounts for solar radiation, temperature changes and the interaction between these two external variables, and model $m_4$ adds
the additional influences of wind and variation of the water speed pump. The models ignore any additional external factors that might influence $C_{MW}$. It is therefore a rather simple model of $C_{MW}$, but yet the fit to the data is quite acceptable.

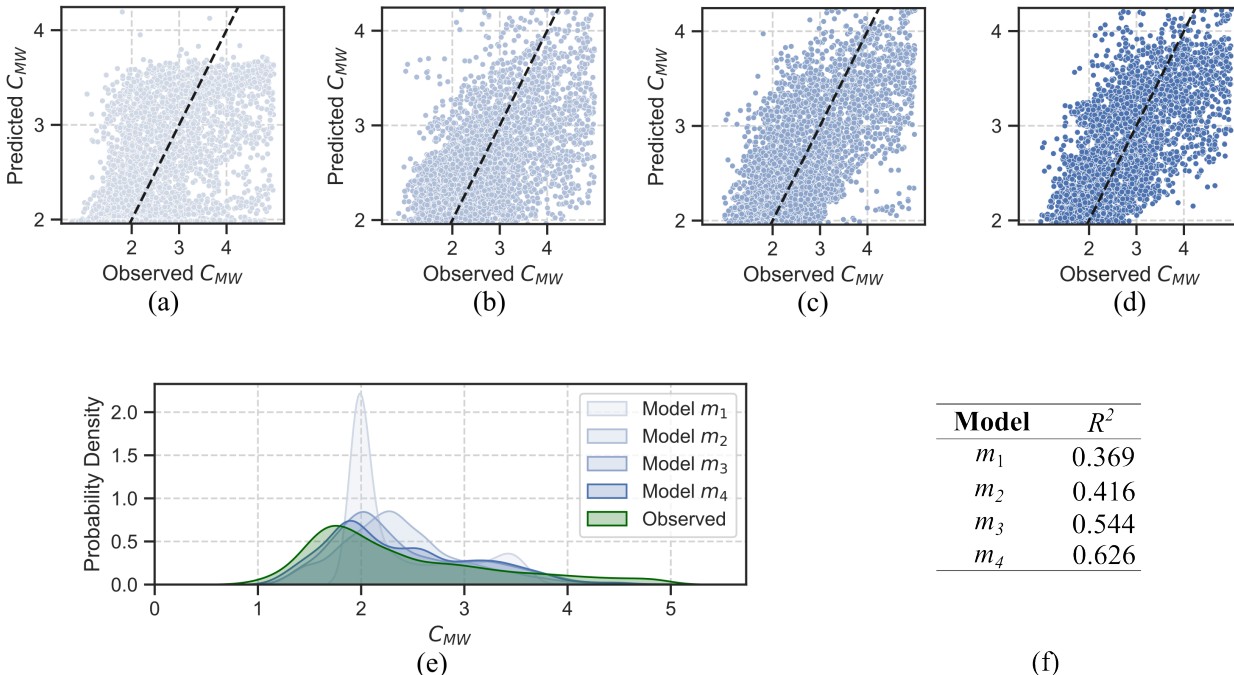

**Figure 11.** Posterior predictive plot for models (a) $m_1$, (b) $m_2$, (c) $m_3$, and (d) $m_4$. The dotted lines indicate where the posterior prediction and the sample match exactly. (e) provides the posterior distribution density for all models. (f) provides the coefficients of determination (R-squared values) for all models.

Finally, we calculated the *WAIC* information criterion which allows us to make a relative comparison of the statistical models. Figure 12 suggests that model $m_4$ has the lowest deviance and is therefore the most adequate model. A lower deviance calculated by *WAIC* indicates a smaller statistical distance between the estimated posterior distribution of the model and the
probability distribution of the observed data which we have also checked manually using the posterior distribution density plots in Figure 11 (e).





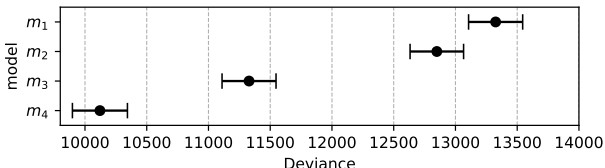

**Figure 12.** Comparison of models using WAIC.

## 6 Discussion

When using statistical or ML methods to analyse observational data, researchers need to distinguish between predictive and analytical models. In a predictive model, the aim is to use the data to train a model that predicts future trends in an outcome of interest. Any observed variable that adds predictive power reduces the uncertainty of the outcome, and these variables are typically strongly associated with the outcome of interest. ML algorithms therefore require large amounts of data to find as many associations in the data as possible. However, purely predictive models are black boxes in terms of which features, or variables, in the data contribute to the outcome and to what extent. A warning example of how adding any variable that adds predictive power can lead to erroneous conclusions on effect sizes is given in Appendix D.

Besides predicting future trends, environmental research also seeks understanding of why an outcome occurs and to quantify the effects that determine them. Therefore, researchers need to move away from purely predictive models, which work only with associations in the data. Instead, they should refocus on analytical models, which provide estimates of the strength of different cause–effect relationships and help answer why-questions. Unfortunately, in most cases, it is not possible to estimate cause-effect relationships from observational data alone. In other fields, such as medicine or social science, researchers can conduct randomised controlled trials (RCTs) to determine the strength of causal relationships. Because RCTs are typically infeasible in environmental research due to the impossibility of isolating a "control" Earth system, alternative strategies must be found.

Here, we demonstrated how prior domain knowledge can be encoded as graphical causal models, providing:

1. Constraints in terms of the directions of cause and effect based on prior knowledge of researchers;

2. A basis for estimating effect sizes from observational data;

3. Independence conditions that enable model validation and provide transparency of assumptions.

> **Prior knowledge of researchers as foundation for causal inference**
>
> By encoding prior knowledge about cause-effect relationships as causal models, associations in observational data can be mapped to causal relationships. This allows to distinguish between different effect sizes of forcing influencing an outcome.





In the following, we revisit the research questions and summarise and discuss the answers we found and their implications for the mercury research community. Then, based on the findings and learnings from the case study, we discuss the use of

causal inference in combination with prior knowledge in terms of its applicability to environmental research in general.

**Answers to research questions**

By applying the causal inference framework suggested in this paper, we found that solar radiation influences measured Hg concentrations in seawater both directly, and indirectly through its effect on sea surface temperature. We verified that the final model $m_4$, although being a simple model of the cause-effect relationships for Hg concentration in seawater, can reconstruct

the probability density of the observed data adequately well with an $R^2$-value of 0.626. After adjusting for additional external influences (RQ2), we identified the following effect sizes:

> **Answer to RQ1 - What is the direct and indirect effect of solar radiation on measured Hg concentration in seawater and subsequently calculated DGM?**
>
> The total effect of solar radiation on $C_{MW}$ and subsequently DGM at the measurement point at Kristineberg, Sweden in April 2020, was $2.77 \cdot 10^{-3}$ $(\mathrm{W/m^2})^{-1}$ pg/L. Out of this total effect, $0.90 \cdot 10^{-3}$ $(\mathrm{W/m^2})^{-1}$ pg/L was an indirect effect of solar radiation mediated by a changing sea surface temperature. The direct effect of solar radiation was $1.86 \cdot 10^{-3}$ $(\mathrm{W/m^2})^{-1}$ pg/L. This means, that about 32% of the effect of solar radiation can be explained by a change of sea surface temperature, which in turn affects the measured $C_{MW}$ and subsequently DGM.

We found that wind has a confounding influence on the effect estimates of solar radiation on $C_{MW}$, and that variations in pump speed significantly affect the measured gaseous Hg concentration, which is explained by Equation 1. Accordingly, the

additional causal modelling of wind as a confounder and pump speed as an instrument-intrinsic factor in model $m_4$ allowed us to adjust the total, indirect, and direct effects of solar radiation on measured mercury concentration in seawater:

> **Answer to RQ2- What are the effect sizes of additional confounding factors affecting the measured Hg concentration in seawater and subsequently calculated DGM?**
>
> Wind speed and water pump speed were identified as confounders and were included in the final statistical model $m_4$. Specifically, wind speed had a direct effect on the measured Hg concentration of $-2.87 \cdot 10^{-2}$ $(\mathrm{m/s})^{-1}$ pg/L. The speed of the water pump, which is an instrument-intrinsic factor influencing the outcome, had a direct effect of $2.21 \cdot 10^{-3}$ $(\mathrm{L/min})^{-1}$ pg/L. By accounting for these additional influences, the estimated direct effect of solar radiation increased by 6.3%, while the indirect effect mediated by sea surface temperature decreased by 1.8%, relative to model $m_3$, in which these factors were not included.





## 6.1 Implications for future mercury research and policies

As mentioned in Section 1.1, several studies report an observed significant correlation between the two variables DGM and
solar radiation. However, considering the regression only between these two factors results in a very simple model, comparable
to our model $m_1$ (Figure 4 (a)). This model is not comprehensive enough to allow for drawing correct causal conclusions. In
contrast, model $m_4$ (Figure 5), which includes mediating effects ($T_S$) and confounding factors ($W$ and $r_W$), produces a noisier
but also more reliable representation of the underlying causal relationships. It is noisier because more features are considered in
the regression model. The model is more reliable because it reduces bias from additional competing effects, confounders, and
background conditions that, if ignored, would give a misleading picture of the underlying causal relationship. For example,
the regression in Figure 13 shows that a regression based on model $m_1$ (Figure 13 (a)) predicts a direct effect in mercury
concentration of about 0.56 pg/L for an increase in solar radiation from 600 W/m$^2$ to 800 W/m$^2$. A regression based on model
$m_4$ (Figure 13 (b)), which includes causal knowledge, predicts only about 0.44 pg/L. Thus, the estimated effect size of solar
radiation on mercury emission is about 21% lower if causal relationships are accounted for. In summary, if causal relationships
are ignored, there is a risk of overestimating the effect of solar radiation on DGM.

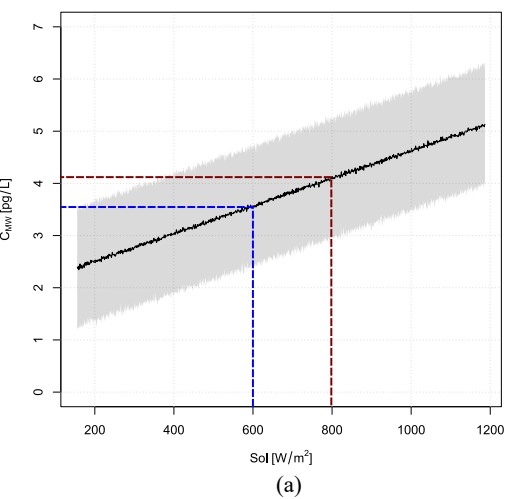

(a)

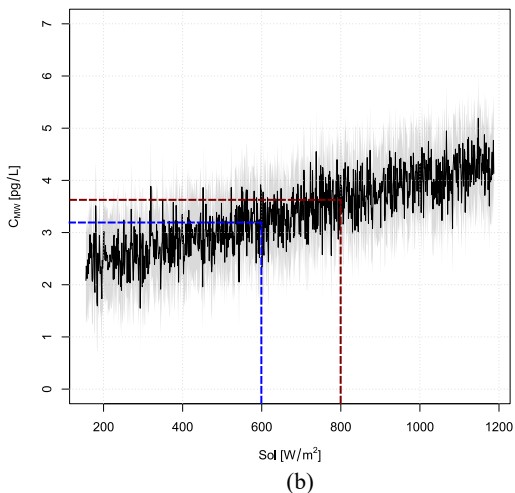

(b)

**Figure 13.** Comparison of regressions of the effect of solar radiation *Sol* on mercury concentration $C_{MW}$ using model $m_1$ (a) and model
$m_4$ (b). Grey shading indicates the 95% confidence interval.

However, to succeed with the challenging task of evaluating for example the effectiveness of the Minamata Convention on
Mercury (United Nations Environment Programme, 2024), better and more reliable computer models are needed to support
sparse mercury measurements. Re-emissions of volatile mercury from sea surfaces is one of the largest contributors to the
mercury load in the atmosphere, which is highly depending on the concentration of DGM. To build better computer models
we need reliable predictors. One of our contributions towards finding more reliable predictors is showing the importance
of modelling prior causal knowledge and including it in statistical models to correctly identify and connect competing and




confounding factors to DGM. The use of graphical causal models strengthens reliability in the results because these models make the underlying causal assumptions transparent and guide the choice of which predictors to include in the regression analysis.

## 6.2 General implications for causal inference in environmental science

Previous attempts to apply causal inference in environmental research have often avoided using prior knowledge. Instead, they tried to discover causal relationships in time series using methods such as Granger causality tests (Runge et al., 2023), potential outcome framework (Reich et al., 2021), or by mimicking randomised controlled trails (Pearce and Lawlor, 2016). However, these approaches rely on strong and often non-transparent assumptions (Maziarz, 2015). Graphical causal models, on the other hand, allow to encode prior assumptions transparently such that the necessary restricting conditions for causal inference from observational data are provided. The importance of a-priori defined graphical causal models has been recognised in other scientific disciplines, such as medicine (Glass et al., 2013), economy (Imbens, 2020), social science (Imbens, 2024), and software engineering (Furia et al., 2019). Scientists in these fields proposed a set of different workflows to work with graphical causal models. Here, we proposed a framework for causal inference using observational data in environmental research with three general elements:

1. First, we suggest incorporating prior knowledge about (assumed) cause-effect relationships in form of a DAG. If a researcher is uncertain about some of the cause-effect relationships, we proposed to define different possible causal models for a given scientific question. Based on mathematical operations that can be performed on the graphical models, known as d-separation (Pearl et al., 2016), independence criteria can be derived for each model. These independence criteria, applied on the observational data, can be used to provide evidence for the selection of one or several candidate causal models.

2. In a second element of the framework, further evidence is collected that show that the proposed statistical or ML models based on the selected causal models are plausible, workable, and adequate.

3. In a final element, the reporting of the results, we argue that the results of the analysis must not only include reporting on the inference outcome, i.e., the parameter estimates. The collection of evidence arguing for the models' plausibility, workability, and adequacy and the incorporated prior knowledge in form of graphical causal models should become a natural part of the result report as well. Only if the causal models that underpin the statistical analysis are available, peers can scrutinise the underlying prior assumptions and argue for or against the findings.

## 7 Conclusions

This study proposed and demonstrated a framework for estimating the effect sizes of multiple forcings on environmental outcomes using observational data. Our case study quantified the direct and indirect effects of solar radiation on measured gaseous Hg, and subsequently dissolved gaseous mercury (DGM), in seawater. We showed that 34% of the total effect was



mediated by sea surface temperature, and that wind and water pump speed acted as confounders. The results from the case study support a framework for inferring effect sizes of cause-effect relationships using observational data which is the key

contribution of this study. The framework includes concise steps for integrating expert knowledge into causal models, validating these models against data, and reporting both statistical inference results and the assumptions underpinning them.

The results are relevant for environmental research in several ways: Today's statistical and even advanced ML models cannot identify and quantify causal relationships in observational data alone. Because we usually cannot conduct randomised controlled trials in environmental research to quantify causal relationships, we must use other methods to identify the strength

of causal relationships. By using causal modelling as part of the analysis of observational data, researchers can explicitly include their prior domain knowledge and assumptions in the otherwise purely automatic statistical analysis of the data. This prior knowledge can be used to impose constraints on statistical models that allow researchers to quantify different effect sizes from observational data. Being able to distinguish between the effect sizes of different forcings on an observed outcome is important because it also allows to distinguish between direct and indirect effects, and thus the effect of future interventions

on the environment. For example, because the final causal models and data suggest that sea surface temperature positively moderates DGM, we know that an increase in sea surface temperature, for example due to climate change, will accelerate the release of mercury and we can quantify this effect using explicit assumptions about causal relationships. In addition, providing prior knowledge in the form of graphical causal models as part of the results of a statistical analysis strengthens the validity of those results because the assumptions about cause-effect relationships made by the researchers are explicitly recorded and can

be scrutinised together with the results of the statistical or ML models.

## 7.1 Limitations and further research

There are limitations to using causal models as a guide for inferring causal knowledge from observational data in environmental research. For example, we have shown that the results of causal inference can be highly sensitive to confounding factors, such as the presence of unmeasured external factors. Future research should explore how to inform future data collection by identifying

sensitivity to unmeasured confounders using causal models. Furthermore, missing and noisy data should be explicitly modelled, as their causes may be entangled with the variables under study.

Another limitation of our approach is that independence criteria alone may not be sufficient to select a correct causal model for analysis. There are situations where several causal models may have identical independence criteria, and we cannot distinguish between them. Future research can explore how the proposed framework can be extended with causal discovery

approaches to suggest candidate causal models (Spirtes and Zhang, 2016). We discussed earlier that current approaches to automatic causal discovery from observational data require too strong and non-transparent assumptions to be useful. However, combining automatic causal discovery with prior expert knowledge could be a promising approach.

Finally, the case study is limited by the limited amount of available data. In future research, it would be interesting to validate effect sizes using long-term measurements over several months or even years. Also, we only studied a few identified

confounding factors. An improvement to this study could be to include for example measurements of dissolved organic carbon since this has been suggested to influence DGM concentration (Ferrara et al., 2003; Amyot et al., 1997).





*Code and data availability.*  A replication package containing the software code and data required to reproduce the findings presented in this paper is available under the DOI https://doi.org/10.7910/DVN/4P59FV[6]

## Appendix A:  Causal inference from observational data

When building predictive or analytical models using statistical methods, such as linear regression, or even advanced machine learning (ML) approaches such as deep learning (DL), the models learn associations between variables from observational data to infer properties of an underlying mathematical structure (Peters et al., 2017). Whether we use statistical modelling or advanced methods of ML, the learned underlying mathematical structure only allows us to identify associations between variables in the dataset which can then be used to build models for prediction and classification (Ghahramani, 2015). We learn

the statistical dependence between variables in the observational data and then infer patterns and trends: For example, we learn that if $X$ changes by 1 unit, on average, $Y$ changes by 2 units, on average, and vice versa. However, statistical models cannot provide the direction of cause- and effect; we cannot infer the causal interdependencies of the underlying system (Runge et al., 2019). Is it X that causes Y to change, or is it $Y$ that affects $X$? Reichenbach extended this problem to common causes: Neither $X$ nor $Y$ could affect each other, instead there could be a third variable $Z$ that is a common cause of both $X$

and $Y$ (Reichenbach and Morrison, 1956). Without additional information, the three cases $X \rightarrow Y$, $Y \rightarrow X$, and $Y \leftarrow Z \rightarrow X$ cannot be distinguished using observational data and statistical modelling alone (Schölkopf, 2022). This limitation even applies to deep neural networks, whose opaque mode of operation usually does not allow the extraction of cause-effect relationships from observational data alone (Montavon et al., 2018).

Causal inference refers to the process of learning about causal relationships and the process of reasoning about outcomes
of interventions and answering counterfactual questions. A common research design for inferring causal relationships are randomised controlled trials (RTC). In an ideal RTC, the subject under study is randomly assigned to different treatments (or effects of something), allowing researchers to isolate the effect of the treatment on the outcome of interest and thus identify and quantify average effect sizes (Hernán and Robins, 2020). This is a rigorous approach to evaluating causal relationships and allows, for example, the effectiveness of interventions to be assessed. Unfortunately, randomised experiments are often

not possible due to prohibitive costs, ethical concerns, or other impracticalities of such experiments. Researchers in many fields, including environmental research, must therefore rely on observational data alone and find methods to infer causality. In environmental monitoring, the assignment of sampling times and sites to "unaffected" and "potentially affected" is not under the control of the researcher and therefore cannot be assigned at random (Stewart-Oaten, 1996). The author suggested that causal inference in environmental research only can be successful if there are diagnostic checks to exclude plausible causal

models that do not fit the data, and measures of model uncertainty for those models that are not excluded. With beginning of the 2010s, political discussions are increasingly focused on the causes and effects of climate change and the impact of human activities on the environment. Environmental monitoring data are used to derive policies and conversation rules. One possible

---

[6]The replication package will be published together with the article. For review purposes, the replication package can be accessed under https://dataverse.harvard.edu/previewurl.xhtml?token=78e59872-5610-4c49-a4e2-6c75fbd5b54a.





approach to extract knowledge about contributing factors to pollution is positive matrix factorisation (PMF) which is a factor analysis tool to judge the level of contribution of different factors to an outcome (Kyllönen et al., 2020). Sun et al. (2022) found

that more than 1000 studies have been published using PMF as a tool to study source appointment for different environmental pollutants, whereof several have used it in mercury studies (Feng et al., 2022; Michael et al., 2016). PMF can be useful to explore latent patterns in data. However, it does not provide mechanisms for distinguishing correlation from causation. Causal modelling has been suggested to be a key aspect in evaluating the effects of environmental policies, such as laws, self-regulation of industry, or governmental oversight (Sills and Jones, 2018). The authors suggest that especially graphical

modelling of cause-effect relationships using DAGs are useful to describe the potential effect of environmental policies.

## Appendix B: Simulation of data and validation of causal models

Before applying the observed data to the models, we conducted a simulation study to validate that the proposed analytical models could correctly identify associations between the variables. This process involved generating simulated observations $C_{MW}$ under three generative models $s_{m_1} - s_{m_3}$, each representing different causal assumptions for the three prediction models

$m_1 - m_3$. For the first dataset, $s_{m_1}$, we assumed that only *Sol* but not $T_S$ has an influence on the generated $C_{MW}^{(1)}$ data. For the second dataset $s_{m_2}$, we assume that only $T_S$ but not *Sol* influences $C_{MW}^{(2)}$. The last dataset $s_{m_3}$ represents an influence of both *Sol* and $T_S$ on $C_{MW}^{(3)}$. The datasets are generated using the following generative process:

$$s_{(m_1)} : C_{\text{MW}}^{(1)} \sim \mathcal{N}(a_C + b_{c,s} \cdot Sol, \sigma) \tag{B1}$$

$$s_{(m_2)} : C_{\text{MW}}^{(2)} \sim \mathcal{N}(a_C + b_{c,t} \cdot T_S, \sigma) \tag{B2}$$

$$s_{(m_3)} : C_{\text{MW}}^{(3)} \sim \mathcal{N}(a_C + b_{c,s} \cdot Sol + b_{c,t} \cdot T_S, \sigma) \tag{B3}$$

In these models, the outcome $C_{NW}^{(i)}$ is normally distributed with a mean including a constant term $a_C$. The parameters $b_{c,s}$ and $b_{c,t}$ represent the effect sizes of solar radiation and sea surface temperature, respectively. Standardised values were used for the predictors to avoid numerical issues. A standard deviation $\sigma$ was assumed to introduce uncertainty. Table B1 list the values for the parameters of the simulation models that generated the test data.

**Table B1.** Parameters for the generative models to create the simulation datasets.

| Parameter | Meaning | $s_{(m_1)}$ | $s_{(m_2)}$ | $s_{(m_3)}$ |
|---|---|---|---|---|
| $a_c$ | Constant term for $C_{MW}$ | 2.4 | 2.4 | 2.4 |
| $b_{c,s}$ | Effect size of solar radiation on $C_{MW}$ | 0.3 | 0.0 | 0.3 |
| $b_{c,t}$ | Effect size of sea surface temperature on $C_{MW}$ | 0.0 | 0.6 | 0.6 |
| $\sigma$ | Standard deviation of the assumed Normal distribution | 0.4 | 0.4 | 0.4 |





The interpretation is as follows: for model $s_{(m_1)}$, setting $b_{c,s} = 0.3$ implies that a one standard deviation change in solar radiation leads to a $b_{c,s}$-times change in the simulated outcome. Similarly, model $s_{(m_2)}$ with $b_{c,t} = 0.6$ implies a corresponding change due to sea surface temperature. Model $s_{(m_3)}$ combines both effects. Figure B1 presents the resulting simulated data for $C_{NW}^{(i)}$ under the three different causal assumptions.

**Table B2.** Parameter estimates for all models under simulated datasets $s_{(m_1)}$, $s_{(m_2)}$, and $s_{(m_3)}$.

| Model | Data | Parameter | Mean | SD | 5% | 95% | $n_{\text{eff}}$ | $\hat{R}$ |
|-------|------|-----------|------|-----|-----|-----|------|-----|
| | $s_{(m_1)}$ | $a_c$ | 2.41 | 0.005 | 2.40 | 2.42 | 1700 | 0.999 |
| | $s_{(m_2)}$ | $a_c$ | 2.40 | 0.009 | 2.38 | 2.41 | 1690 | 1.000 |
| | $s_{(m_3)}$ | $a_c$ | 2.40 | 0.009 | 2.39 | 2.41 | 1602 | 0.999 |
| | $s_{(m_1)}$ | $b_{c,s}$ | 0.30 | 0.005 | 0.29 | 0.30 | 2436 | 1.000 |
| $m_1$ | $s_{(m_2)}$ | $b_{c,s}$ | 0.27 | 0.009 | 0.25 | 0.28 | 1844 | 0.999 |
| | $s_{(m_3)}$ | $b_{c,s}$ | 0.57 | 0.009 | 0.55 | 0.58 | 1854 | 1.000 |
| | $s_{(m_1)}$ | $\sigma$ | 0.40 | 0.004 | 0.39 | 0.40 | 1254 | 1.000 |
| | $s_{(m_2)}$ | $\sigma$ | 0.67 | 0.006 | 0.66 | 0.68 | 1239 | 0.999 |
| | $s_{(m_3)}$ | $\sigma$ | 0.67 | 0.006 | 0.66 | 0.68 | 1575 | 0.998 |
| | $s_{(m_1)}$ | $a_c$ | 2.41 | 0.006 | 2.40 | 2.42 | 1953 | 1.000 |
| | $s_{(m_2)}$ | $a_c$ | 2.40 | 0.005 | 2.39 | 2.41 | 1424 | 0.999 |
| | $s_{(m_3)}$ | $a_c$ | 2.40 | 0.007 | 2.39 | 2.41 | 1700 | 0.999 |
| | $s_{(m_1)}$ | $b_{c,t}$ | 0.13 | 0.006 | 0.12 | 0.14 | 2112 | 0.999 |
| $m_2$ | $s_{(m_2)}$ | $b_{c,t}$ | 0.60 | 0.005 | 0.59 | 0.61 | 1936 | 0.999 |
| | $s_{(m_3)}$ | $b_{c,t}$ | 0.73 | 0.006 | 0.72 | 0.74 | 2006 | 0.998 |
| | $s_{(m_1)}$ | $\sigma$ | 0.48 | 0.004 | 0.47 | 0.48 | 1343 | 1.000 |
| | $s_{(m_2)}$ | $\sigma$ | 0.40 | 0.004 | 0.40 | 0.41 | 1140 | 0.999 |
| | $s_{(m_3)}$ | $\sigma$ | 0.48 | 0.004 | 0.48 | 0.49 | 1056 | 1.001 |
| | $s_{(m_1)}$ | $a_c$ | 2.41 | 0.005 | 2.40 | 2.42 | 1185 | 1.002 |
| | $s_{(m_2)}$ | $a_c$ | 2.40 | 0.005 | 2.39 | 2.41 | 1427 | 0.998 |
| | $s_{(m_3)}$ | $a_c$ | 2.40 | 0.005 | 2.39 | 2.41 | 2005 | 0.998 |
| | $s_{(m_1)}$ | $b_{c,s}$ | 0.30 | 0.006 | 0.29 | 0.31 | 1272 | 1.000 |
| | $s_{(m_2)}$ | $b_{c,s}$ | 0.00 | 0.006 | -0.01 | 0.01 | 1092 | 1.000 |
| | $s_{(m_3)}$ | $b_{c,s}$ | 0.30 | 0.006 | 0.29 | 0.31 | 1174 | 1.001 |
| $m_3$ | $s_{(m_1)}$ | $b_{c,t}$ | 0.00 | 0.006 | -0.01 | 0.01 | 1470 | 1.001 |
| | $s_{(m_2)}$ | $b_{c,t}$ | 0.60 | 0.005 | 0.59 | 0.61 | 1315 | 0.998 |
| | $s_{(m_3)}$ | $b_{c,t}$ | 0.60 | 0.006 | 0.59 | 0.61 | 883 | 1.003 |
| | $s_{(m_1)}$ | $\sigma$ | 0.40 | 0.004 | 0.39 | 0.40 | 863 | 1.000 |
| | $s_{(m_2)}$ | $\sigma$ | 0.40 | 0.004 | 0.40 | 0.41 | 1040 | 0.999 |
| | $s_{(m_3)}$ | $\sigma$ | 0.40 | 0.004 | 0.40 | 0.41 | 1293 | 0.999 |

We then performed Bayesian model inference using the three models on each dataset. The inference results are listed in
Table B2 and visualised in Figure B1.





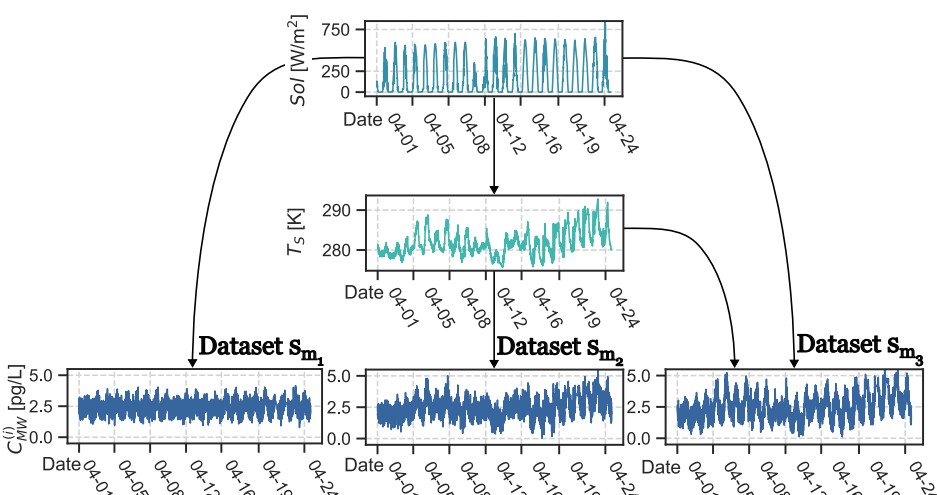

**Figure B1.** Resulting datasets $s_{m_1} - s_{m_3}$ for simulated outcome $C_{NW}^{(i)}$ under three different causal assumptions using observed solar radiation *Sol* and sea surface temperature $T_S$. Arrows indicate assumed direction of cause and effect.

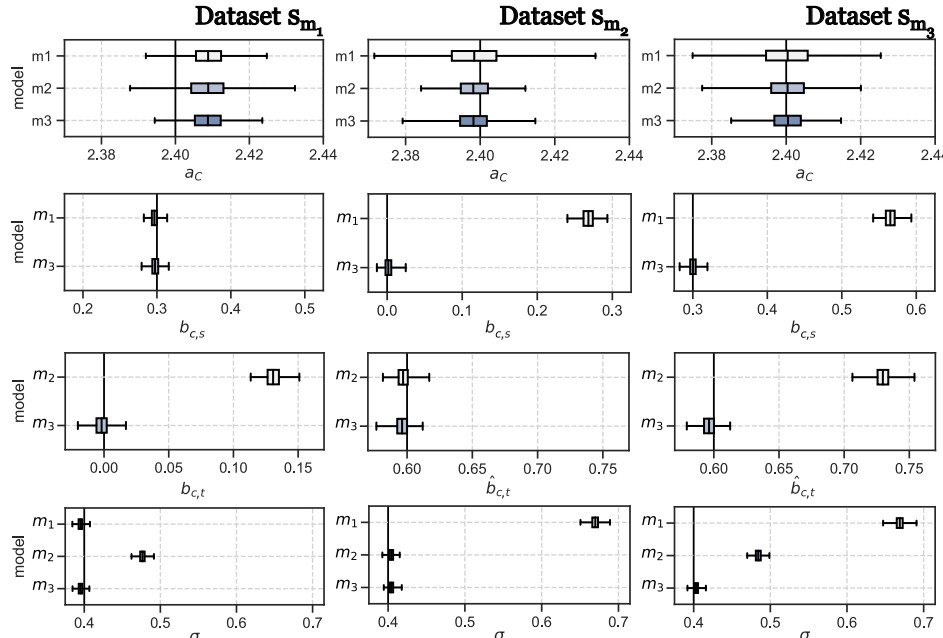

**Figure B2.** Results of model inference under three different simulated datasets $s_{m_1} - s_{m_3}$, each representing different causal assumptions for $C_{MW}$. The left row shows inference under dataset $s_{m_1}$, the middle row under dataset $s_{m_2}$, and the right row under dataset $s_{m_3}$. Solid black lines indicate the true parameter values set for the simulation.



Ideally, models should recover the original parameter values despite the added uncertainty $\sigma = 0.4$. Furthermore, we could examine what happens if we estimate the parameters using an incorrectly specified model for the dataset. For example, model $m_2$ assumes no causal relationship between solar radiation and $C_{MW}$, but the datasets $s_{m_1}$ and $s_{m_3}$ simulate such a causal relationship. Consequently, model $m_2$ incorrectly estimates $b_{c,t}$ for these datasets, mapping effects from solar radiation to sea

surface temperature. In these cases, the estimated $\sigma$ also increases beyond $0.4$ as the model fails to account for all variance in the data. By contrast, model $m_3$ correctly estimates zero effects for irrelevant variables under datasets $s_{m_1}$ and $s_{m_2}$ which indicates the ability to detect statistical independence consistent with expectations listed in Table 1.

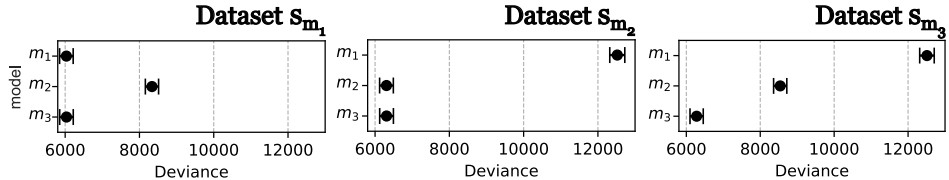

**Figure B3.** Model comparison using WAIC. Lower deviance values indicate that a model is more adequate for the data.

The WAIC-based model comparison, shown in Figure B3 shows:

– For dataset $s_{m_1}$: Models $m_1$ and $m_3$ are adequate.

– For dataset $s_{m_2}$: Models $m_2$ and $m_3$ are adequate.

– For dataset $s_{m_3}$: Only model $m_3$ is adequate.

In conclusion, the results of the simulation study show that the models are *workable*, meaning that, given the chosen priors, they can infer an a-priori determined set of parameters in simulated datasets. We saw that if a model does not by design allow for an association that is present in the dataset, the model will instead incorrectly "map" the missing association onto other

allowed associations, which can inflate unrelated effect sizes. Finally, we saw that model $m_3$ can detect the non-existent of association between variables in the datasets. Paired with domain-derived independence criteria (Table 1), this allows making statements about possible causal relationships in observational data.

**Appendix C: Prior predictive plots simulations**

Prior predictive plots assist in identifying reasonable initial values for the parameters of the prior distributions. This is done by

simulating the model outputs solely using the priors (McElreath, 2020).

Because the predictor values are standardised, we set $a_t = 0$. The outcome variable $C_{MW}$, however, was not standardised, which is why we set the prior mean of $a_c$ to 2.7, corresponding to the mean of the observed $C_{MW}$. These values are reasonable initial choices for $a_t$ and $a_c$ as both parameters represent intercepts of a linear regression. The prior predictive plots were then used to identify initial values for $b_{c,s}, b_{t,s}$, and $b_{c,t}$. Each plot in Figure C1 (a) - (c) shows 100 randomly drawn lines using

models $m_1 - m_4$, with parameter values sampled from the corresponding priors. Observed data are shown as black circles.





**Figure C1.** Prior predictive simulation results. The observed data is shown as grey circles. (a): $a_t \sim \text{Normal}(0,1)$, $b_{t,s} \sim \text{Normal}(0.5, 0.5)$. (b): $a_c \sim \text{Normal}(2.7, 1)$, $b_{c,s} \sim \text{Normal}(0.5, 0.5)$. (c): $a_c \sim \text{Normal}(2.7, 1)$, $b_{c,t} \sim \text{Normal}(0.5, 0.5)$. (d): $a_c \sim \text{Normal}(2.7, 1)$, $b_{c,s} \sim \text{Normal}(20, 20)$, which is an example of an implausible prior.



Our aim was to choose the parameters of the prior distributions such that the simulation results obtained by sampling were plausible given the observed data. Figure C1 (d) illustrates an example of an implausible prior distribution. It was not our aim to match the observed data precisely because priors only serve as a "starting point" for the Bayesian data analysis. The influence of the priors diminishes the more observational data are available as the model iteratively updates its estimates based on the data. Nonetheless, selecting reasonable priors improves the efficiency of inference by reducing the time required to estimate the posterior distributions.

**Appendix D:  A warning about colliders and uninformed machine learning (ML) models**

A common approach might be to include all possible predictor variables in a statistical or ML model. However, a particular situation arises when *colliders* are part of the causal model. A collider is any variable with several causes affecting it. Figure D1 provides an example of a devised example based on the previous model $m_3$. Assume that variable $A$ represents the particle density of algae in the water and that the growth of algae is positively affected by solar radiation. Assume also that mercury has some form of inhibitory effect on algae, which reduces the density of algae in waters with high mercury concentration[7]. Variable $A$ represents a collider in the causal model because it has arrows pointing towards it from both *Sol* and $C_{MW}$. One problem in

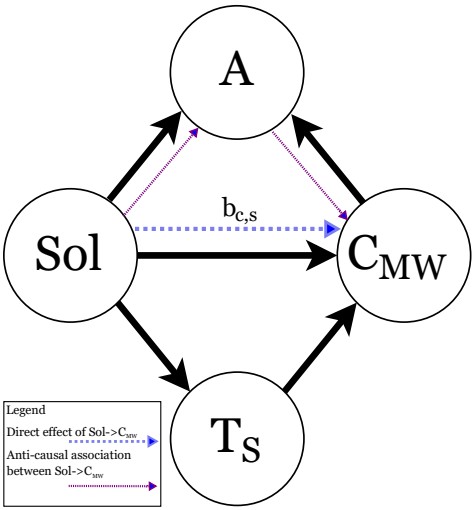

**Figure D1.** Directed Acyclic Graphs (DAGs), which contains a collider $A$.

causally interpreting the results of statistical or ML models that arises in the presence of colliders is *collider bias*. It entails an association between two variables that "flows" against the direction of cause and effect. For example, a researcher might try to build a statistical model that predicts the effect of solar radiation on $C_{MW}$. The researcher conveniently has data on the density of algae particles and assumes that, because algae growth is affected by both solar radiation and Hg concentration, it is a good predictor for $C_{MW}$. We extended the previous simulation model $s_{m_3}$ with a variable $A$ depicting simulated algae growth that

---

[7]This is not based on real measurements but a simulation / thought-experiment




depends on solar radiation and $C_{MW}$ to illustrate the consequence of including a collider as a predictor for estimating the effect

size of solar radiation on $C_{MW}$.

**Table D1.** Added parameters to $s_{m_3}$ for the generation of an extended dataset $s_{m_3}$ that includes a collider $A$.

| Parameter | Meaning | $s_{(m_c)}$ |
|---|---|---|
| $a_a$ | Constant term for algae concentration $A$ | 3.0 |
| $b_{a,s}$ | Effect size of solar radiation on algae concentration $A$ | 0.7 |
| $b_{a,c}$ | Effect size of $C_{MW}$ on algae concentration $A$ | -0.2 |
| $\sigma$ | Standard deviation of the assumed Normal distribution | 0.4 |

Table D1 lists the parameters of the new simulation model $s_{m_c}$. By setting $b_{a,s} = 0.7$, we assume that solar radiation has a strong positive effect on the concentration of algae. Conversely, by setting $b_{a,c} = -0.2$, we assume a weakly negative effect of the presence of mercury on the concentration of algae. All other parameters are identical to those in simulation model $s_{m_3}$ as given in Table B1.

Variable $A$ is included as predictor of $C_{MW}$ by extending model $m_3$ to a new statistical model, $m_c$, specified in Table D2.

**Table D2.** Model specification for collider model $m_c$.

| Model | Specification | |
|---|---|---|
| | $C_{MW_i} \sim \text{Normal}(\mu_i, \sigma)$ | |
| | $T_{S_i} \sim \text{Normal}(\nu_i, \tau)$ | |
| | $A_i \sim \text{Normal}(\xi_i, \vartheta)^{\star}$ | |
| | $\mu_i = a_c + b_{c,s} \cdot Sol_i + b_{c,t} \cdot T_{S_i} + b_{a,c}^{\star} \cdot A_i$ | |
| | $\nu_i = a_t + b_{t,s} \cdot Sol_i$ | |
| $m_c$: | $\xi_i = a_a + b_{a,s}^{\star} \cdot Sol_i$ | (D1) |
| | $a_c \sim \text{Normal}(0, 1)$ | |
| | $a_a \sim \text{Normal}(2, 1)^{\star}$ | |
| | $a_t \sim \text{Normal}(0.5, 1)$ | |
| | $b_{c,s}^{\star}, b_{a,s}^{\star}, b_{c,s}, b_{c,t}, b_{t,s} \sim \text{Normal}(0.5, 0.5)$ | |
| | $\sigma, \tau, \vartheta^{\star} \sim \text{Exponential}(1)$ | |

$^{\star}$Terms that relate to algae growth $A$.

The inference results for model $m_c$ are presented in Table D3 together with the previous results from model $m_3$.

The new model correctly estimates the simulated effect sizes for algae density $b_{a,c}$ and $b_{a_s}$. If not conditioned upon, colliders act as "sinks" for the flow of association (Holmberg and Andersen, 2022). That is, the "flow" of association between *Sol* and $C_{MW}$ via $A$ will be stopped by the collider variable $A$ if we do not control for this variable. However, by including algae as

a predictor for $C_{MW}$, the model inadvertently conditioned on a collider. Thereby, a collider bias is introduced in the estimate of the effect size of solar radiation on $C_{MW}$ because additional association "flows" against the correct direction of cause-and-effect. Therefore, the effect size of solar radiation on $C_{MW}$, $b_{c,s}$, changes from its true value of 0.3 to a higher value of 0.41





**Table D3.** Parameter estimates for models $m_3$ and $m_c$ under simulated data $s_{m_c}$.

| Model | Parameter | Mean | SD | 5% | 95% | $n_{\text{eff}}$ | $\hat{R}$ |
|-------|-----------|------|----|----|-----|------|-----|
| | $a_c$ | 2.39 | 0.005 | 2.39 | 2.40 | 1295 | 1.001 |
| $m_3$ | $b_{c,s}$ | 0.31 | 0.006 | 0.30 | 0.32 | 1319 | 1.001 |
| | $b_{c,t}$ | 0.60 | 0.005 | 0.59 | 0.61 | 1092 | 1.001 |
| | $a_c$ | 2.39 | 0.005 | 2.39 | 2.40 | 1976 | 0.999 |
| | $b_{a,c}$ | -0.17 | 0.009 | -0.19 | -0.15 | 918 | 0.999 |
| $m_c$ | $b_{a,s}$ | 0.69 | 0.005 | 0.68 | 0.70 | 1557 | 1.001 |
| | $b_{c,s}$ | **0.41** | 0.010 | **0.40** | **0.43** | 953 | 0.999 |
| | $b_{c,t}$ | 0.58 | 0.006 | 0.57 | 0.59 | 1389 | 0.998 |

to compensate for the additional "anti-causal" association flowing via $A$. In addition, the uncertainty in the effect size estimate $b_{c,s}$ increases as illustrated in the larger bars in Figure D2 (a).

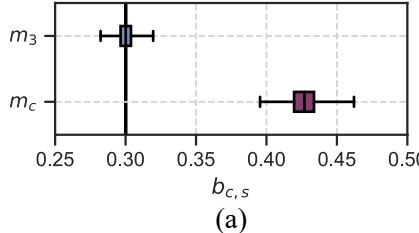
(a)

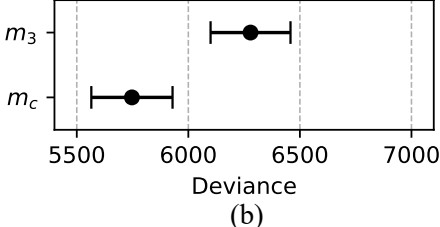
(b)

**Figure D2.** Estimate of effect size $Sol \rightarrow C_{MW}$ (a) and model comparison (b) for model with Algae growth $A$ as predictor ($m_c$) and without $A$ as predictor ($m_3$).

Unfortunately, including additional predictors that are correlated with $C_{MW}$ increases the predictive ability of the model as shown in Figure D2 (b). This means that model $m_c$ can predict $C_{MW}$ better, even if the effect size estimates (the causes to why $C_{MW}$ changes over time) are not correct. The danger is that when using ML model, the algorithm may include all predictor variables that it can find in the data. The result will be a model that predicts the outcome variable very well, but the effect sizes may be wrong. It is therefore important to carefully argue, using causal modelling for example, which variables to include

in a dataset for ML, leading to an *informed* ML. Otherwise, conventional ML approaches, including advanced approaches such as convolutional deep neural networks, might inadvertently condition on collider variables and estimate incorrect effect sizes (Hernán and Robins, 2020).





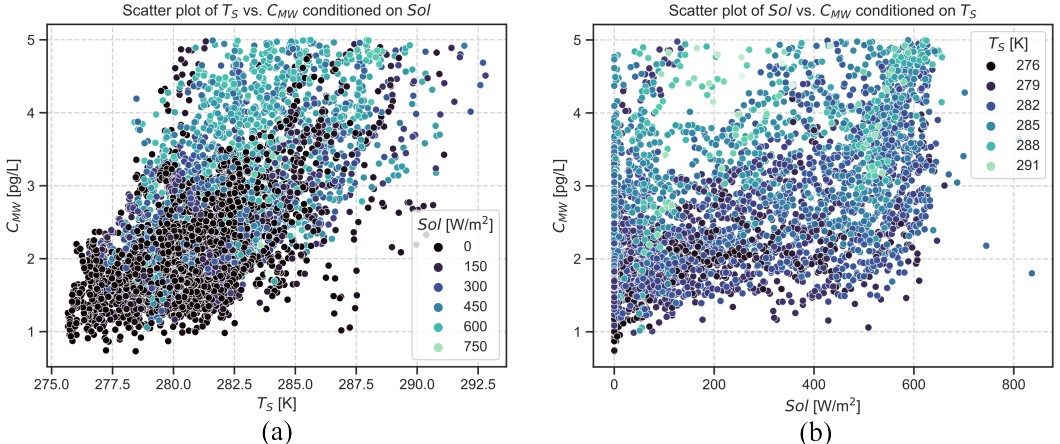

**Figure E1.** Scatter plots for inspecting suggested independence between variables for (a) $m_1$: $T_S \perp C_{MW} \mid Sol$ and (b) $m_2$: $Sol \perp C_{MW} \mid T_S$.

## Appendix E:  Validation through independence checking using scatterplots

### E1   Independence checking

In addition to assessing the estimated effect size coefficients to evaluate independence between variables, we also conducted a manual inspection of scatter plots to visually assess statistical independence between the predictor variables and the outcome. Figure E1 presents the scatter plots corresponding to the independence criteria outlined in Table 1. For example, if $C_{MW}$ were conditionally independent of solar radiation given sea surface temperature, as suggested by model $m_1$, we would expect no apparent pattern or trend between $T_S$ and $C_{MW}$ at different levels of *Sol*. However, Figure E1 (a) indicates a linear relationship
between $T_S$ and $C_{MW}$ at each level of *Sol*. Similarly, Figure E1 (b) indicates a linear relation between *Sol* and $C_{MW}$ which appears to be more distinct with higher values for $T_S$. Overall, the visual inspection of the scatter plots suggests that *statistical independence between the variables is not supported*, implying that the assumed causal models $m_1$ and $m_2$ are not plausible.

## Appendix F:  Parameter estimates under observed data

Table F1 lists the inference results using models $m_1$, $m_2$, and $m_3$ under the observed data.



**Table F1.** Parameter estimates for all models under observed data.

| Model | Parameter | Mean | SD | 5% | 95% | $n_{\text{eff}}$ | $\hat{R}$ |
|---|---|---|---|---|---|---|---|
| | $a_c$ | 2.39 | 0.010 | 2.37 | 2.41 | 1762 | 0.999 |
| | $b_{c,s}$ | 0.55 | 0.009 | 0.53 | 0.56 | 1401 | 1.000 |
| | $\sigma$ | 0.72 | 0.007 | 0.71 | 0.73 | 1362 | 1.000 |
| $m_1$ | $a_t$ | 0.00 | 0.012 | -0.02 | 0.02 | 1453 | 0.999 |
| | $b_{t,s}$ | 0.45 | 0.012 | 0.43 | 0.46 | 1399 | 1.001 |
| | $\tau$ | 0.90 | 0.008 | 0.88 | 0.91 | 1412 | 0.999 |
| | $a_c$ | 2.39 | 0.009 | 2.37 | 2.40 | 1465 | 1.000 |
| | $b_{c,t}$ | 0.58 | 0.009 | 0.57 | 0.60 | 1347 | 0.999 |
| | $\sigma$ | 0.68 | 0.007 | 0.68 | 0.70 | 1493 | 1.000 |
| $m_2$ | $a_t$ | 0.00 | 0.012 | -0.02 | 0.02 | 1271 | 1.000 |
| | $b_{t,s}$ | 0.45 | 0.011 | 0.43 | 0.46 | 1373 | 0.999 |
| | $\tau$ | 0.90 | 0.008 | 0.88 | 0.91 | 1389 | 1.000 |
| | $a_c$ | 2.39 | 0.008 | 2.38 | 2.40 | 1261 | 1.002 |
| | $b_{c,s}$ | 0.36 | 0.009 | 0.35 | 0.37 | 1051 | 1.000 |
| | $b_{c,t}$ | 0.42 | 0.009 | 0.41 | 0.43 | 1223 | 1.001 |
| $m_3$ | $\sigma$ | 0.61 | 0.005 | 0.60 | 0.62 | 1349 | 1.000 |
| | $a_t$ | 0.00 | 0.011 | -0.02 | 0.02 | 1392 | 0.999 |
| | $b_{t,s}$ | 0.45 | 0.011 | 0.43 | 0.46 | 1521 | 1.000 |
| | $\tau$ | 0.90 | 0.008 | 0.88 | 0.91 | 1139 | 0.998 |
| | $a_c$ | 2.39 | 0.007 | 2.38 | 2.40 | 1257 | 1.002 |
| | $b_{c,s}$ | 0.38 | 0.008 | 0.37 | 0.40 | 983 | 0.999 |
| | $b_{c,t}$ | 0.43 | 0.008 | 0.42 | 0.44 | 959 | 1.001 |
| | $b_{c,w}$ | -0.13 | 0.009 | -0.14 | -0.11 | 1068 | 0.999 |
| | $b_{c,r}$ | -0.27 | 0.007 | -0.27 | -0.14 | 1068 | 0.999 |
| | $\sigma$ | 0.55 | 0.005 | 0.54 | 0.55 | 1047 | 1.002 |
| $m_4$ | $a_t$ | 0.00 | 0.011 | -0.02 | 0.02 | 1125 | 0.999 |
| | $b_{t,s}$ | 0.47 | 0.011 | 0.45 | 0.48 | 1098 | 0.998 |
| | $b_{t,w}$ | -0.24 | 0.012 | -0.26 | -0.22 | 921 | 1.000 |
| | $\tau$ | 0.86 | 0.008 | 0.85 | 0.88 | 1095 | 0.998 |
| | $a_r$ | 0.00 | 0.014 | -0.02 | 0.02 | 1047 | 0.999 |
| | $b_{r,t}$ | -0.01 | 0.010 | -0.02 | 0.01 | 811 | 0.999 |
| | $b_{r,s}$ | -0.02 | 0.014 | -0.07 | -0.02 | 999 | 0.998 |
| | $b_{r,w}$ | 0.27 | 0.012 | 0.25 | 0.29 | 1050 | 0.999 |
| | $v$ | 0.95 | 0.009 | 0.94 | 0.97 | 1008 | 1.002 |

*Author contributions.* HMH conceived the idea of applying causal modelling, carried out the data analysis, produced the visualisations, and prepared the manuscript. MNM designed the experiment, conducted the data collection, and prepared the manuscript. Both authors formulated the research questions, and contributed to discussions of the results and their implications.



*Competing interests.* The authors declare that no competing interests are present.

*Acknowledgements.* This research has been supported through FORMAS, the Swedish Research Council for Sustainable Development (grant
no. 2018-01144).





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
