# Peer review of "Technical note: A framework for causal inference applied to solar radiation and temperature effects on dissolved gaseous mercury"

_EGUsphere, 2025_

## Referee Comment (RC2)

Review for Manuscript

Title: Technical note: A framework for causal inference applied to solar radiation and temperature effects on dissolved gaseous mercury
Author(s): Hans-Martin Heyn and Michelle Nerentorp Mastromonaco
MS No.: egusphere-2025-4511
MS type: Technical note

**General**

This paper provides a technical note on application of causal inference to the effects of solar radiation and water temperature on dissolved gaseous mercury (DGM). This research is really interesting, instrumental, and insightful.

This research showcases a wonderful collaboration between experimental scientists and causal inference scholars.

What a Wonderful World is this interdisciplinary field.

This paper is expected to embrace a wide range of readers, including those who know or have a good commend of causal inference already and also those who are lay people, well-trained in experimental sciences, yet knowing little about causal inference and how to use and apply it in their experimental science areas. The present reviewer is among the latter group. Hence, this review will focus on two aspects: (1) experimental and (2) how to help and guide the latter group of readers to follow, understand, and learn how to use causal inference by means of the case study provided by this paper. Some readers, if not many, may share the same or similar feedback as presented in this review.

**Specific**

**0. Paper title**

The paper title uses the word "on dissolved gaseous mercury". Perhaps, this term in the context of this research is kind of vague and could be more specific, say, on levels of DGM, or generation process and mechanism of DGM, or speciation of Hg, etc. So it's a bit unclear what is exactly the effect (effect of solar radiation on what exactly, DGM level, dynamics, production?), since DGM itself is only a particular species of aquatic Hg.

**I. Experimental**

Regarding the in-situ field measurement of DGM, a number of questions arise:

First, the citation for this method seems to use a less relevant paper (by Andersson ME et al., 2008b; see L140 in the present paper). I checked on this and found the relevant references probably would be:

(1) A description of an automatic continuous equilibrium system for measurement of dissolved gaseous mercury. By Andersson, Gardfeldt, and Wangberg, Anal. Bioanal. Chem. 391, 2277-2282, 2008a

(2)  Seasonal and spatial evasion of mercury from the western Mediterranean Sea by Nerentorp Mastromonaco, Gardfeldt, and Wangberg, 2017 (L896-897 in the present paper).

Second, with limited time available, I consulted Ref. #2 above and had some findings as detailed below.

Ref. #2 shows that the researchers also used another manual method, i.e., purge-and-trap method, instead of the in-situ auto-method, to determine the DGM. For this manual method, first, the Hg(0) in a water sample of a certain volume is completely purged out of the water sample using zero air (or pure Ar or N2) and then collected on a Hg trap to analyze the total Hg(0) purged out of the water sample. By measuring the volume of the water sample and the total Hg(0) purged from the water sample and collected on the Hg trap, the DGM can thus be calculated to be DGM = (total Hg(0) purged)/(volume of water sample). This method gives a clear determination of the DGM for the water sample without confusion or misunderstanding.

Moreover, Ref. #2 also mentioned that they compared the DGM results from the auto-method and the manual method and found "a good correlation" between the two method results. This means that the DGM calculated using Eq. 1 and the DGM obtained by the manual method differ, although correlated, that is, one may not replace the other, but one can be obtained from another using the correlation.

However, Ref. #2 does not mention or indicate if they used the correlation (or calibration) to get the DGM corresponding to the actual DGM (calibrated by the manual method), or they simply took the DGM results calculated using the equation of DGM = Ca(1/H + ra/rw) (L141 Eq. 1 in the present paper).  This missing detail is a highly important technical detail, which is connected to the credibility of this auto-method, and subsequent causal inference operations and outcomes.

I'd think the correlation (equation) should be reported and used to get the real DGM as calibrated using the correlation, rather than just using the DGM results directly from the calculations using Eq. 1, for the reasons given below. By the way, it's understandable there is a need to have an in-situ auto method to continuously measure DGM in the field.

But, it remains unclear for the present paper under review, all the DGM results used for the causal inference are those directly from the calculation using Eq. 1, or those after processing using the correlation between the auto and manual methods (calibration of the auto-method by the manual method). This important technical detail needs to be clarified.

The auto-method appears not quite straightforward in conjunction with Eq. 1. By the auto-method, a given water volume is first pumped into the inner cylinder. Then (or simultaneously) zero air is used to purge the Hg(0) in the given water to the headspace of the inner cylinder. Then the air concentration of Hg(0) in that headspace is measured by Lumex (or Tekran 2537A). By the way, the efficiency of the purging is not mentioned or discussed in this paper. The efficiency of purging is certainly critical for the manual method. Incomplete purging of the DGM can cause under-estimation of real DGM level.

It is very curious why Eq. 1 is used to calculate the real DGM of the sea water, instead of using the same approach as the manual method to get the total Hg purged out of the water left in the cylinder headspace and then the DGM thus determined. It is also highly curious why the DGM is the Hg(0)

concentration in the water of the cylinder supposedly at equilibrium with the Hg(0) purged out of the same water then present in the headspace measured by Lumex. Intuitively, this is quite confusing and not revealing. The key point here is why the equilibrium of Hg(0) distribution between air and water gets involved in the DGM determination? In any context, it is the real DGM of interest, not the equilibrium DGM.

It is very hard to see and understand how this so-calculated equilibrium Hg(0) concentration can represent the real DGM in the water sample. First of all, the real DGM should be the one at the equilibrium with the ambient air Hg(0) above the sea, rather than with the Hg(0) purged out of the water sample in the cylinder headspace, unless coincidentally, the Hg(0) in the ambient air has the same concentration as the purged Hg(0) in the headspace. It is very hard to see the materialization of such a coincidence, consistently occurring all the time. Or was this coincidence confirmed experimentally?

Using the Henry's law method to get DGM only gives the Hg(0) concentration in the water at the equilibrium, while as known, water is commonly saturated or often over-saturated with Hg(0), i.e., DGM at equilibrium < or << DGM-real.

Table 3 and Fig. 6f all show quite low levels of DGM, as compared to many studies that reported higher DGM levels for various waters. This suspected underestimation of the DGM might be due to that the calculated DGM is only for the equilibrium condition as calculated using Henry's law.

The unclarity and confusion regarding the meaning and credibility of the DGM calculated using Eq. 1 need to be resolved in the first place before readers go further to see any causal inference using the DGM results.

**II. Causal inference general**

Before and during reading this paper for a while, I always thought this causal inference model or operation can determine if two factors given are actually indeed causally related, instead of simply correlated. In other words, the expectation was that by running the causal inference (going through the entire framework and running the causal inference operations or models), it can be determined if one factor is causally related to another, followed further by the effect size.

But, the more I read through, the more I thought or realized (maybe I'm still wrong or doesn't get it) that actually, it seems that to begin the causal inference, one needs to assume, in the first place, the two factors are indeed causally related, and then running the causal inference through the framework would provide more knowledge about the relationship between the two factors, like the effect size, this percentage for this factor, or that percentage for that factor, etc.

So, top front, it would be very helpful to provide a general description of the causal inference, it's goal, logic assumption and framework, approach, what the causal inference is and can or could do, what we can or could expect the causal inference to offer, and moreover, what the causal inference cannot offer or do. This general introduction is much needed. Or, readers, like me, would be struggling in the confusion about if the causal inference can settle the case to determine the causality, or instead, only can provide more inference about the relationship between two or more factors and the effect size of each factor, beyond simple correlation analysis.

So, if the causal inference cannot determine if two or more given factors are indeed causally related, and which is the cause of which (or otherwise), then this nature of the causal inference needs to be stated/indicated clearly in the very beginning. This would help and benefit many readers, like me, who, inference-via-scientific-experiments oriented, probably first time encounter a detailed case like the one provided by this paper. For example, a lot has been known about how solar radiation can causally induce and enhance DGM generation via photochemical reactions by means of well-controlled manipulative experiments (with only one factor tested in variation and other factors fixed to logically satisfy both necessity and sufficiency requirements for causal-effect relationship determination).

**III. Comments and thoughts**

Line 62 (L62), "Hg...water-to-air evaporation", evaporation refers to the escape of molecules of the liquid from liquid phase of that particular molecule to gas phase (e.g., pure water evaporation), but here, there is no liquid Hg involved, only dissolved gaseous Hg or Hg atoms as the solute in water (the solvent), the liquid is water. So rigorously, Hg evasion or emission, not evaporation, is more appropriate or accurate.

By the way, as mentioned before, three issues are involved here: DGM generation, DGM emission or evasion, and DGM concentrations or levels. The title and the paper use "...causal inference applied to solar radiation and temperature effects on DGM. Then, exactly, which factor we are looking at? The DGM generation or emission, or concentration, which are the factors under consideration or treatment with the causal inference? This is unclear, another potential confusion point.

L103-107, the campaign was 2019-2020, but the data used for this study was from 2024 April 1 to April 25. This is another potential confusion point. Which data were used? If the latter, why mentioning the 2019-2020 campaign?

L140-148, all parameters or quantities should be given together with their individual units, if any.

Here, it may be helpful to mention the DGM, Solar, and T data are given or summarized in Table 3 and Fig. 6. At any rate, the data used for this study need to be presented clearly top front, rather than later. We need to know in the first place clearly what are the measurement data used for this study. This data can help readers to see or inspect, now, before the causal inference, the potential causal relationship, intuitively, or based on previous research experiences, independent of the causal inference.

Fig. 6e has no legend, but it has two parameters, which is for which?

L141, from subsequent info, we know ra/rw < 1, this means for Eq. 1, DGM roughly = Ca/H, if so, why leave the item of ra/rw in the equation. This needs to be discussed. When the whole equation is needed, when the approximate, simplified one may be relevant in use. By the way, if the simplified equation is used, then the question regarding the meaning of the so calculated DGM arises, as discussed previously.

Table 3 and Fig. 6 show the DGM levels are quite low, as mentioned before. This is curious.

L169, it is unclear which step in the framework will determine if the two or more factors are causally related, if the causal inference can determine that?

L180-185, it appears that the causal arrow is what we assign or assume before the causal inference, rather than an outcome of the causal inference. This is, among others, what confuses me.

From time to time, this becomes unclear: the casual inference is for solar and Ca or for solar and DGM?

L250-251, regarding the nature of the effect, direct or indirect, again it seems that we need to pre-assign or assume it like the causal arrow, rather than an outcome of the causal inference.

L306-308, how were the simulated data generated? From the data of Table 3 and Fig. 6, or from running the causal inference model? This is unclear. What software used to generate the simulated data?

A general comment, by the way, throughout this paper, it is always unclear if the causal inference was run or conducted by what software or causal inference model(s), any commercial software? If so, unless it is copyright or patent protected and thus cannot be disclosed, we need to know the brands or names of all the software and models used in this study, and which is used in which step to do what. This important info is missing and needs to disclosed in the early beginning as given by a list (like for experimental work, a list of chemicals and equipment used), like in a methodology section for the causal inference.

Furthermore, each time when a specific causal inference operation along the way going through the framework, we'd like to know what specific software or model(s) was used for this specific step or task or operation, with relevant references provided for more technical details.

L390, how to verify?

L498-499, total effect = direct effect + indirect effect, this is valid only for the cases where both effects are positive or negative, i.e., same direction. If one is positive and the other is negative, that total effect sum is not valid, or what is the meaning of that sum? For example, solar effect on T, two effects, one effect is that solar can enhance DGM generation, leading more DGM in water, while on the other hand, the other effect is that solar can increase water T, which in turn can lead to higher Henry's coefficient, and thus less DGM at the higher T, e.g., at Tw = 1 C, DGM at equilibrium = 7.2 pg/L, at 25 C, DGM = 3.8 pg/L. So, the two effects of solar radiation are opposite in direction. Then, how can these two opposite effects be additive in the causal inference? Or how the causal inference handles the opposite effects? Or the direction of the effect does not matter, since the cause inference tells if the effect is operative or not and in what extent?

L582-583, What can the causal inference tell about the factors and their relationships that we still don't know, as from this particular study regarding DGM? In other words, what are new from the causal inference that has not been achieved by scientific experiments and field measurements?

L588-589, pump speed or water flow rate rw, L119 mentions that rw varied between 0 and 40 L/min. Then, first, if rw = 0, rA/rw is meaningless mathematically; if rw = 40, then rA/rw is 1/5/40 = 0.0375, very small, and so this item can be ignored, then DGMcal = Cmw/H. So this pump speed variation largely limits the accuracy of this auto-method. By the way, it remains hard to grasp or understand why DGM-real can be obtained by Cmw(1/H + rA/rw), how equilibrium gets there and why rA and rw

got involved. The first item in Eq. 1 is about equilibrium and the second one is about the dynamics of the sampling flow, and then why DGM involves both equilibrium and dynamics?

The pump speed involves measurement operational error or artifact, and so it is not a real physical effect for DGM like solar and/or Tw. Pump speed is not a direct effect, nor an indirect effect; it just has operational errors. One is about aquatic mechanisms and processes involving DGM generation kinetics and equilibrium and the other is about DGM measurement and measurement errors. Mixing the two in the causal inference is confusing.

32% effect for solar radiation is due to indirect effect of water temperature. But, as mentioned before, the effects of solar and T on DGM are opposite. This result of 32% effect size seems to show that T has a positive effect just like solar radiation, higher solar higher DGM, but higher T, lower DGM based on equilibrium.

By the way, in many cases as shown by many field studies, the water T varied quite less during a day (as compared to solar radiation), only to a small extend as a result of very high specific heat of pure water (due to the Hydrogen bonding of the highly polar water molecules).  But, 32% is almost 1/3, which means the effect of T is almost very strong.

On the other hand, T can not only change Henry's constant and the Hg air/water distribution equilibrium (constant), but also can change the kinetic rate constants (and rates) of photochemical and/or thermal reduction of Hg(II) to Hg(0). This is another effect of water T. Then this effect is positive, enhancing DGM generation, like solar radiation. Thus, T has two opposite effects: positive to enhance the kinetics, and negative to increase H, then decrease DGM at equilibrium.

Last but not the least, it would be helpful to provide a short glossary of the terms as an appendix, especially those involving causal inference.

Many thanks.

---

## Author Comment (AC1)

**Reviewer 1:**

The paper introduces a Bayesian graphical causal inference framework to investigate solar radiation and temperature effects on dissolved gaseous mercury (DGM) concentrations. This is an exciting contribution with clear potential to advance environmental data analysis.

- *We thank the reviewer for this kind comment. We agree that it is our intention with this technical note to advance environmental data analysis in line with other fields that use Causal and Bayesian methods.*

However, major revisions are required to ensure that the method is applied following best practices and clearly communicated to a broader audience in environmental sciences who may not have a statistical background.

- *We agree with the reviewer that we must ensure that our proposed framework follows best practices and is communicated in a way that is applicable, understandable, and useful for a broader audience in environmental science,*

Major Comments

1. Justification for Bayesian Approach

The study does not explicitly demonstrate that frequentist methods fail or that Bayesian inference provides a clear empirical advantage. No comparison is made (e.g., between regression or structural equation models and their Bayesian alternatives) to show instability or bias under a frequentist framework. Since Bayesian methods are technically more complex, the manuscript should clarify when and why they are preferable and under what conditions their use provides meaningful benefits.

- *We thank the reviewer for the opportunity to justify our methodological choice of using a Bayesian approach. We agree that frequentist methods would not necessarily fail. However, we chose a Bayesian Data Analysis approach for three specific advantages that align with our framework:*
  - *Mediation analysis: A core component of our study is estimating indirect effects (Section 5). In a frequentist approach, estimating indirect effects, which involves the product of regression coefficients typically requires strong approximations (Delta method in Sobel test for example), or bootstrapping. The Bayesian approach allows the simple multiplication of posterior samples of the path coefficients to derive the posterior distribution of the indirect effect. We argue that the "up-front" complexity of the Bayesian setup is rewarded with a rigorous and straightforward quantifaction of mediation.*
  - *Formalising expert knowledge: Our framework aims to offer ways to formalise the use of expert knowledge. Bayesian approaches provide a mathematically consistent mechanism to encode physical constraints and domain knowledge for example in the choice of priors.*

- *Regularisation: Even where prior knowledge is limited, the use of weakly informative priors provide regularization. This ensures stability in parameter estimation in situations with strong correlation of predictor variables, such as the relationship between solar radiation and temperature.*
- *We have added a textbox in Section 4.5 summarising these justifications:*

*Section 4.5 Textbox at #386:*

**Justification for the Bayesian approach**

Bayesian statistical modelling allows the explicit quantification of uncertainty due to mediated effects. Such a mediated effect exists in our case due to the mediation of the effect of solar radiation through changing sea water temperature. While conventional frequentist methods require for such quantitation approximations for the product of coefficient (e.g., in the Sobel test) (Yuan and MacKinnon, 2009), Bayesian inference allows to obtain the mediated effect by simply multiplying the posterior samples from the path coefficient (i.e., the effect sizes for each "arrow" on the causal path). Additionally, the Bayesian approach allows the formal inclusion of expert knowledge through priors. And even when prior knowledge is limited, the use of weakly informative priors provide natural regularisation that can stabilise estimates in the presence of correlated predictors (Lemoine, 2019). We therefore argue that the "up-front" complexity of using a Bayesian approach is rewarded with a more rigorous quantification of mediated effects and greater stability in parameter estimation.

**2. Temporal Novelty and Model Structure (#255)**

The authors claim that previous studies suffered from temporal limitations. While this study uses high-frequency data, the model itself does not incorporate time as a structural or dynamic dimension—it treats each time step as an independent observation. The manuscript should clearly explain how this approach differs from earlier studies and whether the higher temporal resolution truly enhances inference or simply provides finer data granularity.

- *We thank the reviewer for this observation. We agree that our models treat time steps as independent observations and that they do not explicitly model temporal dynamics for example through autoregressive terms. We have revised the manuscript in Section 4.1 to clarify that with "temporal limitations" of previous studies, which deployed discrete sampling strategies, we referred to their low sampling frequency rather than any limitations in their modelling strategy.*
- *The high temporal resolution of the automated sampling deployed in our study is not only aiming for a finer granularity of the data, but it is a prerequisite for being able to separate direct and indirect effects for two specific reasons:*
  - *Solar radiation varies on a timescale of minutes, wheres sea surface temperature, as also highlighted by the second reviewer, responds more slowly due to the thermal inertia. We need a high time resolution to distinguish the immediate photochemical effects of sun radiation from the indirect and slower thermal effects. Low-frequency data would collapse the distinct timescales which makes the effects inseparable.*

- - A second, yet more secondary reason is that large sample sizes lead to a more robust converge on posterior distributions, even if the prior assumptions are non-informative or weak.
  - *We reflect these changes in Section 4.1 #309-319:*

 Earlier studies investigating the correlations between DGM concentration, solar radiation and temperature

310 relied on discrete sampling campaigns with limited temporal resolution (Amyot et al., 1997; Gårdfeldt et al., 2001; Dill et al., 2006). However, solar radiation varies on a timescale of minutes, whereas sea surface temperature responds more slowly due to the thermal inertia of water. Low-frequency data collapse these distinct timescales which makes the variables statistically collinear and inseparable. Therefore, in order to separate the direct effect of solar radiation on mercury concentrations from

the indirect effects mediated by sea surface temperature, the data must contain sufficient variability in both the exposure and

315 the mediator. Also, since sea surface temperature already is used to calculate DGM (see equations 1 and 2), $C_{MW}$ was chosen as outcome variable instead of DGM in this study. This study provides data with high temporal resolution from automated long-term measurements of gaseous Hg concentration, solar radiation and surface seawater temperature. By using causal modelling, this study extends prior correlation-based research by quantifying  the direct and indirect effect sizes of solar radiation on Hg

320 concentration in seawater.

3. Distributional Assumption for C_{MW} (#355)

The assumption of a Normal likelihood for C_{MW}is weakly justified. While the Normal distribution is commonly used, its prevalence does not imply appropriateness; the appeal to the Central Limit Theorem oversimplifies environmental concentration data, which are typically multiplicative and right-skewed -- Figure 11(e) shows a long-tailed distribution. The authors could either demonstrate that residuals are approximately normal (supported by residual–fitted value plots) or acknowledge this limitation and discuss whether a log-normal likelihood would be more appropriate.

- *We thank the reviewer for this observation and suggestion regarding the choice of likelihood for $C_{MW}$. We agree that environmental data are often multiplicative and that the Normal distribution. To address the alternative of a log-normal likelihood, we have added the Appendix "Discussion on the distributional assumption for $C_{MW}$. In this appendix, we plotted, as suggested, the residuals of the Normal model against fitted values (Figure G1) and concluded that the plot suggests an increasing variance with the mean of $C_{MW}$.*

  *As a consequence, we implemented a modified model $m_4^{log}$ that uses a log-normal likelihood. We then compared the parameter estimates of the Normal and Log-Normal models by calculating the implied effects on $C_{MW}$, listed in Table G1.*
  *The results of this comparison show that all effect sizes differ by less than 1%, which is why we decided to accept the original Normal likelihood assumption. However, we have updated the main text at Section 4.5 to acknowledge the limitation of the Normal assumption, and we refer to the Appendix for the detailed analysis.*
  *Section 4.5 #429-430:*

Normal distribution and environmental phenomena typically involve the aggregation of a large number of underlying processes. Furthermore, we had no reason to assume another distribution for the outcome $C_{MW}$. However, Appendix G discusses the
430     alternative choice of a log-normal likelihood for $C_{MW}$ that can often be appropriate for environmental data. As part of the

- *New Appendix G #932-962:*

**Appendix G: Discussion on the distributional assumption for $C_{MW}$**

While assuming that the outcome data is normally distributed can be sensible in many cases, environmental data may show a multiplicative and right-skewed character which may also be indicated in the long-tail distribution of the observed $C_{MW}$ data
935     shown in Figure 11 (e). In order to check if the normal likelihood assumption is appropriate for $C_{MW}$, we plotted the residuals of model $m_4$ against its fitted values. The resulting *residual plot* visualises the predication error between each observation and the model's estimate. If the assumption of normal distributed data for $C_{MW}$ holds, the residuals will be symmetrically distributed around zero with a more or less constant spread. However, the plot in Figure G1 suggests that the spread of the residuals is not constant but instead widens as the predicted $C_{MW}$ increases, indicating that the model's error scales with the
940     magnitude of $C_{MW}$. This pattern would justify the adoption of a Log-Normal likelihood which, unlike the Normal likelihood, models a multiplicative and long-tailed distribution nature of $C_{MW}$.

[Figure]

**Figure G1.** Residual plot which shows the prediction error against the predicted $C_{MW}$

.

**Modified model $m_4$ with log-normal likelihood**

As we cannot conclusively exclude the appropriateness of a Log-Normal likelihood for $C_{MW}$, we modified the likelihood of model $m_4$ listed in Table (2) Equation (9) to

945     $C_{MW_i} \sim \text{Log-Normal}(\mu_i, \sigma).$                                                                                  (G1)

This modification provides a new model $m_4^{\log}$ and it entails that the linear predictor equation for $\mu_i$ now defines the mean on the log-scale. Although the mathematical notation of the equation does not change, the interpretation of the coefficients $b_{c,s}$, $b_{c,t}$, $b_{c,w}$, and $b_{c,r}$ is now on a logarithmic scale which needs to be considered when comparing effect sizes later. Similarly, the priors are now applied to multiplicative effects. However, as we use weakly informative priors together with a large number of

950   observations, the specific choice of prior scale is less critical for the posterior estimates.

Table G1 lists the resulting parameter estimates. To compare the parameter estimates thge table also lists the *implied effect size* for each predictor. This allows us to compare the parameter estimates even if the models use different mathematical scales. We calculated the implied effect size for the normal model as a percentage of the mean concentration, given by $a_c$:

$$\textbf{Effect}(\%) = \frac{b_{c,\cdot}}{a_c} \cdot 100. \tag{G2}$$

955   For the Log-Normal model, which is not additive but multiplicative, we calculated the implied effect size for each parameter by directly using the exponential function:

$$\textbf{Effect}(\%) = (e^{b_{c,\cdot}} - 1) \cdot 100. \tag{G3}$$

The comparison suggests that although the log-normal is mathematically more rigorous, it has a very small effect on the

Table G1. Comparison of estimated parameters for $C_{MW}$ between the Normal ($m_4$) and Log-Normal ($m_4^{\log}$) models. The parameters are standardised. The implied effect is the change in % in $C_{MW}$ per 1 SD increase in predictor.

| Parameter | Model | Posterior Mean [90% CI] | Implied Effect (%) [90% CI] | Diff. (pp) |
|---|---|---|---|---|
| Solar Radiation ($b_{c,s}$) | $m_4$ (Normal) | 0.383 [0.369, 0.395] | +16.0 [15.4, 16.5] | 0.2 |
| | $m_4^{\log}$ (Log-Normal) | 0.147 [0.142, 0.152] | +15.8 [15.3, 16.4] | |
| Surface Temp. ($b_{c,t}$) | $m_4$ (Normal) | 0.429 [0.416, 0.441] | +18.0 [17.4, 18.5] | 0.3 |
| | $m_4^{\log}$ (Log-Normal) | 0.168 [0.163, 0.174] | +18.3 [17.7, 19.0] | |
| Wind Speed ($b_{c,w}$) | $m_4$ (Normal) | -0.125 [-0.138, -0.113] | -5.2 [-5.8, -4.7] | 0.4 |
| | $m_4^{\log}$ (Log-Normal) | -0.058 [-0.063, -0.053] | -5.6 [-6.1, -5.2] | |
| Pump Speed ($b_{c,s}$) | $m_4$ (Normal) | 0.265 [0.254, 0.276] | +11.1 [10.6, 11.6] | 0.8 |
| | $m_4^{\log}$ (Log-Normal) | 0.113 [0.109, 0.118] | +11.9 [11.5, 12.5] | |

"pp" denotes percentage points. Implied effects for $m_4$ are approximate based on mean $C_{MW} \approx 2.39$.

parameter estimates, suggesting that the Normal assumption is a sufficient approximation for the data and that therefore the

960   scientific conclusions regarding the effect sizes for mercury concentration are robust to the choice of likelihood. This robustness to the choice of the likelihood may stem from the small standard deviation compared to the average value for the concentration data (Limpert et al., 2001).

**4. Indirect Effects and DAG Interpretation (#520)**

For model m4, the paper discusses indirect effects through Sol → T_S → C_{MW} and Sol → W → C_{MW} but omits the valid multi-step path Sol → T_S → r_W → C_{MW}. The authors should clarify whether such compound mediation effects are included in the total indirect effect and provide clearer guidance on interpreting direct, indirect, and total effects from the DAG.

- *We thank the reviewer for pointing to the compound mediation path Sol → T_S → r_W → C_{MW}. We have revised Section 5.2 (just before Table 4) and added a note in the table to acknowledge this additional path. We clarify that the multi-step path contributes only weakly to the total indirect effect due to the small estimated effect of sea surface temperature on pump speed b_{r,t}, and which, unlike the estimated effect of sol on pump speed b_{r,s}, contains zero in its 90% credible interval (Table F1). Consequently, we do not interpret the compound path Sol → T_S → r_W → C_{MW} as a substantively important mediation mechanism in model m_4.*

could have cleared the inlet of the pump from algae, resulting in a higher flow speed. Unlike model $m_3$, where the indirect

620 effect of solar radiation on $C_{MW}$ is only mediated by the sea surface temperature $T_S$, model $m_4$ allows for two additional

mediation paths: $Sol \rightarrow r_W \rightarrow C_{MW}$, and $Sol \rightarrow T_S \rightarrow r_W \rightarrow C_{MW}$. However, the latter mediation path contributes only very

weakly to the total indirect effect due to the small effect of surface temperature on pump speed ($b_{r,t}$). In contrast to the effect

of solar radiation on pump speed ($b_{r,s}$), which is also relatively small, the credibility interval of the effect size $b_{r,t}$, listed in

Table F1, contains zero. We therefore cannot exclude the possibility that the effect of sea surface temperature on pump speed

625 is practically negligible and consequently do not interpret this compound path as a substantively important part of the total

indirect effect. In summary, the inclusion of the confounding external factor wind $W$ and instrument-intrinsic factor water

**Table 4.** Estimates of the direct, indirect, and total effects based on observed data in April 2020 without (model $m_3$) and with (model $m_4$) recognition of wind and pump speed as external influences. Bold values are mean values; standard deviations are in parentheses; 90% confidence intervals are depicted in square brackets.

| Effect on measured gaseous Hg $C_{MW}$ | Parameters | Standardised value (std. dev.) [90% confidence interval] | | De-standardised value (std. dev.) [90% confidence interval] | | Unit for de-standardised values | Change due to external influences |
|---|---|---|---|---|---|---|---|
| | | Model $m_3$ | Model $m_4$ | Model $m_3$ | Model $m_4$ | | |
| Direct effect of solar radiation $Sol$ | $b_{c,s}$ | **0.360** (0.009) [0.345, 0.374] | **0.383** (0.008) [0.371, 0.395] | **1.75·10$^{-3}$** (4.36·10$^{-5}$) [1.67, 1.81]·10$^{-3}$ | **1.86·10$^{-3}$** (3.67·10$^{-5}$) [1.80, 1.92]·10$^{-3}$ | $\frac{pg}{L} \cdot \left(\frac{W}{m^2}\right)^{-1}$ | +6.3% |
| Indirect effect of $Sol$[1] | $b_{t,s}b_{c,t}+$ $b_{r,s}b_{c,r}$ | **0.189** (0.006) [0.177, 0.197] | **0.186** (0.007) [0.175, 0.198] | **0.92·10$^{-3}$** (2.91·10$^{-5}$) [0.86, 0.96]·10$^{-3}$ | **0.90·10$^{-3}$** (3.66·10$^{-5}$) [0.84, 0.96]·10$^{-3}$ | $\frac{pg}{L} \cdot \left(\frac{W}{m^2}\right)^{-1}$ | -1.8% |
| Total effect of $Sol$[1] | $b_{c,s}+$ $b_{t,s}b_{c,t}+$ $b_{r,s}b_{c,r}$ | **0.549** (0.011) [0.530, 0.564] | **0.572** (0.011) [0.553, 0.590] | **2.65·10$^{-3}$** (5.33·10$^{-5}$) [2.57, 2.73]·10$^{-3}$ | **2.77·10$^{-3}$** (5.44·10$^{-5}$) [2.68, 2.86]·10$^{-3}$ | $\frac{pg}{L} \cdot \left(\frac{W}{m^2}\right)^{-1}$ | +4.5% |
| Direct effect of $T_S$ | $b_{c,t}$ | **0.420** (0.009) [0.405, 0.435] | **0.429** (0.008) [0.417, 0.441] | **1.53·10$^{-1}$** (3.28·10$^{-3}$) [1.48, 1.58]·10$^{-1}$ | **1.56·10$^{-1}$** (2.91·10$^{-3}$) [1.51, 1.61]·10$^{-1}$ | $\frac{pg}{L} \cdot K^{-1}$ | +1.96% |
| Direct effect of wind speed $W$ | $b_{c,w}$ | – | **-0.125** (0.007) [-0.137, -0.114] | – | **-2.87·10$^{-2}$** (1.61·10$^{-3}$) [-3.14, -2.60]·10$^{-2}$ | $\frac{pg}{L} \cdot \left(\frac{m}{s}\right)^{-1}$ | – |
| Direct effect of pump speed $r_W$ | $b_{c,r}$ | – | **0.265** (0.007) [0.253, 0.277] | – | **2.21·10$^{-3}$** (6.67·10$^{-5}$) [2.11, 2.31]·10$^{-3}$ | $\frac{pg}{L} \cdot \left(\frac{L}{min}\right)^{-1}$ | – |

[1]: The additional compound mediation path $Sol \rightarrow T_S \rightarrow r_W \rightarrow C_{MW}$ ($b_{t,s}b_{r,t}b_{c,r}$) is practically negligible due to the small effect $b_{r,t}$ which includes zero in its 90% credibility interval (see Table F1).

5. Limitation of dependence on DAG specification (#665)

The causal conclusions rely on the correctness of the assumed DAG structure in many aspects, in addition to independence, mis-specified relationships or omitted variables - such as unmodeled nonlinear effects or unobserved confounders - could lead to misleading causal inferences. The authors should discuss the potential impact of those DAG misspecification.

- *We thank the reviewer for suggestion to include a discussion on the potential misspecification of causal models. We agree that the causal conclusion derived from observational data depends on the assumed causal structure and that DAGs, as representation for assumed causal structures, can be misspecified through omitted variables, incorrect directions of cause-and-effect, or inadequate functional assumptions which can affect the causal interpretation of the results. We have therefore revised the manuscript to explicitly reflect and discuss these limitations.*

*In Section 3, when introducing the framework for causal inference, we clarify that a key function of graphical causal models is to make the researchers' prior causal assumptions explicit which opens these assumptions to criticism and possible refinement. Furthermore, as part of the discussion in Section 6, we explicitly state that the causal conclusions are conditional on the assumed causal models and that DAGs are not immune to misspecification. We introduced Table 5, which summarises a set of possible DAG misspecifications such as omitted confounders, unmodelled nonlinearities and missing or misdirected edges, discusses their potential impact on the results, and provides general mitigation strategies.*

- *Changes in Section 3 #239-249:*

A key function of the graphical causal model is to make prior assumptions explicit. By explicitly encoding the researchers'
240   prior causal knowledge as DAG they become open to criticism and possible later refinement. Furthermore, it is necessary to define the direction of cause-and-effect a-priori, because statistical models cannot distinguish between cause and effect as they only identify association but not causation. If the direction of cause and effect is not known, or if the existence of a causal relationship is uncertain a-priori, several alternative causal models can be proposed. Based on the proposed causal models, independence criteria are derived using mathematical methods such as d-separation (Pearl et al., 2016). These independence
245   criteria derived from the assumed causal model can later be used to empirically validate the plausibility of the DAG against the observed data by checking for expected associations, or the lack thereof. Causal relations are not discovered from the data directly but evaluated by assessing whether the observed data are consistent with the independence relations implied by the a priori defined causal models. This concept is referred to as the *faithfulness assumption*, i.e., that the observed data follows the independence criteria suggested in the assumed causal graph (Spirtes et al., 2000). Tools exist, such as DAGitty (Textor et al.,
250   2016) that automatically derive these independence criteria from graphical causal models.

- *Section 6.1 #684-688 and Table 5:*

be infeasible. Causal conclusions, however, are conditional on the assumed causal models. DAGs, as graphical representations
685   of causal knowledge, make prior causal knowledge explicit which allows other researchers to understand and criticise more easily the underlying assumptions. Such criticism is important because causal models are not immune to misspecification, such as by omitting unobserved but relevant confounders, leaving out, or misdirecting edges, which may lead to biased effect estimates. Table 5 lists a set of possible misspecifications and their mitigation strategies.

**Table 5.** Potential impacts of DAG misspecification and generalised mitigation strategies.

| Misspecification | Potential Impact | Possible Mitigation Strategies |
|---|---|---|
| Omitted variable | An unobserved and omitted confounder can create a 'back-door' path which can lead to biased effect estimates. | Explicitly documenting assumed causal structures as DAGs allows for easier peer review and criticism. Another strategy can be to determine the required strength of an unobserved confounder to negate an assumed causal relationship. |
| Unmodelled nonlinearity | DAGs themselves do not communicate assumptions about linearity or nonlinearity. Then, especially when using GLM, a linear approximation may miss threshold effects or misrepresent rates of change in complex systems. | The use of posterior predictive checks and visual residual analysis (see Appendix G) can he used to detect systematic misfits. |
| Missing or misdirected edges | Incorrect or missing edges may reverse the interpreted flow of causality which potentially can lead to collider bias (see Appendix D) or incorrect interventions. | The justification of the direction of cause-and-effect using physical laws, temporal precedence, or literature. |

Minor Comments

**1.** #330

The priors (e.g., Normal(0.5, 1), Normal(0.5, 0.5)) appear somewhat arbitrary and not elicited from domain experts. **The study would be strengthened by (a) justifying these priors through expert input or empirical reasoning, or (b) using uninformative priors.**

- *We thank the reviewer for raising the point regarding the justification for the choice of priors. We have revised the paragraph that provides the rationale and the role of the used priors in Section 4.5.*
*Specifically, we now explicitly state that the priors are weakly informative rather than expert-elicted or non-informative, and we explain why this choice is appropriate for our analysis. We further added a clarification that uninformative priors are not generally preferable in applied regression models, and we refer to recent methodological work that recommends weakly informative prior as a principled default in BDA (Lemoine, 2019). Finally, we also emphasise that the plausibility of the priors was asses using prior predictive simulations.*

- *Section 4.5 #4481-454:*

440 **Priors**

In general, a *prior* tells researchers what assumptions are made about a parameter before they see any observed data. These assumptions can range from highly informative, where the distribution encodes strong prior beliefs about the parameter values, through weakly informative priors that provide mild regularisation, to non-informative priors that have very little influence on the posterior distribution. It is important to note that priors are continuously updated with the available observed data. With

445 each iteration in BDA, the posterior will be used as new prior for the next iteration. That means, that the more data is available, the less influence prior beliefs have. With each iterative update, the prior distribution will be more influenced by the data distribution and therefore become increasingly dominated by the likelihood. In BDA, weakly informative priors are preferred in applied regression modelling because they provide mild regularisation (Lemoine, 2019). This prevents, for example, extreme parameter values, while at the same time allowing the data to shape the posterior distribution.

450     For the models of this study, we used weakly informative priors for all parameters. Because all predictor variables were standardised, such that the coefficients represent effects on a common scale, we used Normal priors with modest location and scale parameters that encode a coarse, "order-of-magnitude" expectation about plausible effect sizes and allowing both positive and negative effect sizes. Furthermore, the Normal distribution can represent a wide range of shapes from perfectly symmetric to slightly skewed which makes it a suitable choice if no other strong information is available about the shape of the prior

455 distribution. We assumed exponential distributions for the parameters related to the variances because these must always be positive. The plausibility of the priors was assessed using prior predictive simulations for our models $m_1$ to $m_4$, which are presented as supplementary material in Appendix C.

2. #445

   Please clarify how model convergence was assessed under the Bayesian MCMC framework. Including trace plots or diagnostics is important for verifying convergence. A useful reference is: *Reich, Brian J., and Sujit K. Ghosh. Bayesian Statistical Methods. Chapman and Hall/CRC, 2019.*

   - *We thank the reviewer for suggesting to improve the documentation of the model convergence under the Bayesian MCMC framework in the manuscript. We have revised the manuscript accordingly to explicitly describe how we assessed convergence, including visual trace plots. In Section 4.9 (Paragraph Workability) we refer to a new appendix section (Appendix H) that presents trace plots and provides a detailed discussion of the convergence assessment.*
   - *Section 4.9 #503-504:*

500 statistical toolboxes such as rethinking or the underlying Stan library may raise when evaluating the posterior distributions of the models. As part of the model validation for the proposed models we provide the $\hat{R}$-values and effective sample sizes for each model together with the detailed inference results in Appendix F. We also checked for warnings of divergent transitions while training the models on the data. A detailed discussion of the convergence assessment, including visual trace plots for the effect size parameters, can be found in Appendix H.

- *New Appendix H #964-975:*

**Appendix H:  Workability: Assessing the convergence under Bayesian MCMC**

965 We conducted Bayesian inference using a Markov Chain Monte Carlo (MCMC) sampling approach with Hamiltonian Monte Carlo implemented in Stan (Stan Development Team) using the rethinking interface by McElreath (2020). We assessed convergence using both quantitative diagnostics, including $\hat{R}$ and the effective samples size (ESS/$n_{\text{eff}}$) as well as visual diagnostics, following standard recommendation for Bayesian workflows (Vehtari et al., 2021; Reich and Ghosh, 2019). First, the $\hat{R}$ values for all parameters were close to 1 and $< 1.01$ as reported in Table F1. Second, all parameters have effective sample sizes

970 (ESS/$n_{\text{eff}}$) exceeding 10% of the total sample sizes which we assume sufficiently large (see Vehtari et al. (2021) and Furia et al. (2022) for a discussion on the sufficient ESS for BDA). Finally, we visually inspected the trace plots to verify adequate mixing, absence of strange divergent behaviour and stationarity. The trace plots are provided in Figure H1 and show no indication of non-convergence such as slow trends, chain separation or autocorrelation. Together, these diagnostics provide evidence that the MCMC chains converged.

[Figure]

**Figure H1.** Trace plots for all effect size parameters of models $m_1$ (red), $m_2$ (orange), $m_3$ (green), and $m_4$ (blue).

3. #445

   Both R2 and WAIC are reported and appear consistent. However, if they diverged, how should this be interpreted? A short explanation of their conceptual difference would improve clarity.

   - *We thank the reviewer for this suggestion. We added in Section 4.9 a brief clarification on the difference between R^2 (in-sample explanatory fit) and WAIC (expected out-of-sample predictive accuracy estimate). We also discuss how a potential divergence can be interpreted.*

   - *Addition made to Section 4.9 #518-521:*

adequacy of an individual model. Instead, information criteria such as $WAIC$ can be used to compare models against each other. We provide $WAIC$ scores for all models as part of the model evaluation in Section 5.3 and in Figure 12. Wheras WAIC provides an expected out-of-sample predictive accuracy estimate, the coefficient of determination $R^2$ summarises in-sample
520  explanatory fits. In this analysis, WAIC and $R^2$ are consistent for the models, but if they were to diverge, it would indicate that a model either fits the observed data well but generalises poorly, or generalises well but shows a reduced in-sample fit.

**4.** #605

Figure 13(b) seems to show narrower confidence intervals than (a), but this is hard to discern. The figure could be redesigned for better contrast. Also, revise the phrasing "noisier but also more reliable," as "noisier" typically suggests lower precision.

- *We thank the reviewer for suggesting to improve Figure 13 (b). We have revised the Figure to improve contrast and interpretability by plotting now the posterior mean regression functions and their associated 95% posterior credible intervals instead of the earlier posterior predictive simulations which included observational noise. This change allows for a better direct visual comparison of the effect sizes. We have also revised the corresponding Section 6.1 to remove the "noiser" term and to clarify why model $m_4$ provides a less biased and more causally interpretable estimate by accounting for mediating and confounding processes.*

- *Revised Figure 13:*

[Figure]

**Figure 13.** Comparison of posterior mean regressions of the effect of solar radiation *Sol* on mercury concentration $C_{MW}$ using model $m_1$ (dotted line) and model $m_4$ (solid line). Grey shading indicates the 95% confidence interval. Vertical reference lines at *Sol*= 600 and 800 W/m² are the solar radiation levels used to compare effect sizes between the models.

- *Section 6.2 (former 6.1) #711-721:*

**6.2 Implications for future mercury research and policies**

705 As mentioned in Section 1.1, several studies report an observed significant correlation between measured gaseous mercury and solar radiation. However, considering the regression only between these two factors results in a very simple model, comparable to our model $m_1$ (Figure 4 (a)). This model is not comprehensive enough to allow for drawing correct causal conclusions. In contrast, model $m_4$ (Figure 5) explicitly incorporates both mediation by sea surface temperature ($T_S$), confounding by wind speed ($W$), and an instrument-intrinsic influence through the pump speed ($r_W$). The model is more reliable because it reduces

710 bias from additional competing effects, confounders, and background conditions that, if ignored, would give a misleading picture of the underlying causal relationship. Figure 13 illustrates the practical implication of this difference. For an increase in solar radiation from 600 W/m$^2$ to 800 W/m$^2$, model $m_1$ predicts an increase in measured mercury of about 0.56 pg/L. In contrast, the causally adjusted model $m_4$ predicts a smaller increase of only about 0.42 pg/L. Thus, the estimated effect size of solar radiation on mercury emission is about 25% lower if causal relationships are accounted for. In summary, if causal

715 relationships are ignored, there is a risk of overestimating the effect of solar radiation on gaseous mercury.

**5.** #615

The rationale for preferring graphical causal models over alternatives (e.g., Granger causality, potential outcomes) is generally sound. Graphical models do enhance transparency and facilitate the integration of mechanistic knowledge. However, they do not eliminate assumptions or guarantee correctness. Traditional causal frameworks are not inherently "non-transparent" but rely on different theoretical foundations. Acknowledging this nuance would make the argument more balanced.

- *We thank the reviewer for suggesting a more balanced comparison between graphical causal models and other causal frameworks. We have therefore revised Section 6.2 to clarify that alternative causal frameworks are not inherently non-transparent but instead formalise assumptions using different constructs such as exchangeability assumptions. We further emphasise that the primary contribution of graphical causal models lies in making prior causal assumptions explicit and inspectable rather than removing prior assumptions altogether.*

- *Section 6.3 (former 6.2) #731-737:*

730 hand, allow to encode prior assumptions transparently such that the necessary restricting conditions for causal inference from observational data are provided. This does not mean that graphical causal models remove the need for prior assumptions, nor do they guarantee the correctness or completeness of prior causal knowledge. As with other causal frameworks, such as potential outcome frameworks or Granger causality, the validity of any causal claim depends on the underlying prior assumptions and the adequacy of the data. Other causal frameworks are not inherently "non-transparent" but they use different, and often more

735 implicit, mechanisms to communicate prior assumptions such as exchangeability assumptions (Hernán and Robins, 2020) or stationarity requirements. In this sense, the primary contribution of graphical causal models is to offer a particularly explicit and inspectable representation of prior causal knowledge. The importance of defining prior causal knowledge as graphical causal models has been recognised in other scientific disciplines, such as medicine (Glass et al., 2013), economy (Imbens, 2020), social science (Imbens, 2024), and software engineering (Furia et al., 2019). Scientists in these fields proposed a set

**6.** #805

Appendix E Figure E1, used to validate statistical independence, could be clearer. Adding fitted lines with distinct colors for different temperature levels would improve readability and interpretation.

- *We thank the reviewer for suggesting adding fitted lines with distinct colours to the scatter plots. We have revised Figure E1 accordingly.*

[Figure]

**Figure E1.** Scatter plots for inspecting suggested independence between variables for (a) $m_1$: $T_S \perp C_{MW} \mid Sol$ and (b) $m_2$: $Sol \perp C_{MW} \mid T_S$.

---

## Author Comment (AC2)

**Reviewer 2:**

**General**

This paper provides a technical note on application of causal inference to the effects of solar radiation and water temperature on dissolved gaseous mercury (DGM). This research is really interesting, instrumental, and insightful.

This research showcases a wonderful collaboration between experimental scientists and causal inference scholars.

What a Wonderful World is this interdisciplinary field.

This paper is expected to embrace a wide range of readers, including those who know or have a good commend of causal inference already and also those who are lay people, well-trained in experimental sciences, yet knowing little about causal inference and how to use and apply it in their experimental science areas. The present reviewer is among the latter group. Hence, this review will focus on two aspects: (1) experimental and (2) how to help and guide the latter group of readers to follow, understand, and learn how to use causal inference by means of the case study provided by this paper. Some readers, if not many, may share the same or similar feedback as presented in this review.

- *We thank the reviewer for the kind words about our paper. Indeed, one of the aims of the paper was to demonstrate successful collaboration between two different research disciplines; something the research community needs more of. The paper is hopefully useful to a wide range of readers and aims to inform, demonstrate and inspire future research collaborations between research fields.*

**Specific**

1. **Paper title**

The paper title uses the word "on dissolved gaseous mercury". Perhaps, this term in the context of this research is kind of vague and could be more specific, say, on levels of DGM, or generation process and mechanism of DGM, or speciation of Hg, etc. So it's a bit unclear what is exactly the effect (effect of solar radiation on what exactly, DGM level, dynamics, production?), since DGM itself is only a particular species of aquatic Hg.

- *We are thankful for the nice suggestions to make the title more informative and clearer. We suggest a change of the title to "Technical note: A framework for casual inference applied to solar radiation and temperature effects on measured levels of gaseous elemental mercury in seawater." The title in the paper has been changed accordingly, see below:*

**Technical note: A framework for causal inference applied to solar radiation and temperature effects on measured levels of gaseous elemental mercury in seawater**

**2. Experimental**

Regarding the in-situ field measurement of DGM, a number of questions arise:

First, the citation for this method seems to use a less relevant paper (by Andersson ME et al., 2008b; see L140 in the present paper). I checked on this and found the relevant references probably would be:

- A description of an automatic continuous equilibrium system for measurement of dissolved gaseous mercury. By Andersson, Gardfeldt, and Wangberg, Anal. Bioanal. Chem. 391, 2277-2282, 2008a

- Seasonal and spatial evasion of mercury from the western Mediterranean Sea by Nerentorp Mastromonaco, Gardfeldt, and Wangberg, 2017 (L896-897 in the present paper).

- *We agree with the reviewer that the two suggested references fit better to L140 than the previous added reference. The suggested Ref#1 has now replaced the previous reference on L140 in the revised paper. Also, another reference has been added: "Gårdfeldt et al., 2002.: "Comparison of procedures for measurements of dissolved gaseous mercury in seawater performed on a Mediterranean cruise." Analytical and Bioanalytical chemistry 374.6 (2002): 1002-1008.", which further describes how the equations used to calculate DGM, using the same continuous system as used in this study, were derived and verified.*

- *Section 2.1 #154-155:*

$C_{MW}$ measured with the analyser can be used to calculate the concentration of dissolved gaseous mercury (DGM) in incoming seawater. If $C_{a0}$ is removed from Hg, the equation to calculate DGM can be simplified to: (Andersson et al., 2008a; 155 Gårdfeldt et al., 2002):

Second, with limited time available, I consulted Ref. #2 above and had some findings as detailed below.

Ref. #2 shows that the researchers also used another manual method, i.e., purge-and-trap method, instead of the in-situ auto-method, to determine the DGM. For this manual method, first, the Hg(0) in a water sample of a certain volume is completely purged out of the water sample using zero air (or pure Ar or N2) and then collected on a Hg trap to analyze the total Hg(0) purged out of the water sample. By measuring the volume of the water sample and the total Hg(0) purged from the water sample and collected on the Hg trap, the DGM can thus be calculated to be DGM = (total Hg(0) purged)/(volume of water sample). This method gives a clear determination of the DGM for the water sample without confusion or misunderstanding.

- *We thank the reviewer for opening this interesting discussion. We believe that it is indeed true that the discrete manual purge and trap method earlier has been the most common method for analysing DGM in water samples. The reviewer is right, when using the manual method, the approach is to completely purge the sample. DGM is then calculated by dividing the amount of purged Hg by the sample*

*volume. We do agree with the reviewer that this method is more straight-forward and leaves less confusion regarding the experimental performance. However, discrete sampling would not have been an appropriate method to use for our study. The number of data points needs to be large for the statistical significance of our model, and a high time resolution of Hg analyses is crucial to match fast changes in solar radiation. Using a manual method would require an immense workload and would result in an insufficient time resolution that is needed for this study.*

Moreover, Ref. #2 also mentioned that they compared the DGM results from the auto-method and the manual method and found "a good correlation" between the two method results. This means that the DGM calculated using Eq. 1 and the DGM obtained by the manual method differ, although correlated, that is, one may not replace the other, but one can be obtained from another using the correlation.

- *We agree with the reviewer that Ref#2 (also Ref #1and Gårdfeldt et al., 2002) compared DGM results from the automated- and manual methods with good correlations. Although this is a nice discussion point, it was not part of our paper to compare manual and automated methods to measure DGM in surface water.*

However, Ref. #2 does not mention or indicate if they used the correlation (or calibration) to get the DGM corresponding to the actual DGM (calibrated by the manual method), or they simply took the DGM results calculated using the equation of DGM = Ca(1/H + ra/rw) (L141 Eq. 1 in the present paper). This missing detail is a highly important technical detail, which is connected to the credibility of this auto-method, and subsequent causal inference operations and outcomes.

- *We understand the reviewer's concern regarding the confusion. As mentioned before, a rigorous comparison between the automated and manual sampling was performed in Ref#1. To our knowledge, Ref#2 compared the two methods at five measurement points, showing some agreement between the methods. The comparison was performed to check and compare the two methods, not to calibrate them against each other. Ref #2 used equation 1 to calculate DGM concentrations in their paper, an equation that has been used to calculate DGM in many other papers, without the necessity to re-calculate the values using calibration against the manual method.*

I'd think the correlation (equation) should be reported and used to get the real DGM as calibrated using the correlation, rather than just using the DGM results directly from the calculations using Eq. 1, for the reasons given below. By the way, it's understandable there is a need to have an in-situ auto method to continuously measure DGM in the field.

But, it remains unclear for the present paper under review, all the DGM results used for the causal inference are those directly from the calculation using Eq. 1, or those after processing using the correlation between the auto and manual methods (calibration of the auto-method by the manual method). This important technical detail needs to be clarified.

- *We agree with the reviewer that the automated method would benefit from being compared to other field methods to measure and calculate DGM concentrations in seawater (although, to our knowledge, not many other methods exist yet). However, in our study, the calculated DGM concentrations were not used in the causal model. For the model we used measured (not calculated) Hg concentrations (CMW) for comparing with solar radiation and temperature. The reason for this choice was because Henry's law constant, which is used in equation 1 to calculate DGM, is temperature dependent (see equation 2) and therefore, calculated DGM cannot be used in the causal model when comparing DGM to temperature since it would cause an uncontrollable feedback loop. Although we believe equation 1 to be correct for calculating DGM, we only used this equation to calculate DGM values used for demonstration and comparison in Table 3 and Figure 6.*

*We are sorry for the confusion. Throughout the text we have now tried to be clearer that we are studying measured gaseous elemental Hg (CMW) rather than DGM. In Section 2.1 we also added an explanation why we made this choice, see below:*

*Section 2.1 #159-163:*

$r_w$ denote the flow rates of purging air and seawater (l/min), respectively. When studying Equations (1) and (2) it becomes
160    clear that sea water temperature is already integrated into the calculation of DGM, which can cause uncontrollable feedback loops when studying direct effects between DGM and sea surface temperature in our model. To avoid this problem, $C_{MW}$ was chosen as a outcome variable instead of DGM in this study. Calculated DGM concentrations, which in this study only are presented for comparison, are presented in Table 3 and Figure 6 (f) in Section 5.1.

The auto-method appears not quite straightforward in conjunction with Eq. 1. By the auto-method, a given water volume is first pumped into the inner cylinder. Then (or simultaneously) zero air is used to purge the Hg(0) in the given water to the headspace of the inner cylinder. Then the air concentration of Hg(0) in that headspace is measured by Lumex (or Tekran 2537A). By the way, the efficiency of the purging is not mentioned or discussed in this paper. The efficiency of purging is certainly critical for the manual method. Incomplete purging of the DGM can cause under-estimation of real DGM level.

<y curious why Eq. 1 is used to calculate the real DGM of the sea water, instead of using the same approach as the manual method to get the total Hg purged out of the water left in the cylinder headspace and then the DGM thus determined. It is also highly curious why the DGM is the Hg(0) concentration in the water of the cylinder supposedly at equilibrium with the Hg(0) purged out of the same water then present in the headspace measured by Lumex. Intuitively, this is quite confusing and not revealing. The key point here is why the equilibrium of Hg(0) distribution between air and water gets involved in the DGM determination? In any context, it is the real DGM of interest, not the equilibrium DGM.

It is very hard to see and understand how this so-calculated equilibrium Hg(0) concentration can represent the real DGM in the water sample. First of all, the real DGM should be the one at the equilibrium with the ambient air Hg(0) above the sea, rather than with the Hg(0) purged out of the water sample in the cylinder headspace, unless

coincidentally, the Hg(0) in the ambient air has the same concentration as the purged Hg(0) in the headspace. It is very hard to see the materialization of such a coincidence, consistently occurring all the time. Or was this coincidence confirmed experimentally?

Using the Henry's law method to get DGM only gives the Hg(0) concentration in the water at the equilibrium, while as known, water is commonly saturated or often over-saturated with Hg(0), i.e., DGM at equilibrium < or << DGM-real.

- *We understand that this confusion remained, but we hope that we now have explained our choice of using equation 1 to calculate DGM concentrations used for demonstration and comparison with other studies. In manual sampling, Henry's law is not needed, as it is needed in Equation 1 when using the automated method. Henry's law constant, that shows how much a gas dissolves in water at equilibrium, is used in equation 1 to compensate for the choice of measuring equilibrium concentrations rather than purging the total amount of Hg in the sample, as in manual methods. This approach of measuring equilibrium concentrations is also used for measuring other gases in water, such as $CO_2$, using similar methods and equations, using Henry's law constant for $CO_2$ in seawater. Hence, the theory behind this method is not original but well studied. See for example "Wanninkof, R. and K. Thoning (1993) Measurement of fugacity of $CO_2$ in surface water using continuous and discrete sampling methods. Mar. Chem. 44: 189- 204".*

Table 3 and Fig. 6f all show quite low levels of DGM, as compared to many studies that reported higher DGM levels for various waters. This suspected underestimation of the DGM might be due to that the calculated DGM is only for the equilibrium condition as calculated using Henry's law.

The unclarity and confusion regarding the meaning and credibility of the DGM calculated using Eq. 1 need to be resolved in the first place before readers go further to see any causal inference using the DGM results.

- *We thank the reviewer for demonstrating that a comparison with literature is missing in our paper. The calculated DGM concentrations (that are in this paper only presented for comparison and not used in the analysis) showed an average concentration of 14 (5-28) pg/L. For comparison, surface DGM was measured using an in situ purging system in mars/April 2015, also at the Swedish west coast at Råö/Rörvik station, about 160 km south of Kristineberg. Here, the average DGM concentration was 13 pg/L, which is in good agreement with our results (Nerentorp Mastromonaco PhD thesis, 2016). A literature review has now been added in section 5.1, see below:*

the pump speed $r_W$ and the measured Hg concentration $C_{MW}$. Calculated DGM, shown in Figure 6 (f), show similar diurnal patterns as for $C_{MW}$. The average concentration during the measurement period was 14 pg/l (Table 3). During the summers in 1997 and 1998, Gårdfeldt et al. (2001) measured DGM by manual sampling at 20 cm depth in open seawater, about 1 km from the Kristineberg Marine Research Station, resulting in DGM concentrations varying between 40-100 pg/L. However, it differs about 20 years between their and our measurements. More recent continuous measurements of DGM, performed in spring 2015 at the Råö/Rörvik station in Sweden (about 160 km south of Kristineberg), showed an average DGM surface concentration of 13 pg/l (Mastromonaco, 2016), which is in good agreement with our study. The literature review presented in Mastromonaco et al. (2017) show surface DGM concentrations varying between 11 to 32 pg/l in the Baltic Sea (15-20 pg/l in spring), 11 to 52 pg/l in the North Sea, 12 pg/l in the North Atlantic Ocean (summer) and about 20 to 30 pg/l in the Mediterranean Sea.

3. **Causal inference general**

Before and during reading this paper for a while, I always thought this causal inference model or operation can determine if two factors given are actually indeed causally related, instead of simply correlated. In other words, the expectation was that by running the causal inference (going through the entire framework and running the causal inference operations or models), it can be determined if one factor is causally related to another, followed further by the effect size.

But, the more I read through, the more I thought or realized (maybe I'm still wrong or doesn't get it) that actually, it seems that to begin the causal inference, one needs to assume, in the first place, the two factors are indeed causally related, and then running the causal inference through the framework would provide more knowledge about the relationship between the two factors, like the effect size, this percentage for this factor, or that percentage for that factor, etc.

- *We thank the reviewer for giving this insightful comment about the lacking information regarding how to interpret and understand how the causal model works. It is indeed true that for the model to work you need to first have an idea how and if two factors are related. That's why it is important to draw your DAGs correct before running the model and interpret the results. That is what we describe in the paper to be prior scientific knowledge. We have improved the description of what causal models can do and what they cannot do throughout the manuscript.*

A key function of the graphical causal model is to make prior assumptions explicit. By explicitly encoding the researchers' prior causal knowledge as DAG they become open to criticism and possible later refinement. Furthermore, it is necessary to define the direction of cause-and-effect a-priori, because statistical models cannot distinguish between cause and effect as they only identify association but not causation. If the direction of cause and effect is not known, or if the existence of a causal relationship is uncertain a-priori, several alternative causal models can be proposed. Based on the proposed causal models, independence criteria are derived using mathematical methods such as d-separation (Pearl et al., 2016). These independence criteria derived from the assumed causal model can later be used to empirically validate the plausibility of the DAG against the observed data by checking for expected associations, or the lack thereof. Causal relations are not discovered from the data directly but evaluated by assessing whether the observed data are consistent with the independence relations implied by the a priori defined causal models. This concept is referred to as the *faithfulness assumption*, i.e., that the observed data follows the independence criteria suggested in the assumed causal graph (Spirtes et al., 2000). Tools exist, such as DAGitty (Textor et al., 2016) that automatically derive these independence criteria from graphical causal models.

**6.1 What causal inference adds beyond experiments and field observations**

The causal framework in this study did not aim to discover previously unknown physical processes governing the formation of gaseous mercury in the oceans. Instead, the contribution lies in *quantifying how known processes jointly contribute to observed variability under observational conditions* outside of a laboratory. Specifically, using the suggested causal framework, it is possible to (i) separate total observed association between solar radiation and measured mercury into direct and temperature-mediated components, (ii) quantify the relative importance of these causal pathways, and (iii) adjust effect estimates for confounding influences such as environmental influences and instrument-intrinsic factors that are difficult to control in field observations. While laboratory and field experiments showed that solar radiation and sea surface temperature influence mercury emissions, the proposed causal framework allows these effects to be estimated simultaneously from observational data under explicitly and transparently stated causal assumptions. This causal inference technique therefore provides effect size estimates that are directly interpretable for large-scale modelling efforts or policy assessments, where controlled experiments may be infeasible. Causal conclusions, however, are conditional on the assumed causal models. DAGs, as graphical representations of causal knowledge, make prior causal knowledge explicit which allows other researchers to understand and criticise more easily the underlying assumptions. Such criticism is important because causal models are not immune to misspecification, such as by omitting unobserved but relevant confounders, leaving out, or misdirecting edges, which may lead to biased effect estimates. Table 5 lists a set of possible misspecifications and their mitigation strategies.

hand, allow to encode prior assumptions transparently such that the necessary restricting conditions for causal inference from observational data are provided. This does not mean that graphical causal models remove the need for prior assumptions, nor do they guarantee the correctness or completeness of prior causal knowledge. As with other causal frameworks, such as potential outcome frameworks or Granger causality, the validity of any causal claim depends on the underlying prior assumptions and the adequacy of the data. Other causal frameworks are not inherently "non-transparent" but they use different, and often more implicit, mechanisms to communicate prior assumptions such as exchangeability assumptions (Hernán and Robins, 2020) or stationarity requirements. In this sense, the primary contribution of graphical causal models is to offer a particularly explicit and inspectable representation of prior causal knowledge. The importance of defining prior causal knowledge as graphical causal models has been recognised in other scientific disciplines, such as medicine (Glass et al., 2013), economy (Imbens, 2020), social science (Imbens, 2024), and software engineering (Furia et al., 2019). Scientists in these fields proposed a set

So, top front, it would be very helpful to provide a general description of the causal inference, it's goal, logic assumption and framework, approach, what the causal inference is and can or could do, what we can or could expect the causal inference to offer, and moreover, what the causal inference cannot offer or do. This general introduction is much needed. Or, readers, like me, would be struggling in the confusion about if the causal inference can settle the case to determine the causality, or instead, only can provide more inference about the relationship between two or more factors and the effect size of each factor, beyond simple correlation analysis.

So, if the causal inference cannot determine if two or more given factors are indeed causally related, and which is the cause of which (or otherwise), then this nature of the causal inference needs to be stated/indicated clearly in the very beginning. This would help and benefit many readers, like me, who, inference-via-scientific-experiments oriented, probably first time encounter a detailed case like the one provided by this paper. For example, a lot has been known about how solar radiation can causally induce and enhance DGM generation via photochemical reactions by means of well-controlled manipulative experiments (with only one factor tested in variation and other factors fixed to logically satisfy both necessity and sufficiency requirements for causal-effect relationship determination).

- *We are very grateful for this concrete suggestion to improve the paper. We have added clarifications to the Introduction (Section 1) and to the outline of the proposed framework in Section 3. In particular, we highlight that our suggested framework does not establish causality from observational data alone. Instead, causal inference means, in this context, estimating direct and indirect effect sizes conditional on explicitly stated causal assumptions encoded as graphical causal models following the methodology outlined by Pearl et al. (2016). We hope that the additional information helps set the reader's expectation early and to clarify how the proposed framework goes beyond usual correlation analysis while remaining conditional on the correctness and completeness of prior scientific knowledge. For example, we knew before writing this paper that solar radiation can induce DGM generation by photochemical processes. We also knew that temperature affect DGM in some way, but what we did not know was how it was all connected. Is it rather that solar radiation affects the measured mercury concentration ($C_{MW}$) indirectly by temperature increase alone? This we early realized by running the model that this is not true. Then arose the question, how much of the $C_{MW}$ generation is affected by only solar radiation and only temperature? In this model we could "turn off/lock" the effect of one factor to see how much the other factor affected $C_{MW}$ and vice versa. With the help of the model we could apply lab experiments on real measured data in the field. And this is the strength of this framework.*

- *Section 1 #44-52:*

40  machines that magically predict future data points from observational data. Instead, they are particularly interested in understanding cause-effect relationships to suggest interventions that reduce pollutants in the environment. Causal knowledge, or in other words the analysis of cause-effect relationships, is one of the 'fundamental goals of science' (Vowels et al., 2022; Rose and van der Laan, 2011).

Pearl et al. (2016) highlight that causal questions, i.e., question about what are causes and effects, usually cannot be answered
45  from observational data alone. Instead, additional assumptions are needed that specify an assumed causal structure underlying the data-generating process. Causal inference from observational data is therefore not assumption-free. Its conclusions depend on the correctness and completeness of the prior knowledge represented as graphical causal model. Accordingly, the framework presented in this paper does not aim to discover causal structure from data alone, nor does it aim to provide a definitive proof of causation. Instead, its scope is to offer a transparent and principled way to reason about causal effect sizes using observational
50  environmental data and prior knowledge, and to assess the compatibility of that prior knowledge with the observed data. By making prior knowledge and assumptions about cause-and-effect relationships explicit as graphical models, causal conclusions drawn from observational data can be scrutinised, criticised, and revised.

This paper reports the results of a case study on extracting causal knowledge about the contribution of different environmental

- *Section 3 #195-202:*

observational environmental data and prior scientific knowledge from researchers. Here, causal relationships are not discovered from the observational data itself, but are assumed based on prior experimental and scientific knowledge, such as laboratory studies demonstrating photochemical DGM production under controlled conditions. The scope of the proposed framework is then to quantify how multiple established or assumed causal processes jointly contribute to observed variability under natural,
200  intervention-free field conditions. Using causal models, as suggested in this framework, conceptually allows individual causal pathways to be "switched off" within the model. This allows assessing the causal pathways' relative contribution without the need to physically intervene in the environmental system, which often is impossible in field observation.

**III. Comments and thoughts**

Line 62 (L62), "Hg...water-to-air evaporation", evaporation refers to the escape of molecules of the liquid from liquid phase of that particular molecule to gas phase (e.g., pure water evaporation), but here, there is no liquid Hg involved, only dissolved gaseous Hg or Hg atoms as the solute in water (the solvent), the liquid is water. So rigorously, Hg evasion or emission, not evaporation, is more appropriate or accurate.

- *We thank the reviewer for noting this mistake and suggesting improvements. We have changed the word "evaporation" to "evasion" instead in the paper. See for example #70-71:*

accounts for almost 50% of the annual contributions to the atmospheric mercury load. This is because much of the oceans'
70   surfaces are supersaturated with elemental mercury compared to the atmosphere, resulting in net water-to-air evasion  (AMAP, 2021). Understanding the drivers behind formation of dissolved gaseous mercury (DGM) and subsequent

By the way, as mentioned before, three issues are involved here: DGM generation, DGM emission or evasion, and DGM concentrations or levels. The title and the paper use "...causal inference applied to solar radiation and temperature effects on DGM. Then, exactly, which factor we are looking at? The DGM generation or emission, or concentration, which are the factors under consideration or treatment with the causal inference? This is unclear, another potential confusion point.

- *We are grateful that the reviewer pointed out this confusion point. We have taken the suggestion (stated in the beginning of this review) from the reviewer to change the title of the paper to make it clearer. The title is now "Technical note: A framework for casual inference applied to solar radiation and temperature effects on measured levels of gaseous elemental mercury in seawater." We have also replaced "DGM" with gaseous elemental mercury (CMW), where appropriate in the text. Examples of changes in the paper are presented below:*

- *Abstract #10 and #12:*

10   effect sizes of solar radiation and sea surface temperature on levels of  gaseous elemental mercury $(C_{MW})$  in seawater measured at the west coast of Sweden. Our causal analysis reveals that 32% of the total effect of solar radiation on $(C_{MW})$  is mediated indirectly via changes in sea surface temperature. Wind and instrumentation intrinsic factors biased

- *Introduction #54-55:*

This paper reports the results of a case study on extracting causal knowledge about the contribution of different environmental processes to the observed levels of  gaseous elemental mercury $(C_{MW})$ in seawater
55   . Although measurements of gaseous mercury in water is not yet a requirement within any EU directive,

L103-107, the campaign was 2019-2020, but the data used for this study was from 2024 April 1 to April 25. This is another potential confusion point. Which data were used? If the latter, why mentioning the 2019-2020 campaign?

- *We thank the reviewer for noticing this error that simply was a typing mistake. The real period for the measurement period is "2020-04-01 to 2020-04-25". This has been changed accordingly in the paper, see below:*

- *Section 2 #115:*

the Skagerrak Sea which is classified as a natural reserve. With its shallow waters it serves as an important reproduction site

115    for shellfish. The data for this study were collected during the period 2020-04-01 to 2020-04-25, which is an interesting time

L140-148, all parameters or quantities should be given together with their individual units, if any.

- *The reviewer is right, and we are grateful for pointing this out. We have added units for the factors presented in the equations, accordingly, see below:*

- *Section 2.1 #157 and #159:*

where $C_{MW}$ is the measured Hg concentration in the air outflow from the purging system (pg/l) , $H'$ is the dimensionless Henry's law constant that describes the partitioning of mercury between the gaseous and aqueous phase. The variables $r_A$ and $r_w$ denote the flow rates of purging air and seawater (l/min), respectively. When studying Equations (1) and (2) it becomes

160    clear that sea water temperature is already integrated into the calculation of DGM, which can cause uncontrollable feedback

Here, it may be helpful to mention the DGM, Solar, and T data are given or summarized in Table 3 and Fig. 6. At any rate, the data used for this study need to be presented clearly top front, rather than later. We need to know in the first place clearly what are the measurement data used for this study. This data can help readers to see or inspect, now, before the causal inference, the potential causal relationship, intuitively, or based on previous research experiences, independent of the causal inference.

- *We thank the reviewer for pointing this out. Although we decided not to present any measurement results in the method part, we now added information in section 2 about where the data is presented so it will be easier for the reader to find it, see examples below:*

- *Section 2 #116-117:*

115    for shellfish. The data for this study were collected during the period 2020-04-01 to 2020-04-25, which is an interesting time period for our case study due to the good mixture between dark and sunlit hours in Scandinavia at this time of the year. All data are presented in Table 3 and Figure 6 in Section 5.1.

Fig. 6e has no legend, but it has two parameters, which is for which?

- *We thank the reviewer for this suggestion. We added a legend to Figure 6(e).*

[Figure]

(e)

L141, from subsequent info, we know ra/rw < 1, this means for Eq. 1, DGM roughly = Ca/H, if so, why leave the item of ra/rw in the equation. This needs to be discussed. When the whole equation is needed, when the approximate, simplified one may be relevant in use. By the way, if the simplified equation is used, then the question regarding the meaning of the so calculated DGM arises, as discussed previously.

- *We agree with the reviewer that it's true that when rw is much bigger than ra, this term in the equation is small (in our case in average the factor would be 1.5/2.8 = 0.5). The reason why we feel it necessary to present the two factors in equation 1 is because we in our model discuss the influence of the water flow on measured CMW. The reviewer is again right with the point that the meaning of the calculated DGM is pointless for our study since we use CMW in the causal model. However, we think that many readers would find it interesting to compare calculated DGM concentrations in this study with other studies, since CMW is not a factor commonly reported.*

Table 3 and Fig. 6 show the DGM levels are quite low, as mentioned before. This is curious.

- *The reviewer is again pointing out a good point that no literature comparison was presented in the paper. Although the reviewer finds the DGM concentration rather low, we do believe that the calculated DGM concentrations are in good agreement with other studies. A comparison with literature has been added to section 5.1, see below.*

- *Section 5.1 #537-545:*

the pump speed $r_W$ and the measured Hg concentration $C_{MW}$. Calculated DGM, shown in Figure 6 (f), show similar diurnal patterns as for $C_{MW}$. The average concentration during the measurement period was 14 pg/l (Table 3). During the summers in 1997 and 1998, Gårdfeldt et al. (2001) measured DGM by manual sampling at 20 cm depth in open seawater, about 1 km from
540    the Kristineberg Marine Research Station, resulting in DGM concentrations varying between 40-100 pg/L. However, it differs about 20 years between their and our measurements. More recent continuous measurements of DGM, performed in spring 2015 at the Råö/Rörvik station in Sweden (about 160 km south of Kristineberg), showed an average DGM surface concentration of 13 pg/l (Mastromonaco, 2016), which is in good agreement with our study. The literature review presented in Mastromonaco et al. (2017) show surface DGM concentrations varying between 11 to 32 pg/l in the Baltic Sea (15-20 pg/l in spring), 11 to
545    52 pg/l in the North Sea, 12 pg/l in the North Atlantic Ocean (summer) and about 20 to 30 pg/l in the Mediterranean Sea.

L169, it is unclear which step in the framework will determine if the two or more factors are causally related, if the causal inference can determine that?

- *We thank the reviewer for this question. Causal inference itself cannot determine if two or more factors are causally related from observational data alone. It is necessary to define a priori causal assumptions in the form of causal models as suggested in our framework. However, it is possible to check if the a priori causal assumptions encoded in the causal model "fits" to the observed data. If two factors are causally not(!) related, they are also independent (except in very very very rare circumstances in which two opposing causal effects exactly cancel each other out). As the causal model provides (automatically) a set of independence criteria between variables, these criteria can be checked against*

*the observed data. If they match, the model is said to be "faithful (see Spirtes et al., 2000) to the data. We clarify this concept now explicitly in Section 3:*

- *Section 3 #244-249:*

independence criteria are derived using mathematical methods such as d-separation (Pearl et al., 2016). These independence
245 criteria derived from the assumed causal model can later be used to empirically validate the plausibility of the DAG against
the observed data by checking for expected associations, or the lack thereof. Causal relations are not discovered from the data
directly but evaluated by assessing whether the observed data are consistent with the independence relations implied by the a
priori defined causal models. This concept is referred to as the *faithfulness assumption*, i.e., that the observed data follows the
independence criteria suggested in the assumed causal graph (Spirtes et al., 2000). Tools exist, such as DAGitty (Textor et al.,
250 2016) that automatically derive these independence criteria from graphical causal models.

L180-185, it appears that the causal arrow is what we assign or assume before the causal inference, rather than an outcome of the causal inference. This is, among others, what confuses me.

- *We thank the reviewer for this bringing up this point. The reviewer is right in that causal arrows in the proposed framework are specified a priori and are not an outcome of the causal inference itself because it is not possible to directly estimate the direction of cause-and-effect from data alone (a computer cannot distinguish associations from causations). In the proposed framework, the a priori causal models provide a qualitative specification of assumed cause-effect directions based on domain knowledge and experimental evidence. Causal inference, as suggested in our framework, then provides the quantitative effect sizes conditional on this assumed causal structure, and it evaluates, via the earlier discussed independence criteria, whether the observational data are consistent with the qualitative causal model. We have clarified this distinction now in Section 3 of the manuscript:*

- *Section 3 #219-220 and #222-224:*

with the arrow → indicating that solar radiation is a cause of changes in surface temperature, and not the other way around.
Note that in this framework, causal arrows are not inferred from data but represent a priori assumptions about cause-effect
220 directions derived from domain knowledge or experimental evidence. The graphical representation of causal models through
DAGs is qualitative, i.e., it provides information about the direction of cause-and-effects between variables, but it does not
provide information about the strength or functional properties of the causal relationships. Causal inference, as proposed in
this framework, provides the quantification of effect sizes and it evaluates whether the observational data is consistent with the
assumed qualitative causal structure. Pearce and Lawlor (2016) provide an overview of properties of DAGs representing causal
225 models:

From time to time, this becomes unclear: the casual inference is for solar and Ca or for solar and DGM?

- *The reviewer is right, and we thank the reviewer for pointing out this issue since this was causing confusion in our paper. We have now added information about our choice to use CMW instead of DGM in our model and have changed the text throughout the paper to be clear that we used and studied CMW, not DGM, in the model, see text below:*

- *Section 2.1 #159-163:*

$r_w$ denote the flow rates of purging air and seawater (l/min), respectively. When studying Equations (1) and (2) it becomes

160 clear that sea water temperature is already integrated into the calculation of DGM, which can cause uncontrollable feedback loops when studying direct effects between DGM and sea surface temperature in our model. To avoid this problem, $C_{MW}$ was chosen as a outcome variable instead of DGM in this study. Calculated DGM concentrations, which in this study only are presented for comparison, are presented in Table 3 and Figure 6 (f) in Section 5.1.

L250-251, regarding the nature of the effect, direct or indirect, again it seems that we need to pre-assign or assume it like the causal arrow, rather than an outcome of the causal inference.

- *We thank the reviewer for raising this point about the nature of direct and indirect effect and the role of causal models. We agree that the classification of effects as direct or indirect is not discovered automatically by the causal inference itself, but is rather defined a priori by the assumed causal model. In our proposed framework, the DAG specifies in a transparent way which causal paths are assumed to exist, and thereby also if an effect acts directly or indirectly on an outcome. Then, our with such an a priori assumed causal model, our frameworks allows to estimate the magnitude of the corresponding direct and indirect effects, conditional on the assumed causal model.*
*To clarify this potential ambiguity, we have added a clarification in Section 4.1:*

- *Section 4.1 #304-306:*

Causal models allow us to distinguish between a direct effect, which includes the part of the total effect of a forcing that acts immediately on an outcome, and the indirect effect, which accounts for the share of the effect size that is mediated through another factor. In other words, the distinction between direct and indirect effects is defined with respect to an a priori assumed

305 causal graph. With such a graph, our frameworks estimates the magnitude of these effects *conditional on the specified causal paths* which usually cannot be identified from observational data alone.

L306-308, how were the simulated data generated? From the data of Table 3 and Fig. 6, or from running the causal inference model? This is unclear. What software used to generate the simulated data?

A general comment, by the way, throughout this paper, it is always unclear if the causal inference was run or conducted by what software or causal inference model(s), any commercial software? If so, unless it is copyright or patent protected and thus cannot be disclosed, we need to know the brands or names of all the software and models used in this study, and which is used in which step to do what. This important info is missing and needs to disclosed in the early beginning as given by a list (like for experimental work, a list of chemicals and equipment used), like in a methodology section for the causal inference.

Furthermore, each time when a specific causal inference operation along the way going through the framework, we'd like to know what specific software or model(s) was used for this specific step or task or operation, with relevant references provided for more technical details.

- *We thank the reviewer for this important comment regarding the transparency about the used software and code. We have clarified in the manuscript that the simulated data were generated by forward-sampling from generative models that represent the assumed causal models. The simulations were used only to verify that the statistical models can recover known parameters and were not used as a substitute for inference from observational data later in the paper. We have clarified this in Section 4.4., including a statement listing the software packages used for creating the simulated data.*
  *In addition, based on the reviewer's recommendation, we have added a dedicated paragraph at the beginning of Section 4 that lists the software and modelling tools used throughout the causal inference workflow. We emphasise also that the full software implementation, including simulation code, model specifications, diagnostics, and creation of the visualisations, are publicly accessible in the replication package which hopefully supports full reproducibility of our results.*

- *New paragraph in Section 4 #289-294:*

  **Software and implementation**

  All steps of the framework were implemented using open-source software. DAGs and implied conditional independence rela-
  290   tions were derived using `DAGitty` (Textor et al., 2016). Bayesian statistical models were specified in R with the `rethinking`
  package (McElreath, 2020) and `Stan` (Stan Development Team) as underlying inference engine. Data preprocessing and visu-
  alisations were performed in both R and `Python` using standard specific libraries. No commercial causal inference software
  or simulation software was used. All code and data required to reproduce the steps of the causal framework are provided in the
  replication package accompanying this manuscript.

- *Section 4.4 #371-375:*

  370   **4.4   Step 4: Generate simulated data based on causal models and identified independence criteria.**

  Simulated data were generated for each of the proposed causal models. Each simulated dataset was generated from a data-
  generating process using forward-sampling with fixed parameter values that reflect the causal assumptions encoded in the
  DAGs. As software, we used R with the `rethinking` package and `Stan` as underlying inference engine to implement the
  generative models. The simulations serve as a verification step to test if the statistical models can recover known parameters
  375   under assumed causal structure. They do not serve as a substitute for inference on observational data. Further details and results
  of the simulation are presented as supplementary material in Appendix B.

L390, how to verify?

- *We thank the reviewer for this question. In Step 6, we added more details about the verification process, including when we considered verification to be successful.*

- *Section 4.6 #460-463:*

  **4.6   Step 6: Verify the models on the simulated data.**

  In this step we show that the models can estimate the parameters set for the simulation and identify independence relations
  460   in simulated data. Each model was verified on the simulated data sets created in Step 4 by comparing the posterior parameter
  estimates with the known parameter values used in the data-generating process. The verification was considered successful if
  the posterior means recovered the true parameter values and if parameters corresponding to absent causal paths were estimated
  close to zero. The results of the parameter estimates for all models under simulated data are given in Appendix B.

L498-499, total effect = direct effect + indirect effect, this is valid only for the cases where both effects are positive or negative, i.e., same direction. If one is positive and the other is negative, that total effect sum is not valid, or what is the meaning of that sum? For example, solar effect on T, two effects, one effect is that solar can enhance DGM generation, leading more DGM in water, while on the other hand, the other effect is that solar can increase water T, which in turn can lead to higher Henry's coefficient, and thus less DGM at the higher T, e.g., at Tw = 1 C, DGM at equilibrium = 7.2 pg/L, at 25 C, DGM = 3.8 pg/L. So, the two effects of solar radiation are opposite in direction. Then, how can these two opposite effects be additive in the causal inference? Or how the causal inference handles the opposite effects? Or the direction of the effect does not matter, since the cause inference tells if the effect is operative or not and in what extent?

- *We thank the reviewer for raising this important conceptual point. The total effect is the sum of direct and indirect effects. If some factors are negative and some positive, some of the effect would cancel each other out. The total effect would then be the sum that is left. This definition holds regardless of the sign of the individual causal paths. We have clarified this in Section 5.2.*
  *We agree that calculated DGM is indeed negatively influenced by seawater temperature, as evident when studying Equation 1 and 2. However, we further clarify that the causal model in this study is specified at the level of measured mercury concentration $C_{MW}$, rather than an isolated subprocess such as equilibrium partitioning governed by Henry's law. Empirically, the inferred effect of seawater temperature on $C_{MW}$ is positive in the observational data, indicating that temperature-related processes in this measurement context dominate the sub-mechanisms described by Henry's law. We have clarified this aswell in the manuscript under Section 5.2.*

- *Section 5.2 #581-582:*

580   Indirect Effect$_{Sol \to C_{MW}} = b_{t,s} \cdot b_{c,t}.$                                                (13)

In summary, a part of the association between *Sol* and $C_{MW}$ "flows" via $T_S$. The total effect of *Sol* on $C_{MW}$ is the sum of direct effect and indirect effect. This definition holds regardless of the sign of the individual path-specific effects: indirect effects with opposite signs represent competing causal mechanisms that (partially) can cancel each other out.

  *#591-595:*

$m_1$, is the sum of the direct and indirect effects, thus in fact the total effect of *Sol* on $C_{MW}$. Although individual mechanisms, such as the temperature dependence of Henry's law, may suggest opposing effects on equilibrium DGM, the causal model in this study is specified for measured mercury concentrations $C_{MW}$. Empirically, the inferred effect $b_{c,t}$ of seawater temperature $T_S$ on $C_{MW}$ is positive in the observational data, which suggest that temperature-related processes in this measurement context
595   are stronger than the opposing sub-mechanisms.

L582-583, What can the causal inference tell about the factors and their relationships that we still don't know, as from this particular study regarding DGM? In other words, what are new from the causal inference that has not been achieved by scientific experiments and field measurements?

- *We thank the reviewer for this important question. We have added a subsection in the Discussion outlining the novelty and contribution of causal inference within*

*mercury emissions from oceans in particular and environmental research in general. We clarify in this subsection that the novelty of causal inference does not lie in identifying new physical mechanisms but in quantifying and decomposing the effect of already hypothesised, or in lab experiments discovered, physical drivers but using only observational and intervention-free data. The insight our proposed framework provide go beyond correlation analyses and complement therefore experimental studies by providing effect size estimates that are valid under explicitly, and transparently, stated causal assumptions. The new subsection also contain a discussion on the limitation*

- *New Section 6.1 #680-695:*

**6.1 What causal inference adds beyond experiments and field observations**

The causal framework in this study did not aim to discover previously unknown physical processes governing the formation of gaseous mercury in the oceans. Instead, the contribution lies in *quantifying how known processes jointly contribute to observed variability under observational conditions* outside of a laboratory. Specifically, using the suggested causal framework, it is possible to (i) separate total observed association between solar radiation and measured mercury into direct and temperature-mediated components, (ii) quantify the relative importance of these causal pathways, and (iii) adjust effect estimates for confounding influences such as environmental influences and instrument-intrinsic factors that are difficult to control in field observations. While laboratory and field experiments showed that solar radiation and sea surface temperature influence mercury emissions, the proposed causal framework allows these effects to be estimated simultaneously from observational data under explicitly and transparently stated causal assumptions. This causal inference technique therefore provides effect size estimates that are directly interpretable for large-scale modelling efforts or policy assessments, where controlled experiments may be infeasible. Causal conclusions, however, are conditional on the assumed causal models. DAGs, as graphical representations of causal knowledge, make prior causal knowledge explicit which allows other researchers to understand and criticise more easily the underlying assumptions. Such criticism is important because causal models are not immune to misspecification, such as by omitting unobserved but relevant confounders, leaving out, or misdirecting edges, which may lead to biased effect estimates. Table 5 lists a set of possible misspecifications and their mitigation strategies.

**Table 5.** Potential impacts of DAG misspecification and generalised mitigation strategies.

| Misspecification | Potential Impact | Possible Mitigation Strategies |
|---|---|---|
| Omitted variable | An unobserved and omitted confounder can create a 'back-door' path which can lead to biased effect estimates. | Explicitly documenting assumed causal structures as DAGs allows for easier peer review and criticism. Another strategy can be to determine the required strength of an unobserved confounder to negate an assumed causal relationship. |
| Unmodelled nonlinearity | DAGs themselves do not communicate assumptions about linearity or nonlinearity. Then, especially when using GLM, a linear approximation may miss threshold effects or misrepresent rates of change in complex systems. | The use of posterior predictive checks and visual residual analysis (see Appendix G) can be used to detect systematic misfits. |
| Missing or misdirected edges | Incorrect or missing edges may reverse the interpreted flow of causality which potentially can lead to collider bias (see Appendix D) or incorrect interventions. | The justification of the direction of cause-and-effect using physical laws, temporal precedence, or literature. |

L588-589, pump speed or water flow rate rw, L119 mentions that rw varied between 0 and 40 L/min. Then, first, if rw = 0, rA/rw is meaningless mathematically; if rw = 40, then rA/rw is 1/5/40 = 0.0375, very small, and so this item can be ignored, then DGMcal = Cmw/H. So this pump speed variation largely limits the accuracy of this auto-method. By the way, it remains hard to grasp or understand why DGM-real can be obtained by Cmw(1/H + rA/rw), how equilibrium gets there and why rA and rw got involved. The first

item in Eq. 1 is about equilibrium and the second one is about the dynamics of the sampling flow, and then why DGM involves both equilibrium and dynamics?

- *We thank the reviewer for discussing this issue further. The second term in equation 1 is present due to the design of the system where the contact time between water and air is crucial to determine if the system is in steady state or not. This is important when calculating the efficiency of the system of how much mercury can be extracted from the system. The flow rates of air and water do affect the calculated DGM, as the reviewer also has noted with the above comment. Since equation 1 to calculate DGM only is used for demonstration in our study (and not used in the causal study), we advise the reviewer and the reader to further explore the derivation of the equation where it is originally explained in Andersson et al. 2008a.*

The pump speed involves measurement operational error or artifact, and so it is not a real physical effect for DGM like solar and/or Tw. Pump speed is not a direct effect, nor an indirect effect; it just has operational errors. One is about aquatic mechanisms and processes involving DGM generation kinetics and equilibrium and the other is about DGM measurement and measurement errors. Mixing the two in the causal inference is confusing.

- *We thank the reviewer for raising the importance of differentiating between real physical processes of mercury emissions and measurement-related artefacts. We agree that pump speed does not represent a physical process. To address this concern, we modified the terminology throughout the entire manuscript to consistently refer to pump speed as instrument-intrinsic factor. The causal analysis remains focused on disentangling the effect of environmental processes, such as solar radiation and sea surface temperature. However, the statistical models recognise the disturbing influence a varying pump speed.*

32% effect for solar radiation is due to indirect effect of water temperature. But, as mentioned before, the effects of solar and T on DGM are opposite. This result of 32% effect size seems to show that T has a positive effect just like solar radiation, higher solar higher DGM, but higher T, lower DGM based on equilibrium.

By the way, in many cases as shown by many field studies, the water T varied quite less during a day (as compared to solar radiation), only to a small extend as a result of very high specific heat of pure water (due to the Hydrogen bonding of the highly polar water molecules). But, 32% is almost 1/3, which means the effect of T is almost very strong.

On the other hand, T can not only change Henry's constant and the Hg air/water distribution equilibrium (constant), but also can change the kinetic rate constants (and rates) of photochemical and/or thermal reduction of Hg(II) to Hg(0). This is another effect of water T. Then this effect is positive, enhancing DGM generation, like solar radiation. Thus, T has two opposite effects: positive to enhance the kinetics, and negative to increase H, then decrease DGM at equilibrium.

- *It is a very interesting point raised by the reviewer and we thank the reviewer for this nice discussion point. We agree that calculated DGM is negatively correlated*

*with temperature, via the calculation of Henry's law coefficient, see equations 1 and 2. However, in our study, we removed this issue when choosing to instead study the measured Hg concentration (CMW). Our findings in this study were that our measured CMW showed a positive correlation to measured seawater temperature where an increase of 1K would lead to an increase in CMW of 0.156 pg/l (see Table 4). Our study also showed that the fraction of this temperature effect, that was associated with the indirect effect of solar radiation affecting the temperature, was 32%, which can be observed when looking at the standardised values of model m4 in Table 4: 0.186 (indirect effect of Sol mediated by sea surface temperature) versus 0.429 (the total effect of Ts) = 32%. This interestingly shows that the temperature effect on CMW can be explained by the indirect effect on solar radiation to only 32%. Other effects of temperature on CMW could be, as the reviewer mentioned, for example changing kinetic rates of abiotic, biotic and thermal reduction processes.*

Last but not the least, it would be helpful to provide a short glossary of the terms as an appendix, especially those involving causal inference.

- *We thank the reviewer for this suggestion. We have added a short glossary in appendix.*

   *New Appendix I #976-994:*

**Appendix I: Glossary of terms from causal inference and mercury chemistry**

**I1  Causal inference related terms**

**Causal inference**  is the estimation of effect sizes under explicit assumptions about the causal structure underlying the data.

**Causal models**  are an explicit specification of assumed cause-effect relationships between variables in the data.

980 **Confounder**  is a variable that causally influences both an exposure and an outcome of interest which can lead to biased effect estimates.

**Conditional independence**  is the independence between two variables given a third variable.

**d-separation**  is a graphical method on DAGs for deriving conditional independence relations from a causal model.

**Directed acyclic graphs (DAGs)**  are a graphical representation of a causal model in which nodes represent variables and directed edges causal directions.

985 **Direct effect**  is the component of an effect that is represented by a direct causal path between two variables.

**Indirect effect**  is the component of an effect that is mediated by one more more intermediated variables.

**Total effect**  is the sum of direct and indirect effects.

**I2  Mercury related terms**

**Dissolved gaseous mercury (DGM)**  is gaseous mercury species dissolved in water.

990 **Elemental mercury ($Hg^0$)**  is the volatile, gaseous form of mercury.

**Measured gaseous mercury ($C_{MW}$)**  is the concentration of elemental mercury measured in the gas phase extracted from seawater.

**Mercury evasion**  is the emission of elemental mercury from seawater into the atmosphere.

**Sea surface temperature ($T_S$)**  is the temperature of surface seawater at the influx to the measurement device.

**Solar radiation (*Sol*)**  is the incoming radiation from the sun measured at the experiment side.